# DATACOMP:
# In search of the next generation of multimodal datasets

**Samir Yitzhak Gadre\*[2], Gabriel Ilharco\*[1], Alex Fang\*[1], Jonathan Hayase[1],**
**Georgios Smyrnis[5], Thao Nguyen[1], Ryan Marten[7,9], Mitchell Wortsman[1],**
**Dhruba Ghosh[1], Jieyu Zhang[1], Eyal Orgad[3], Rahim Entezari[10], Giannis Daras[5],**
**Sarah Pratt[1], Vivek Ramanujan[1], Yonatan Bitton[11], Kalyani Marathe[1],**
**Stephen Mussmann[1], Richard Vencu[6], Mehdi Cherti[6,8], Ranjay Krishna[1],**
**Pang Wei Koh[1,12], Olga Saukh[10], Alexander Ratner[1,13], Shuran Song[2],**
**Hannaneh Hajishirzi[1,7], Ali Farhadi[1], Romain Beaumont[6],**
**Sewoong Oh[1], Alex Dimakis[5], Jenia Jitsev[6,8],**
**Yair Carmon[3], Vaishaal Shankar[4], Ludwig Schmidt[1,6,7]**

## Abstract

Multimodal datasets are a critical component in recent breakthroughs such as CLIP, Stable Diffusion and GPT-4, yet their design does not receive the same research attention as model architectures or training algorithms. To address this shortcoming in the machine learning ecosystem, we introduce DATACOMP, a testbed for dataset experiments centered around a new candidate pool of 12.8 billion image-text pairs from Common Crawl. Participants in our benchmark design new filtering techniques or curate new data sources and then evaluate their new dataset by running our standardized CLIP training code and testing the resulting model on 38 downstream test sets. Our benchmark consists of multiple compute scales spanning four orders of magnitude, which enables the study of scaling trends and makes the benchmark accessible to researchers with varying resources. Our baseline experiments show that the DATACOMP workflow leads to better training sets. Our best baseline, DATACOMP-1B, enables training a CLIP ViT-L/14 from scratch to 79.2% zero-shot accuracy on ImageNet, outperforming OpenAI's CLIP ViT-L/14 by 3.7 percentage points while using the same training procedure and compute. We release DATACOMP and all accompanying code at www.datacomp.ai.

## 1 Introduction

Recent advances in multimodal learning such as CLIP [111], DALL-E [115, 116], Stable Diffusion [123], Flamingo [8], and GPT-4 [103] offer unprecedented generalization capabilities in zero-shot classification, image generation, and in-context learning. While these advances use different algorithmic techniques, e.g., contrastive learning, diffusion, or auto-regressive modeling, they all rest on a common foundation: large datasets containing paired image-text examples. For instance, CLIP's training set contains 400 million image-text pairs, and Stable Diffusion was trained on the two billion examples from LAION-2B [129]. This new generation of image-text datasets is 1,000 times larger than previous datasets such as ImageNet, which contains 1.2M images [37, 126].

Despite the central role of image-text datasets, little is known about them. Many state-of-the-art datasets are proprietary, and even for public datasets such as LAION-2B [129], it is unclear how design choices such as the data source or filtering techniques affect the resulting models. While there are thousands of ablation studies for algorithmic design choices (loss function, model architecture, etc.), datasets are often treated as monolithic artifacts without detailed investigation. Moreover,

---

\*Equal contribution, randomly ordered. Correspondence to contact@datacomp.ai. [1]University of Washington [2]Columbia University [3]Tel Aviv University [4]Apple [5]UT Austin [6]LAION [7]AI2 [8]Juelich Supercomputing Center, Research Center Juelich [9]University of Illinois Urbana-Champaign [10]Graz University of Technology [11]Hebrew University [12]Google Research [13]Snorkel AI

37th Conference on Neural Information Processing Systems (NeurIPS 2023) Track on Datasets and Benchmarks.

Table 1: Zero-shot performance of CLIP models trained on different datasets. DATACOMP-1B, assembled with a simple filtering procedure on image-text pairs from Common Crawl, leads to a model with higher accuracy than previous results while using the same number of multiply-accumulate operations (MACs) or less during training. See Section 3.5 for details on the evaluation datasets.

| Dataset | Dataset size | # samples seen | Architecture | Train compute (MACs) | ImageNet accuracy |
|---|---|---|---|---|---|
| OpenAI's WIT [111] | 0.4B | 13B | ViT-L/14 | $1.1 \times 10^{21}$ | 75.5 |
| LAION-400M [128, 28] | 0.4B | 13B | ViT-L/14 | $1.1 \times 10^{21}$ | 72.8 |
| LAION-2B [129, 28] | 2.3B | 13B | ViT-L/14 | $1.1 \times 10^{21}$ | 73.1 |
| LAION-2B [129, 28] | 2.3B | 34B | ViT-H/14 | $6.5 \times 10^{21}$ | 78.0 |
| LAION-2B [129, 28] | 2.3B | 34B | ViT-g/14 | $9.9 \times 10^{21}$ | 78.5 |
| DATACOMP-1B (ours) | 1.4B | 13B | ViT-L/14 | $1.1 \times 10^{21}$ | **79.2** |

datasets currently lack the benchmark-driven development process that has enabled a steady stream of improvements on the model side and isolates data enhancements from changes to the model. These issues impede further progress in multimodal learning, as evidenced by recent work showing that public datasets currently do not match the scaling behavior of proprietary alternatives [28].

In this paper, we take a step towards a more rigorous dataset development process. Our first and central contribution is **DATACOMP, a new benchmark for multimodal dataset design**. DATACOMP flips the traditional benchmarking paradigm in machine learning where the dataset is fixed and researchers propose new training algorithms. Instead, we hold the entire training code and computational budget constant so that participants innovate by proposing new training sets. To evaluate the quality of a training set, we score the resulting model with a testbed of 38 classification and retrieval tasks such as ImageNet [37], ImageNetV2 [121], DTD [30], EuroSAT [63], SUN-397 [146], and MSCOCO [26].

DATACOMP focuses on two key challenges that arise when assembling large training datasets: what data sources to train on, and how to filter a given data source. Each challenge corresponds to one track in our benchmark. To facilitate the *filtering track*, our second contribution is **COMMONPOOL, a dataset of 12.8B image-text pairs collected from Common Crawl** and currently the largest public image-text dataset. We release CommonPool as an index of image url-text pairs under a CC-BY-4.0 license, and apply content checks in its construction to remove unsafe or unwanted content. In the *filtering track*, the goal of participants is to find the best subset of COMMONPOOL to train on. In the second track, *Bring Your Own Data* (BYOD), participants may leverage any data source, as long as it does not overlap with our evaluation testbed.

Our third contribution is an investigation of **scaling trends for dataset design**. In particular, DATACOMP contains *four* scales, where we vary the training budget and the candidate pool size from 12.8M to 12.8B samples (see Table 2). Expressed in GPU hours, the cost of a single training run ranges from 4 to 40,000 GPU hours on the A100 cluster we used for development. The different scales enable researchers with different resources to participate in our benchmark. Moreover, our results show that the ranking of filtering approaches is largely consistent across scale.

Our fourth contribution is **over three hundred baseline experiments**, including techniques such as querying captions for relevant keywords, filtering based on image embeddings, and applying a threshold on CLIP scores. A key result from our baselines experiments is that smaller, more stringently filtered datasets can lead to models that generalize *better* than larger datasets coming from the same pool. At the 12.8B scale, our best filtering baseline increases ImageNet zero-shot accuracy by 6.9 percentage points (pp) relative to the unfiltered pool (see Table 3). For the BYOD track, our initial experiments show that 109M additional data points (less than 1% of the 12.8B pool) improve the CLIP-filtered subsets of COMMONPOOL by up to 1.2 pp ImageNet accuracy (see Table 18).

Finally, our fifth contribution is **DATACOMP-1B, a new state-of-the-art multimodal dataset**. We obtain DATACOMP-1B by combining our two most promising filtering baselines. DATACOMP-1B enables training a CLIP ViT-L/14 model to an ImageNet zero-shot accuracy of 79.2% (see Table 1), corresponding to a $9\times$ computational cost reduction when compared to a larger CLIP ViT-g/14 model trained on LAION-2B for about $3\times$ longer. Moreover, our model outperforms OpenAI's original CLIP ViT-L/14 by 3.7 percentage points, while using the same compute budget.

To make DATACOMP a shared environment for controlled dataset experiments, we publicly release our candidate pool url index, our tooling for assembling these pools, our filtering baselines, and our

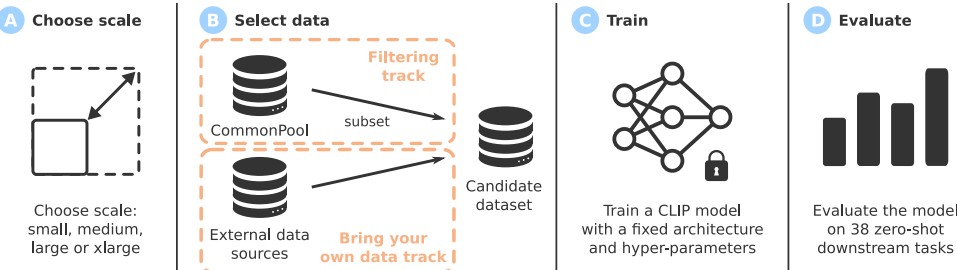

Figure 1: DATACOMP participant workflow. A) Choose a scale based on resource constraints. B) Design a dataset, in either the filtering or BYOD track. C) Train a CLIP model on the designed dataset using a fixed architecture and hyperparameters (Section 3.4). D) Evaluate the trained model on a suite of diverse downstream tasks (Section 3.5).

code for training and evaluating models at `www.datacomp.ai`. We believe that our infrastructure will help put research on dataset design on rigorous empirical foundations, draw attention to this understudied research area, and lead to the next generation of multimodal datasets.

## 2 Related Work

We review the most closely related work and include additional related work in Appendix C.

**The effects of data curation.** Classical work considers dataset cleaning and outlier removal [74, 152, 124, 125] to discard samples that may lead to undesirable model bias. A related line of work develops coreset selection algorithms [61, 7, 46, 11, 94, 145, 32], which aim to select data subsets that lead to the same performance as training on the entire dataset. These techniques appear to scale poorly to larger data regimes [51, 6]. More recent efforts in subset selection often operate on already curated datasets [98, 141, 130, 16, 33, 106] (e.g., CIFAR-10, ImageNet) or on smaller data regimes (e.g., YFCC-15M [111, 140]). These settings often do not reflect newer training paradigms that involve (1) *noisy* image-text pairs instead of category labeled images and (2) large scale datasets (e.g., billions of samples). While data-centric investigations have led to community competitions like DCBENCH [43] and DATAPERF [97], existing benchmarks have likewise operated at small data scales [100] compared to datasets like LAION-2B [129], which contains over two billion images. DATACOMP bridges this gap by aligning data-centric investigation with large scale image-text training.

There has also been renewed interest in dataset pruning and deduplication. Sorscher et al. [135] show that data pruning can improve traditional scaling trends on ImageNet, but do not consider image-text training or larger datasets. Raffel et al. [113] remove sentence redundancies when creating the C4 corpus. Subsequent work further demonstrated the benefits of deduplication for better language modeling [90]. Radenovic et al. [110] introduce CAT filtering for image-text datasets—a rule-based system to retain high quality samples. Abbas et al. [6] propose SemDeDup, which starts with the CAT-filtered LAION-440M subset, further employing clustering to remove semantic duplicates. DATACOMP facilitates data-centric investigation at an even larger scale (i.e., 12.8B sample scale) and provides a common experimental setting for fair comparison amongst dataset creation algorithms.

**Large-scale multimodal datasets.** Datasets have been instrumental to building multimodal models like CLIP [111], Flamingo [8], Stable Diffusion [123], DALL-E [115, 116] and GPT-4 [103]. These methods succeeded by training on large, heterogeneous datasets rather than solely through advanced modelling techniques. For example, OpenAI's CLIP trains on 400M image-text pairs from the web, roughly $300\times$ the size of ImageNet [37]. Prior work on scaling image-text datasets also provides promising trends with respect to zero-shot model performance [73, 107]. Additional large scale datasets like FILIP-300M [149], FLD-900M [153], and PaLI-10B [25] were constructed to train multimodal models. However, many datasets used to train such models (including the dataset for OpenAI's CLIP) are proprietary, making it hard to conduct data-centric investigations.

Even for public image-text datasets like SBU [104], Flickr30k [151], MS-COCO [26], TaiSu [92], Conceptual Captions [131], CC12M [24], RedCaps [38], WIT [136], Shutterstock [101], YFCC-

Table 2: Experimental configurations, with compute in multiply-accumulate operations (MACs).

| Scale | Model | Train compute (MACs) | Pool size and # samples seen |
|---|---|---|---|
| small | ViT-B/32 | $9.5 \times 10^{16}$ | 12.8M |
| medium | ViT-B/32 | $9.5 \times 10^{17}$ | 128M |
| large | ViT-B/16 | $2.6 \times 10^{19}$ | 1.28B |
| xlarge | ViT-L/14 | $1.1 \times 10^{21}$ | 12.8B |

100M [140], COYO-700M [20], LAION-400M [128], or LAION-2B [129] little is known about what constitutes a good image-text dataset. Preliminary analysis suggests that different image-text data sources lead to CLIP models with different properties [101]. However, previous work is limited to smaller scale data (10-15M examples). Birhane et al. [15] examine LAION-400M and find NSFW imagery and racial slurs, centering the dangers in web-scale multimodal datasets. To combat toxicity, we preprocess our pool to remove NSFW content and blur human faces detected in images. For more details on our safety preprocessing see Section 3.2, Appendices E and G.

## 3 The DATACOMP benchmark

DATACOMP is meant to facilitate data-centric experimentation. While traditional benchmarks emphasize model design, DATACOMP is centered around dataset development, where the resulting datasets can be used to train high accuracy models. We focus on large image-text datasets and quantify a dataset submission by training a CLIP model on it from scratch [111] and evaluating on 38 downstream image classification and retrieval tasks. We additionally have three secret test sets, which will be released after a year, to guard against overfitting. To facilitate such investigations, we provide a candidate pool of uncurated image-text pairs sourced from the public internet. Our benchmark offers two tracks: one where participants must filter samples from the pools we provide, and another where participants can use external data. Moreover, DATACOMP is structured to accommodate participants with diverse levels of computational resources: each track is broken down into four scales with varying compute requirements. We now discuss high-level design decisions, construction of a 12.8B image-text data pool to facilitate the competition, benchmark tracks, model training, and evaluation.

### 3.1 Competition design

**Overview.** In many areas of machine learning, larger datasets lead to better performing models [87, 79, 73, 107, 66, 28, 19, 111, 112]. Hence comparing only datasets with the same size is a natural starting point. However, this approach is flawed as controlling the dataset size ignores critical curation constraints: candidate pool size (i.e., number of image-text pairs to harvest) and training compute. For instance, assembling a dataset like LAION-2B consists of identifying *data sources* (e.g., Common Crawl or Reddit) and *filtering* the data source. Notably, *the final dataset size is a design choice* and is only upper-bounded by the data sources. Hence, the true data constraint is the size of the reservoir of samples: *candidate pool* to be filtered. To make DATACOMP a realistic benchmark, we therefore fix the candidate pool in the filtering track, but give participants control over the training set size.

Compute cost is another relevant constraint. To put datasets of different size on equal footing, we specify the total *number of training samples seen*. Consider the 12.8B compute scale and filtered datasets $A$ and $B$, with 6.4B and 3.2B image-text pairs respectively. At this scale, we train by making two passes over $A$, while making four passes over $B$. A key result from our experiments is that smaller, more stringently filtered datasets can lead to models that generalize *better*.

**Competition tracks.** Two key procedures in assembling a training dataset are filtering a data source [128, 129, 20] and aggregating data sources [36, 37]. To reflect this structure, DATACOMP has two tracks: *filtering*, where participants select a subset of the samples from COMMONPOOL, and *Bring Your Own Data* (BYOD), where participants can use any source of data. Key decisions for each tracks are described in Sections 3.2 and 3.3, respectively. For full competition track rules see Appendix A.

**Competition compute scales.** To facilitate study of scaling trends and accommodate participants with various computational resources, we structure DATACOMP using four scales of compute: small, medium, large and xlarge. Each new scale increases the number of samples seen during training by

10× (from 12.8M to 12.8B samples seen), and the pool we provide by the same factor (from 12.8M samples to 12.8B samples). Table 2 gives the experimental configuration used for each scale. For the `small` scale, our runs took 4 hours on an A100 GPU, and for the `xlarge` scale 81 hours on 512 GPUs.

## 3.2 COMMONPOOL generation, for the filtering track

We construct a large-scale pool of image-text pairs, COMMONPOOL, from Common Crawl [3]. CommonPool is distributed as an image url-text pair index under a CC-BY-4.0 license. Our pool construction pipeline has four steps: url extraction and data download, NSFW detection, evaluation set deduplication, and face blurring. We additionally provide per sample metadata (e.g., CLIP features). Starting from the `xlarge` COMMONPOOL, we take successive random subsets to create `large`, `medium`, and `small` COMMONPOOL (e.g., `medium` is a subset of `large`).

**Extracting urls and dowloading data.** We first use `cc2dataset` [1], which utilizes Apache Spark [155], to extract pairs of image urls and nonempty alt-text from all Common Crawl snapshots from 2014 to 2022. We then deduplicate the url-text pairs and randomly shuffle. This step results in ∼88B possible samples. Not all samples are downloadable; other samples are not suitable due to NSFW content or overlap with our evaluation sets. We attempt to download ∼40B samples using `img2dataset` [5] resulting in ∼16.8B image-text pairs. For more details, see Appendix D.

**Safety preprocessing.** Since Common Crawl is a snapshot of the internet, we require strict preprocessing to remove unsafe content. We use Detoxify [60] to prune samples that contain unsafe text (e.g., obscene, sexually explicit, or threatening language). We also discard samples with explicit visual content. To do so, we train a classifier on CLIP ViT-L/14 [111] features, using the NSFW dataset used in LAION-5B [129]. We validate our classifier against the Google commercial image safety API. See Appendix E for details. Around 19% of image-text pairs are considered NSFW, taking the pool of ∼16.8B downloads to ∼13.6B samples.

**Evaluation set deduplication.** To prevent accidental overfitting to certain test sets in our evaluation suite, we perform a thorough near-duplicate removal between the candidate pool and our evaluation sets, using a state-of-the-art image deduplication model [150]. Appendix F contains additional details. The model flags ∼3% of the 16.8B images as near-duplicates, reducing the ∼13.6B pool to ∼13.1B samples. From here we select a random subset to get the `xlarge` pool of 12.8B samples.

**Face detection & blurring.** To protect the privacy of individuals, we detect and blur faces from images in our pool using a face detector [53]. As observed by Yang et al. [148], obfuscating faces has little impact on model performance, as we also observe in our experiments (Appendix G).

**Pool metadata.** To bootstrap participants we distribute metadata for each sample in COMMONPOOL (e.g., image url, alt-text, original image resolution, CLIP features, and CLIP similarity scores). Following Carlini et al. [22], we release SHA256 hashes for each image to guard against data poisoning in subsequent COMMONPOOL downloads. For additional details see Appendix H. We open-source our metadata processing pipeline as `dataset2metadata` [4].

## 3.3 The bring your own data (BYOD) track

While COMMONPOOL can be used to study different filtering techniques, state-of-the-art models often train on data from different sources. For instance, the Flamingo model [8] uses both multimodal massive web (M3W) and ALIGN datasets [73]. To facilitate non-proprietary research on curating data from many sources, we instantiate a separate DATACOMP track to allow participants to combine multiple data streams. For example, participants could construct a training set from CC12M [24], YFCC100M [140], and data sources they label themselves. In Section 4.2 and Appendix P.2 we describe our exploration using existing public, image-text datasets. These datasets are acquired from their respective sources and are not re-release as part of DATACOMP.

## 3.4 Training

We create a common experimental setting that enables comparable experiments by fixing the training procedure. We closely follow the CLIP training recipe proposed by Radford et al. [111]: training

models from scratch with a contrastive objective over images and captions. Given a set of image-caption pairs, we train an image encoder and a text encoder such that the similarity between the representations of images and their corresponding text is maximized relative to unaligned pairs.[1] For each scale, we fix the model architecture and hyperparameters (see Table 2). We pick Vision Transformers (ViTs) [39] as the image encoder, considering the better scaling trends observed by Radford et al. [111] compared to ResNets [62]. Models are trained for a fixed number of steps determined by the scale (Table 2), using the OpenCLIP repository [69]. See Appendix N for details.

### 3.5 Evaluation

We evaluate on a suite of 38 image classification and retrieval tasks. We also study two additional fairness tasks, detailed in Section 5 and Appendix Q. As discussed in Section 3.2, we remove test set images from DATACOMP to avoid contamination. Image classification datasets range from satellite imagery recognition to classifying metastatic tissues. In total we have (with some overlap): 22 of the datasets evaluated in Radford et al. [111], 6 ImageNet distribution shifts (i.e., ImageNet-Sketch [143], ImageNet-V2 [121], ImageNet-A [65], ImageNet-O [65], ImageNet-R [64], and ObjectNet [13]), 13 datasets from VTAB [156], and 3 datasets from WILDS [83, 127]. Retrieval datasets include Flickr30k [151], MSCOCO [26], and the WinoGAViL commonsense association task [17]. To aggregate results over all evaluation tasks, we average the preferred metric for each task.

DATACOMP adopts a zero-shot evaluation protocol: models are tested without training on the evaluation tasks. This approach is computationally efficient and measures a model's ability to perform well without any additional training. We find a strong rank correlation ($>0.99$) between performance in linear probe zero-shot settings (Appendix Figure 16). Additional details are in Appendix O.

## 4 Baselines

### 4.1 Filtering baselines

We study six simple filtering methods for the filtering track; see Appendix P.1 for further details.

**No filtering.** We simply use the entire pool as the subset, without any filtering. Since each pool size is equal to the sample budget, training consists of one pass over the data.

**Random subsets.** To isolate the effects of increasing the compute budget from increasing the dataset size, we form subsets consisting of 1%, 10%, 25%, 50% and 75% of the pool chosen at random.

**Basic filtering.** We consider many simple filtering operations inspired by Schuhmann et al. [128] and Byeon et al. [20]: filtering by *language* (English captions, using either fasttext [77] or cld3 [2]); filtering by *caption length* (over two words and five characters); and filtering by *image size* (smaller dimension above 200 pixels and aspect ratio below three). We also experiment with combining language and caption length filtering and combining language, caption length, image size fitering. Unless otherwise specified, "basic" refers fasttext English, caption length, and image size filtering.

**CLIP score and LAION filtering.** We experiment with CLIP score filtering (also employed by LAION), where we take only examples having cosine similarity scores between CLIP image and text embeddings that exceed a pre-defined threshold. We investigate a range of thresholds and two OpenAI CLIP models for computing the scores: the ViT-B/32 model (as in LAION) and the larger ViT-L/14. We also combine CLIP score thresholds and cld3 English filtering to reproduce the LAION-2B filtering scheme. Table 16 in Appendix P.1 summarizes the different CLIP score configurations.

**Text-based filtering.** We select examples that contain text overlapping with ImageNet class names, which serve as a proxy for relevance to downstream tasks. Specifically, we select English captions (according to fasttext) that contain words from ImageNet-21K or ImageNet-1K [37] class synsets.

---

[1] More precisely, given a batch of data $\{(x_1, y_1), ..., (x_B, y_B)\}$ with images $x$ and captions $y$, we train the image encoder $g$ and text encoder $v$ with the loss $\ell = \frac{1}{2} \sum_{i=1}^{B} \frac{\sigma_{ii}}{\sum_{j=1}^{B} \sigma_{ij}} + \frac{1}{2} \sum_{i=1}^{B} \frac{\sigma_{ii}}{\sum_{j=1}^{B} \sigma_{ji}}$, where $\sigma_{ij} = \exp \langle g(x_i), h(y_j) \rangle$. We also use a learnable temperature parameter as in Radford et al. [111].

Table 3: Zero-shot performance for select baselines in the *filtering* track. On all scales, filtering strategies lead to better performance than using the entire, unfiltered pool. The intersection between imaged-based and CLIP score strategies performs well on most tasks and scales. For all metrics, higher is better (see Appendix O for details). ∩ denotes the intersection of filtering strategies.

| Scale | Filtering strategy | Dataset size | Samples seen | ImageNet | ImageNet dist. shifts | VTAB | Retrieval | Average over 38 datasets |
|---|---|---|---|---|---|---|---|---|
| small | No filtering | 12.8M | 12.8M | 0.025 | 0.033 | 0.145 | 0.114 | 0.132 |
| | Basic filtering | 3M | 12.8M | 0.038 | 0.043 | 0.150 | 0.118 | 0.142 |
| | Text-based | 3.2M | 12.8M | 0.046 | 0.052 | 0.169 | 0.125 | 0.157 |
| | Image-based | 3M | 12.8M | 0.043 | 0.047 | 0.178 | 0.121 | 0.159 |
| | LAION-2B filtering | 1.3M | 12.8M | 0.031 | 0.040 | 0.136 | 0.092 | 0.133 |
| | CLIP score (L/14 30%) | 3.8M | 12.8M | 0.051 | 0.055 | 0.190 | 0.119 | 0.173 |
| | Image-based ∩ CLIP score (L/14 30%) | 1.4M | 12.8M | 0.039 | 0.045 | 0.162 | 0.094 | 0.144 |
| medium | No filtering | 128M | 128M | 0.176 | 0.152 | 0.259 | 0.219 | 0.258 |
| | Basic filtering | 30M | 128M | 0.226 | 0.193 | 0.284 | 0.251 | 0.285 |
| | Text-based | 31M | 128M | 0.255 | 0.215 | 0.328 | 0.249 | 0.307 |
| | Image-based | 29M | 128M | 0.268 | 0.213 | 0.319 | 0.256 | 0.312 |
| | LAION-2B filtering | 13M | 128M | 0.230 | 0.198 | 0.307 | 0.233 | 0.292 |
| | CLIP score (L/14 30%) | 38M | 128M | 0.273 | 0.230 | 0.338 | 0.251 | 0.328 |
| | Image-based ∩ CLIP score (L/14 30%) | 14M | 128M | 0.297 | 0.239 | 0.346 | 0.231 | 0.328 |
| large | No filtering | 1.28B | 1.28B | 0.459 | 0.378 | 0.426 | 0.419 | 0.437 |
| | Basic filtering | 298M | 1.28B | 0.516 | 0.423 | 0.446 | 0.480 | 0.458 |
| | Text-based | 317M | 1.28B | 0.561 | 0.465 | 0.465 | 0.352 | 0.466 |
| | Image-based | 293M | 1.28B | 0.572 | 0.454 | 0.483 | 0.479 | 0.476 |
| | LAION-2B filtering | 130M | 1.28B | 0.553 | 0.453 | 0.510 | 0.495 | 0.501 |
| | CLIP score (L/14 30%) | 384M | 1.28B | 0.578 | 0.474 | 0.538 | 0.466 | 0.529 |
| | Image-based ∩ CLIP score (L/14 30%) | 140M | 1.28B | 0.631 | 0.508 | 0.546 | 0.498 | 0.537 |
| xlarge | No filtering | 12.8B | 12.8B | 0.723 | 0.612 | 0.611 | 0.569 | 0.621 |
| | LAION-2B filtering | 1.3B | 12.8B | 0.755 | 0.637 | 0.624 | 0.620 | 0.636 |
| | CLIP score (L/14 30%) | 3.8B | 12.8B | 0.764 | 0.655 | 0.643 | 0.588 | 0.650 |
| | Image-based ∩ CLIP score (L/14 30%) | 1.4B | 12.8B | 0.792 | 0.679 | 0.652 | 0.608 | 0.663 |

**Image-based filtering.** We select a subset of examples whose visual content overlaps with ImageNet classes. After applying English language (fasttext) and caption length filtering, we cluster the image embeddings extracted by the OpenAI ViT-L/14 model for each image into 100K groups using Faiss [75]. We then find the nearest neighbor group for every ImageNet training example, and keep examples belonging to these groups. We apply this procedure using either ImageNet-21K (14M images) or ImageNet-1K (1.2M images), forming two subsets.

### 4.2 BYOD baselines

We experiment with multiple external data sources, including four moderately sized datasets (10 to 58M samples) studied by Nguyen et al. [101]—CC12M [24], YFCC15M [140, 111], RedCaps [38] and Shutterstock [101]—and the larger LAION-2B [129]. Additional experiments, along with more details about the data sources are provided in Appendix P.2. We consider these data sources as they are and do not perform additional preprocessing. We also present experiments combining some of the data sources (using only the external datasets, or in addition to data from our pool).

## 5 Results and discussion

### 5.1 Building better datasets

**Main results.** Our key results are in Table 3. Most notably, the intersection between image-based filtering and CLIP score filtering excels on most tasks. The exception is at the small scale and for retrieval datasets.[2] Furthermore, other filtering strategies like basic, CLIP score, image-based, text-based filtering show better downstream performance when compared to no filtering. A much larger suite of experiment results can be found in Appendix R.

**DATACOMP leads to better image-text datasets.** We hope DATACOMP catalyzes the search for the next generation of multimodal datasets. We contribute DATACOMP-1B, which is the output of the Image-based ∩ CLIP score (L/14 30%) baseline filter at the xlarge scale of the filtering track.

---

[2]Cherti et al. [28] also observe that models rank differently on classification and retrieval tasks.

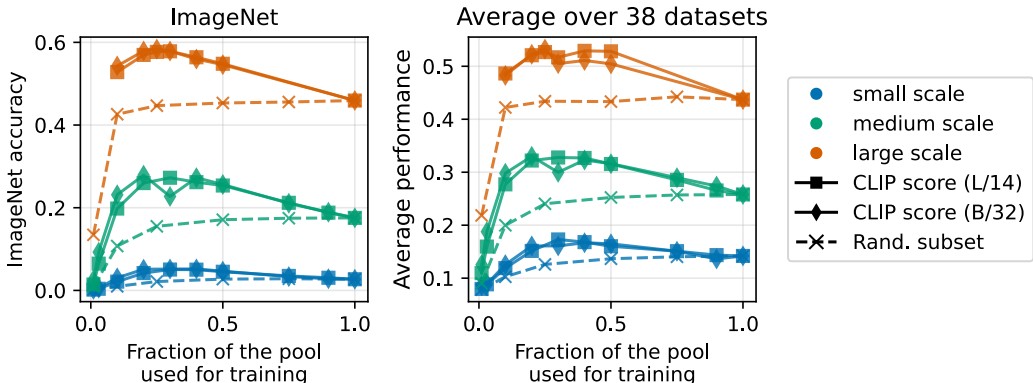

Figure 2: Performance of random subsets (dotted line) and CLIP score filtering (solid line) when varying the subset size. When taking random subsets, larger subsets are always better. For CLIP score filtering, subsets with intermediate size perform best.

Our dataset is comprised of 1.4B samples, which not only is *smaller* than the LAION-2B dataset with 2.3B samples, but also comes from a smaller pool. Nevertheless, a CLIP L/14 trained on DATACOMP-1B outperforms the LAION-2B competitor by 6.1 percentage points on ImageNet (see Table 1). Moreover, training on DATACOMP-1B improves ImageNet accuracy by 3.7 percentage points over OpenAI's ViT-L/14 trained with the same compute budget. Additionally, even if we restrict ourselves to 400M samples, we can still find a subset of DATACOMP-1B that outperforms OpenAI's ViT-L/14, as seen in Table 24. These results demonstrate the impact that DATACOMP can make and provide a foundation upon which participants can build.

**External data sources can improve performance.** Appendix P.2 Table 18 shows results for several baselines in the BYOD track. We find several instances where adding external data sources improves performance over using just data from COMMONPOOL. For example, at the `large` scale, combining CLIP-filtered data from COMMONPOOL with external data from CC12M [24], YFCC15M [140, 111], RedCaps [38] and Shutterstock [101] boosts ImageNet accuracy by 4.3 percentage points. See Appendix P.2 for more experiments and details.

**Trade-off between data diversity and repetition.** In Figure 2, we see that randomly selecting subsets of the pool has little effect and degrades performance substantially when only small fractions are used. When filtering with CLIP scores, the optimal training set comes from selecting ∼30% of the pool with the highest scores. The difference in performance trends between random subsets and CLIP score filtering highlights the importance of filtering strategies for selecting samples.

## 5.2 DATACOMP design analyses

**COMMONPOOL and LAION are comparable with the same filtering.** To validate our pool construction, we show that we can build datasets comparable to LAION-2B by employing their filtering technique on our pool. LAION-2B selects all samples where the caption is in English and the cosine similarity score from a trained ViT-B/32 CLIP model is above 0.28. We compare this filtering approach on our pool using the same number samples, 130M samples at the `large` scale. We find that the different data sources perform comparably: 55.3% vs 55.7% accuracy on ImageNet, and 0.501 vs 0.489 average performance over our evaluation sets using our pool and LAION-2B, respectively.

**Consistency across scales.** We find that the ranking between filtering strategies is typically consistent across different scales. This is illustrated in Figure 3, which shows that the baselines at `small` and `medium` scales are positively correlated. Moreover, as shown in Appendix Table 22, the rank correlations of performance is high, between 0.71 and 0.90 for different scale pairs.

**Consistency across training changes.** DATACOMP fixes the training procedure, so a natural question is whether better datasets from DATACOMP are better outside of DATACOMP. While DATACOMP-1B is trained at the `xlarge` scale, we show in Appendix Table 23 that even when substituting the ViT-L/14

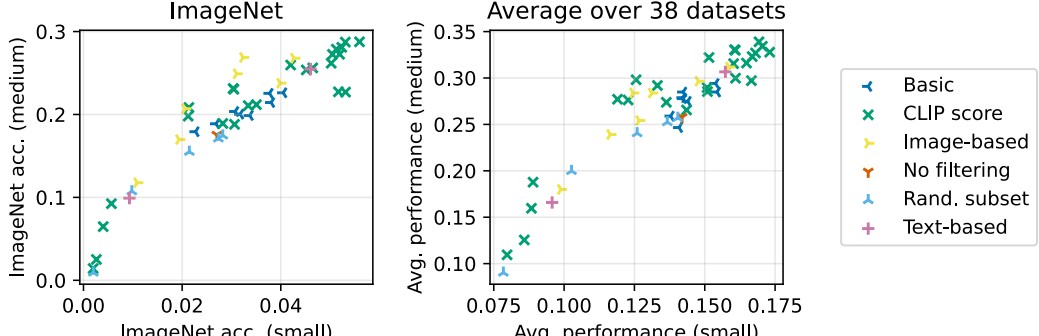

Figure 3: Correlation between `small` and `medium` scale baselines. Smaller scales can serve as useful guides for larger scales. Results for additional scales are shown in Appendix Figure 22.

for a ViT-B/16 or ViT-B/32, training on DATACOMP-1B outperforms training on OpenAI's WIT and LAION-2B. Additionally, we found that modifying hyperparameters such as training steps and batch size minimally affects the relative ordering of different data curation methods on downstream performance. Details on hyperparameter ablations are in Appendix L.

### 5.3 Evaluation trends

**ImageNet accuracy is indicative, but not the complete picture.** Similarly to Kornblith et al. [84], in Appendix Figure 25 we find that ImageNet performance is highly correlated with the average performance across all datasets we study, with an overall correlation of 0.99. [3] However, ImageNet performance is not representative of all evaluation tasks, as the correlation between ImageNet accuracy and accuracy on other individual datasets varies substantially, in some cases even exhibiting a negative correlation, as discussed in Appendix R.

**Robustness and fairness.** While typical models trained on a target task suffer large performance drops under data distribution shift, zero-shot CLIP models are known to exhibit strong performance across many distributions [111]. In Appendix Figure 26, we show that CLIP models trained with data from our pool are more robust to distribution shift than ImageNet-trained models from Taori et al. [139]'s testbed. Examining geographic diversity, we find that our models are better than ImageNet-trained models, but fall short of models fine-tuned on diverse curated datasets (see Appendix Figure 21). We also perform a face classification analysis and identify demographic biases in our models: notably, the BYOD datasets we consider can increase the risk of misclassification. See Appendix Q for more fairness and diversity analyses.

## 6 Limitations and conclusion

In terms of societal risks, creating an index of image-text pairs from the public internet can be problematic. The internet contains unsafe, toxic, and sensitive content, which ideally should not percolate into machine learning datasets. Though we take steps to remove NSFW content and blur human faces to protect privacy, we hope future work will further explore the biases and risks from COMMONPOOL and DATACOMP-1B. We see several additional directions for future work, including 1) Curating more data sources. 2) Improved data filtering algorithms. 3) Further supervision signals (e.g., image captions coming from captioning models). 4) Additional input modalities (e.g., video, 3D objects). 5) Broader evaluations for vision-and-language and robotics tasks.

Overall, we see DATACOMP as a first step towards improving training datasets, and hope our new benchmark will foster further research. By providing a controlled experimental setting, DATACOMP enables researchers to iterate on dataset design on rigorous empirical foundations. We open-source all of our code, data, and infrastructure, and hope these resources will help the community build the next generation of multimodal datasets.

---

[3]Note that unlike Kornblith et al. [84] we evaluate zero-shot performance rather than transfer learning.

## Acknowledgements

SYG and JH are supported by NSF Graduate Research Fellowships. GS is supported by the Onassis Foundation - Scholarship ID: F ZS 056-1/2022-2023. GD has been supported by the Onassis Fellowship (Scholarship ID: F ZS 012-1/2022-2023), the Bodossaki Fellowship and the Leventis Fellowship. This research has been supported by NSF Grants AF 1901292, CNS 2148141, DMS 2134012, TRIPODS II-DMS 2023166, Tripods CCF 1934932, IFML CCF 2019844 and research gifts by Western Digital, WNCG IAP, UT Austin Machine Learning Lab (MLL), Cisco, the Len Blavatnik and the Blavatnik Family Foundation, the Stanly P. Finch Centennial Professorship in Engineering, Open Philanthropy, Google, Microsoft, and the Allen Institute for AI.

We would like to thank Amro Abbas, Danny Bickson, Alper Canberk, Jessie Chapman, Brian Cheung, Tim Dettmers, Joshua Gardner, Nancy Garland, Sachin Goyal, Huy Ha, Zaid Harchaoui, Ari Holtzman, Andrew Hundt, Andy Jones, Adam Klivans, Ronak Mehta, Sachit Menon, Ari Morcos, Raviteja Mullapudi, Jonathon Shlens, Brandon McKinzie, Alexander Toshev, David Grangier, Navdeep Jaitly, Kentrell Owens, Marco Tulio Ribeiro, Shiori Sagawa, Christoph Schuhmann, Matthew Wallingford, and Ross Wightman for helpful feedback at various stages of the project. We are particularly grateful to Daniel Levy and Alec Radford for early encouragement to pursue this project and feedback on the experimental design.

We thank Stability AI and the Gauss Centre for Supercomputing e.V.[4] for providing us with compute resources to train models. We are thankful for the compute time provided through the John von Neumann Institute for Computing (NIC) on the GCS Supercomputer JUWELS Booster [78] at Jülich Supercomputing Centre (JSC), and for storage resources on JUST [50] granted and operated by JSC, as well as computing and storage resources from the Helmholtz Data Federation (HDF).

---

[4] https://gauss-centre.eu

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
