# Appendix

## Contents

# A  Benchmark rules

We provide concrete rules below for the two competition tracks that comprise DATACOMP: filtering and BYOD. Additionally, we provide a checklist, which encourages participants to specify design decisions, which allows for more granular comparison between submissions.

## A.1  Filtering track rules

- Participants can enter submissions for one or many different scales: `small`, `medium`, `large` or `xlarge`, which represent the raw number of image-text pairs in CommonPool that should be filtered.
- After choosing a scale, participants generate a list of uids, where each uid refers to a COMMONPOOL sample. The list of uids is used to recover image-text pairs from the pool, which is used for downstream CLIP training.
- Duplicate uids are allowed.
- Participants are *not* allowed to modify the training procedure.  Hence, changing hyperparameters, model architecture, optimizer, compute budget, or number of training steps is not allowed. Changing any other training details is also not allowed.
- Participants are strongly encouraged to submit and open-source both the list of uids and the code used to generate this list; however, this is not required.
- To avoid overfitting, we do not permit running any code or algorithmic dependence on the test images of the evaluation tasks. However, use of other images associated with these tasks (e.g., supervised training sets) is permitted.
- Participants can use templates or class labels from the downstream tasks in their filtering algorithms.

For clarity, we include some examples of permitted and forbidden uses:

- ✓ We **permit** using the ImageNet class label "triceratops" in a filtering algorithm.
- × We **forbid** examining individual or aggregate predictions on the test sets of the evaluation tasks.

## A.2  Bring your own data track: amendments

To facilitate more open-ended exploration, we provide amendments to the Track 1 competition to allow for more diverse submissions in Track 2.

- Participants are allowed to augment COMMONPOOL data with existing datasets, so long as these data sources do not contain test images from the evaluation tasks. Participants can use data from any COMMONPOOL; however, they are not required to do so.
- Assembling one's own dataset is allowed; however, test images from the evaluation tasks can neither be contained nor otherwise used to construct said dataset. We encourage releasing the image urls or the images themselves in addition to the text for each image. We also encourage rigorous documentation of face-blurring and other data safety checks (see Section 3.2 for more details). We reserve the right to run our own safety code on participant provided data and disqualify entries that do not meet adequate safety standards.

**Checklist.**  The following checklist provides the basis for more fine-grained comparison between submissions.

- ☐ Images from the evaluation tasks are included in my submission.  If yes, please specify which datasets.
- ☐ I used an existing datasets (e.g., YFCC100M [140]) in my submission. If yes, please specify which datasets. (Note: applies to BYOD only)
- ☐ I curated my own data. If yes, please provide (1) image data or urls, (2) text for each image, (3) list of safety steps taken including but not limited to face blurring, explicit content image and text filtering. (Note: applies to BYOD only)

# B   Contributions

For this section, contributors are ordered alphabetically.

## B.1   Candidate pool

**Candidate pool lead.**  Vaishaal Shankar

**Data collection.**  Romain Beaumont, Vaishaal Shankar

**Pre-processing and metadata.**  Giannis Daras, Alex Fang (content filtering lead), Samir Yitzhak Gadre (metadata lead), Ryan Marten (deduplication lead), Vivek Ramanujan, Vaishaal Shankar, George Smyrnis (face blurring lead)

## B.2   Participant tooling

**Participant tooling lead.**  Gabriel Ilharco

**Resharder.**  Romain Beaumont, Yair Carmon, Alex Fang, Jonathan Hayase (lead), Gabriel Ilharco, Vivek Ramanujan, Vaishaal Shankar, Georgios Smyrnis

**Training.**  Mehdi Cherti, Gabriel Ilharco, Jenia Jitsev, Vivek Ramanujan, Georgios Smyrnis, Mitchell Wortsman (lead)

**Evaluation.**  Romain Beaumont, Yonatan Bitton, Mehdi Cherti, Dhruba Ghosh (lead), Gabriel Ilharco

**Additional infrastructure.**  Stephen Mussmann, Sarah Pratt

## B.3   Baselines

**Baselines lead.**  Yair Carmon

**Filtering track.**  Yair Carmon, Rahim Enterazi, Alex Fang, Samir Yitzhak Gadre, Gabriel Ilharco, Kalyani Marathe, Thao Nguyen, Eyal Orgad (co-lead), Georgios Smyrnis, Mitchell Wortsman, Jieyu Zhang (co-lead)

**BYOD track.**  Gabriel Ilharco, Thao Nguyen

**Experiment babysitting.**  Alex Fang, Gabriel Ilharco, Samir Yitzhak Gadre

## B.4   Leadership and Advising

**Advising.**  Romain Beaumont, Yair Carmon, Alexandros G. Dimakis, Ali Farhadi, Hannaneh Hajishirzi, Jenia Jitsev, Pang Wei Koh, Ranjay Krishna, Stephen Mussmann, Sewoong Oh, Alexander Ratner, Olga Saukh, Ludwig Schmidt, Vaishaal Shankar, Shuran Song, Richard Vencu

**Leadership.**  Yair Carmon, Alexandros G. Dimakis, Jenia Jitsev, Sewoong Oh, Ludwig Schmidt, Vaishaal Shankar

**Overall project lead.**  Ludwig Schmidt

# C   Additional related work

Here we expand on the related work described in Section 2.

Image dataset safety is an active area of research, especially in the context of large-scale dataset construction. In addition to Birhane et al. [15], who study problematic content in LAION-400M, Yang et al. [147] study the ImageNet dataset and reveal limitations associated with the ImageNet curation strategy—with negative implications for downstream model fairness. Prabhu & Birhane [108] also study the ImageNet dataset and find pornographic content. Both Birhane et al. [15] and Prabhu & Birhane [108] survey ethical conundrums and harms that are borne out of improper dataset curation. In an effort to combat dataset toxicity, we conduct NSFW preprocessing (Section 3.2, Appendix E) and blur detected faces (Section 3.2, Appendix G) during pool construction. We also conduct preliminary fairness evaluations (Section 5.3, Appendix Q) for models trained on our data. We hope COMMONPOOL will serve as a research artifact for future work examining dataset safety.

Beyond data selection, Chan et al. [23] investigate the effects of dataset distribution on emergent properties of transformers, while Fang et al. [44] look at the relationship between data and model robustness to distribution shifts. We hope our extensive evaluation suite comprised of 38 diverse tasks will facilitate similar studies when training multimodal models at large scale.

Others study how to reduce the burdens of training data annotation in the curation process. Classic approaches include distant supervision [67], crowd-sourced labels [154], heuristic rules [9] and feature annotation [96], among others. A recent line of work known as data programming or programmatic weak supervision [118, 119, 157, 158] attempts to reduce annotation cost and is found in many industry applications [10, 120]. In data programming, developers write programmatic labeling functions to automatically label a large amount of unlabeled data. The labeling functions could produce noisy and conflicting labels, so researchers have developed methods to aggregate noisy votes to produce the final training labels [117, 47, 133].

Previous literature also studies methods for training data attribution, which seek to link a model's behavior (e.g., its accuracy on a particular task or subset of data) to particular subsets of its training data. Such methods include influence functions, a classic technique from robust statistics [57, 35] that uses a second-order Taylor expansion to approximate the effect of removing a training point on the learned model parameters [81, 82, 58, 52], as well as methods that fit attribution functions directly to the dynamics of repeated training runs [49, 109, 71, 56]. Training data attribution methods assume that we have already trained a model, though they can be subsequently used to refine the training data (e.g., by identifying potentially mislabeled training points [81]). Our focus in this paper is instead on data curation methods—that is, methods for selecting a subset of the training data to train a model in the first place.

In the context of natural language processing, Swayamdipta et al. [138] proposes a tool for characterizing samples in a dataset based on training dynamics, labelling instances as ambiguous, easy to learn or hard to learn. Previous literature such as work by Le Bras et al. [88], Li & Vasconcelos [91], Gururangan et al. [55] advocate for removing easy instances from the training data. Ethayarajh et al. [41] propose a measure of how difficult a dataset is to learn, $\mathcal{V}$-usable information. Such techniques could be promising directions of further exploration in the context of our benchmark.

Finally, another related line of work is studying scaling trends. In addition to Sorscher et al. [135], researchers have investigated how model performance changes as a function of compute budget, model size, and number of training samples [79, 66, 21, 28]. However, this line of work does not consider how dataset design may affects scaling trends. Beyond dataset size, we measure the effects of different dataset sources and filtering strategies. While scaling trends are central to our investigations, the purpose of our benchmark is to search for the next generation of large multimodal datasets to facilitate more accurate and reliable models.

# D   Parsing Common Crawl

Common Crawl releases metadata files for the websites that they index (i.e., WAT files). They release these files approximately once a month. We consider all files available from 2014 through November of 2022. We first parse these files, utilizing Apache Spark [155] to extract image urls and corresponding alt-text. We map each url, text pair to a uid hash and remove duplicates. This

Table 4: Detoxify positive rates by threshold on 1 million caption subset of Common Crawl.

| Threshold | Toxicity | Severe Toxicity | Obscene | Identity Attack | Insult | Threat | Sexual Explicit |
|-----------|----------|-----------------|---------|-----------------|--------|--------|-----------------|
| 0.01 | 9.5% | 1.0% | 33.4% | 1.8% | 35.0% | 1.3% | 2.0% |
| 0.1 | 3.6% | 0.1% | 0.8% | 0.3% | 1.4% | 0.1% | 1.0% |

Table 5: Comparing LAION-2B CLIP based NSFW filtering model to Google Vision API Safe Search adult category on a 40,000 random subset of Common Crawl.

| Threshold | False Positive Rate (Relative to Google) | True Positives (Manual Review) | Model Positive Rate | Google API Positive Rate |
|-----------|------------------------------------------|--------------------------------|---------------------|--------------------------|
| 0.1 | 3.6% | 2 | 14.4% | 3.5% |
| 0.2 | 0.6% | 2 | 9.1% | 3.5% |
| 0.3 | 0.3% | 3 | 7.2% | 3.5% |

results in 88 billion url, text pairs, which are randomized via a distributed shuffle. Note, we do not consider image content when running uid deduplication at this step. Hence, two identical images with different urls and the same caption would both be retained.

# E    Not safe for work (NSFW) filtering

Our data is sourced from Common Crawl, which contains snapshots of the web. Therefore, we apply multiple layers of NSFW content filtering to remove problematic images and captions from COMMONPOOL.

First, we filter our captions with Detoxify [60], a language model for toxic comment classification. Specifically, we use the multilingual XLM-RoBERTa [34] variant. The model outputs scores between zero and one for the following categories: toxicity, severe toxicity, obscene, identity attack, insult, threat, and sexually explicit. As we had no ground truth for our data, we manually spot check a 1 million random subset of COMMONPOOL at varying thresholds. We found that a threshold of 0.1 provided good coverage of filtering out NSFW text. If any of the detoxify category scores exceeds the threshold, the sample is discarded. Qualitatively, we found that the model struggled with multilingual content, acronyms, and innuendo. Even at 0.1, we noticed there are some captions that are NSFW. However, lowering the threshold further heavily affected false positives. We therefore use a 0.1 threshold for all NSFW categories, which on a random subset of one million captions achieves positive rates shown in Table 4.

Second, on the vision side, we use a modified version of LAION-5B's [129] CLIP-based binary classification NSFW model, which takes CLIP ViT-L/14 visual embeddings as input. We remove the initial multi-category encoder from the model, and retrain on the same data with an initial normalization layer followed by a 4-layer multilayer perceptron. Our retrained model matches the performance of the original model on their manually annotated testset. Specifically, we achieve 97.4% classification accuracy on a held out test set compared to 96.1% for the original LAION NSFW image filtering model. Additional details about the training data can be found in Appendix C.5 of the LAION-5B paper. In brief, the training data contains 682K images that is roughly balanced with images from safe for work and NSFW categories.

To evaluate our model and determine a threshold, we used Google Vision API's SafeSearch explicit content detector to generate labels for an 40,000 random subset of our candidate pool. Specifically, an image is NSFW if SafeSearch classifies it as likely or very likely adult (i.e., sexually explicit). As shown in Table 5, we found that by thresholding at 0.1 we achieve high recall relative to SafeSearch and very few true positives after manual review. We also manually reviewed images classified by SafeSearch as likely or very likely racy and found that the images were either benign, subjectively suggestive but not explicit, or already found in the set of images labeled as adult.

# F    Deduplication against evaluation sets

To prevent data leakage, we filter COMMONPOOL by removing duplicate and near-duplicate matches of evaluation set images. See Figure 4 for example query images from Common Crawl and corresponding near-duplicates in our evaluations sets. We consider images as duplicates when

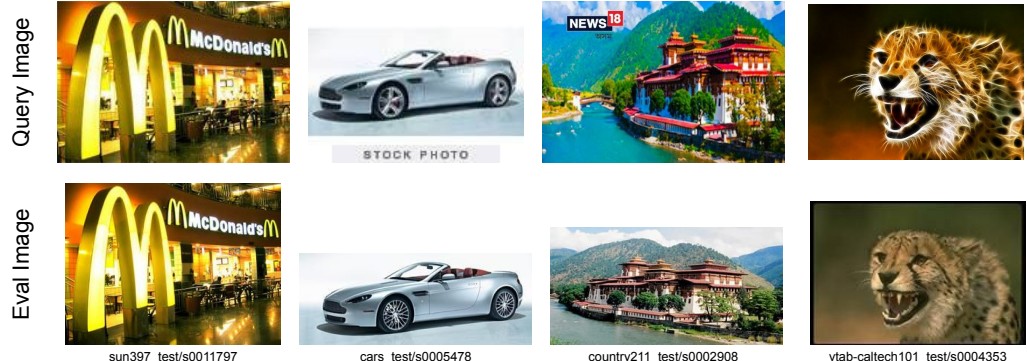

sun397_test/s0011797    cars_test/s0005478    country211_test/s0002908    vtab-caltech101_test/s0004353

Figure 4: Candidate images (top) that are detected as duplicates against images in the evaluation sets (bottom) are removed from the pool. In addition to exact duplicate images, near-duplicates with variable aspect ratios, JPEG compression, overlays, color adjustment, and artistic rendering are also detected.

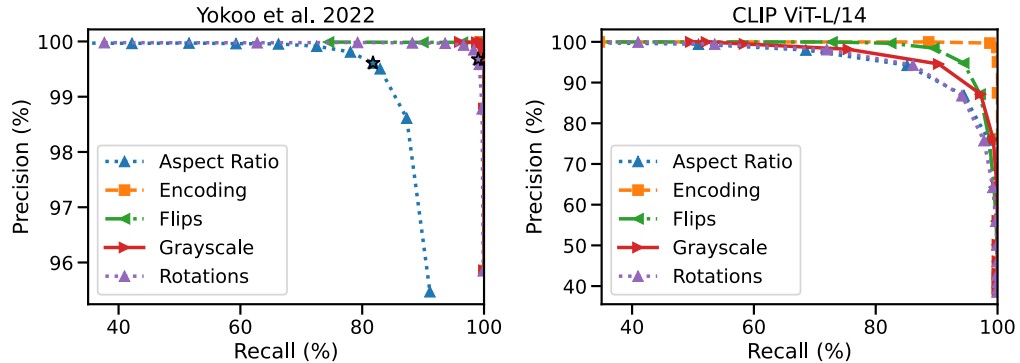

Figure 5: Analysis of different de-duplication strategies across a variety of image transformations. We see that the model introduced by Yokoo [150] is better in almost every transformation, with the exception of very aggressive aspect ratio modification.

the cosine similarity between a query (Common Crawl image) feature and a reference (evaluation image) feature is higher than a fixed threshold. We employ the deduplication model proposed by Yokoo [150], which earned 1st place in the Facebook AI Image Similarity Challenge (ISC) [40]. We choose a cosine similarity threshold of 0.604169 to maximize the true duplicates detected, without removing too many false duplicates from the pool. We compare against OpenAI's CLIP ViT-B/32 as a baseline on ISC. We find that for our threshold, the ISC model achieves precision 0.9 and recall 0.8. At a threshold of 0.96, CLIP achieves the same precision 0.9, but a significantly worse recall of 0.02. Approximately 2.8% of downloaded samples are flagged as evaluation set near-duplicates.

To verify the performance of our de-duplication models with greater granularity, we modify the evaluation procedure in Douze et al. [40] to include transformations which are representative of naturally-occurring duplications on the Internet. Specifically, we study: 1) jpeg compression (encoding), 2) image flips, 3) image rotations, 4) aspect ratio modifications, and 5) grayscaling. To do this, we sample 20% of the images from each of our evaluation datasets uniformly at random to serve as a reference set of about 140,000 images. Next we sample 560,000 images uniformly at random from LAION-2B to serve as distractors, for a 4-to-1 distractor to reference ratio. Finally, we apply each of the augmentations above and use threshold filtering to determine duplicates. Figure 5 shows the results from the deduplication model [150] compared with OpenAI's CLIP ViT-L/14. At high recall values, we see that CLIP filtering results in removing over 2× the data as that of the deduplication model from Yokoo [150].

Table 6: Face detection performance on a set of 3293 random images from COMMONPOOL.

|           | SCRFD-10G | Amazon Rekognition |
|-----------|-----------|--------------------|
| Accuracy  | 93.87     | 96.57              |
| Precision | 75.87     | 86.09              |
| Recall    | 90.53     | 93.75              |

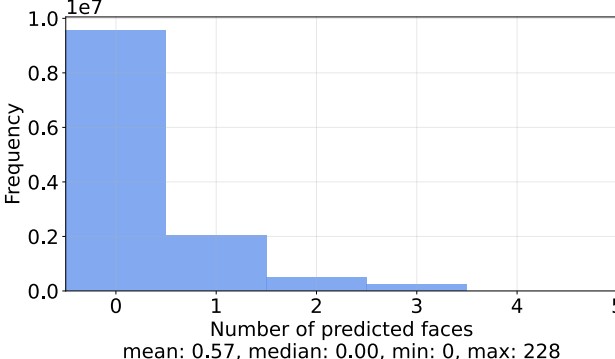

Figure 6: Frequency of predicted number of faces in the `small` COMMONPOOL.

# G   Face blurring

As an extra step to safeguard against issues of privacy that may arise from the use of data scraped from the web, we include face blurring as part of our pool creation. To create face metadata, we use the SCRFD face detector [53] to extract bounding boxes for the faces in our images. These bounding boxes are included as part of the image metadata in our pool. We make use of the pretrained SCRFD-10G model. We use the same preprocessing as the one described in the official repository of the paper, with the exception of providing $224 \times 224$ input images (by padding each image to square and then resizing) to limit computation costs. Invoking this model provides us with bounding boxes along with an associated score, which we then compare against a threshold of $0.3$ to keep or discard this bounding box. This threshold is the default one used in the repository of SCRFD for the visualization of bounding boxes, and we found it to perform well on our data as discussed next.

In Table 6 we can see the result of face detection on a set of 3293 images from COMMONPOOL. We evaluate the detection on whether the image has visible faces or not (where images such as cartoon drawings of non-real human faces are not considered as positives), and whether the detector has detected these visible faces. We considered an image as a true positive if all the clearly visible faces in the image were detected, based on the above thresholding process. We did not do extensive box labeling. True positives are instead determined by human inspection. We compare the quality of these detections with the Amazon Rekognition system, which is the one upon which the face detections on ImageNet were based [148]. Note that in this scenario, the recall of the detectors is more important than precision (as detecting a few more bounding boxes across our pool does not affect privacy).

To utilize these bounding boxes on our data, we apply a standard blurring pipeline, as proposed by Yang et al. [148]. The result of this process is an image where the faces is blurred and there is a smooth transition from blurred to clean parts of the image. In Figure 6 we see the distribution of faces for the `small` COMMONPOOL. Note that the majority of images do not contain faces.

As part of our competition pipeline, images are by default blurred during the download process. In Table 7 we can see the results of training on a set of images with the size of our `medium` scale after filtering with each method, with and without the application of face blurring as provided by our detector. We can see that the difference in performance is small, which suggests that the application of face blurring does not significantly affect the performance on our downstream tasks. However, we note that this design decision may be more detrimental in generative settings, especially when a generative model needs to output faces. Our competition is primarily focused on discriminative tasks, and as such when designing our dataset, we wished to prioritize the safety and privacy of individuals through blurring faces in our download tooling by default.

Table 7: Effect of face blurring on zero-shot performance. Face blurring improves the privacy preservation of our dataset, while affecting model performance negligibly. Results shown for training on a set of images with the size of our `medium` scale, after filtering with each method.

| Filtering | Face blurring | ImageNet acc. | Avg. performance |
|---|---|---|---|
| CLIP score (B/32, thresh. 0.3) + English filtering | × | 0.209 | 0.246 |
| | ✓ | 0.196 | 0.243 |
| CLIP score (B/32, 30%) | × | 0.287 | 0.301 |
| | ✓ | 0.282 | 0.298 |

Finally, we evaluated the detector we used for potential biases. More specifically, we used the detector on the validation set of the FairFace dataset [80]. We found that the central face of the image was detected in all the images of the validation set, regardless of subgroup annotate in the dataset.

## H    DATACOMP COMMONPOOL creation pipeline

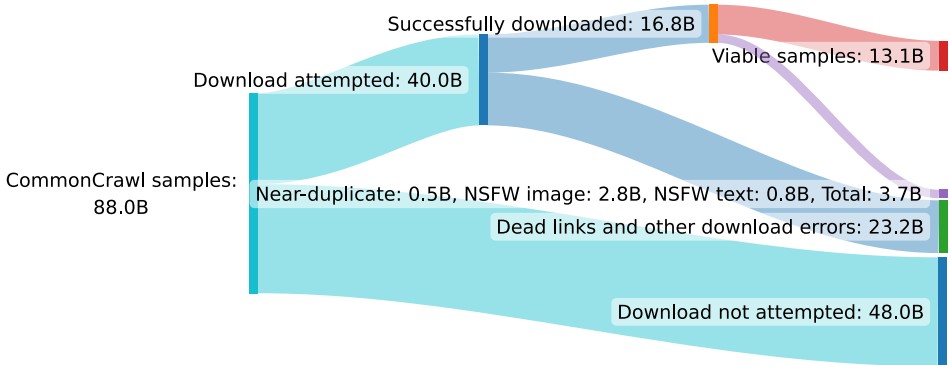

Figure 7: Data funnel from potential samples in Common Crawl to 13.1B image-text pairs that were suitable for COMMONPOOL. We sampled uniformly 12.8B datapoints for the `xlarge` COMMONPOOL.

Table 8: Provided metadata for COMMONPOOL.

| Generation Time | Label | Additional notes |
| --- | --- | --- |
| | uid | |
| | url | Link to the image. |
| Step 2 | text | Image caption. |
| | original_width | |
| | original_height | |
| | sha256 | Safeguard for data poisoning. |
| | clip_b32_similarity_score | |
| | clip_b32_image_features | In separate file. |
| | clip_b32_text_features | In separate file. |
| Step 1 | clip_l14_similarity_score | |
| | clip_l14_image_features | In separate file. |
| | clip_l14_text_features | In separate file. |
| | face_bboxes | |
| | nsfw_image_score | |
| Step 2, dropped during Step 3 | nsfw_text_score | |
| | dedup_score | |

Creating COMMONPOOL was a multistep process, which involved (1) parsing image urls and alt-text from Common Crawl dumps and downloading these images, (2) tagging images with metadata and (3) conducting safety content filtering and evaluation set duplication. In this section we provide an overview of the data pipeline used to create COMMONPOOL. For an overview of our "data funnel" see Figure 7.

1. For the first step, we use parse Common Crawl metadata files to harvest image-text pairs (Section D). We use `img2dataset` [5] to obtain ~16.8B downloaded samples. This is the first, unfiltered version of COMMONPOOL, and contains only basic information for our images (i.e., the original image height, width, and alt-text caption). During this step we also resize images such that their largest dimension does not exceed 512 pixels. This eases storage requirements for large images, but is still larger than the 224 pixel resolution used for later training stages.

2. For the second step, we process our unfiltered pool and create richer metadata for each image-text pair. We generate the following for each sample:
   - CLIP ViT-B/32 and CLIP ViT-L/14 image and text features, with their associated similarities.
   - NSFW scores for the image and the text, using the analysis described in Appendix E.
   - Deduplication score for the image, as described in Appendix F.

- Bounding boxes for faces detected in the image, using the method described in Appendix G.

3. For the third and final step, we filter our image-text pairs based on the metadata generated during the second stage. We filter out image-text pairs where the NSFW and deduplication scores exceed the respective thresholds (Section E). From the images that pass through this filtering, we keep only the desired amount (e.g., 12.8B images from the `xlarge` COMMONPOOL). Smaller pools are telescoping subsets of larger pools. We package the metadata and image urls, which is made publicly available to the participants. Note, we do not release raw image data but rather image urls pointing to images.

A summary of the metadata for each sample is found in Table 8. To validate our pipeline for duplication and CLIP feature correctness, we also take ImageNet train though metadata generation as a unit test. Using the deduplication features, we detect that 100% of the images are in fact duplicates. Additionally using the CLIP ViT-B/32 and CLIP ViT-L/14 image features and corresponding text features from OpenAI's 80-prompt ensemble, we achieve 63.36% and 75.54% top-1 accuracies, which match the performance reported in the CLIP paper [111].

When creating pools of different scale (i.e., number of samples), we ensure that smaller pools are subsets of larger pools. For instance, the `small` COMMONPOOL is a subset of the `xlarge` COMMONPOOL.

After COMMONPOOL is created, the participants can then download the final image-text pairs using the provided files via `img2dataset`. To further ease the computational burden on participants, we additionally provide metadata for each sample in COMMONPOOL. Note that when downloading, our `img2dataset` configuration automatically blurs faces. Hence this is an automatic step on not something participants must do ad hoc.

# I COMMONPOOL statistics

To provide more information about the kinds of samples in our COMMONPOOL, we conduct additional analysis on the `small` pool, which is an i.i.d. sample of downloaded data and a subset of the larger pools.

In Figure 8 we show CLIP similarity similarity scores between images and their corresponding text. We notice a flatter distribution of CLIP ViT-L/14 scores than corresponding B/32 scores.

Turning our attention to images in COMMONPOOL, in Figure 9, we visualize the aspect ratios and sizes of original images (i.e., before they are downloaded and resized). In Figure 10, we display a distribution of image height and width after *download* resizing. Notice that the majority of images are around $224 \times 224$ pixels, which is the final resized resolution used for training.

Analysing the textual component of each sample, we visualize frequency of the number of CLIP BPE tokens in the captions (Figure 11) and most common languages (Figure 12). Token counts follow a long-tailed distribution with much more mass in the short sequence range, while English is the predominant language in COMMONPOOL according to fasttext and cld3.

We also look at url statistics. In Figure 13 we see common domain names in COMMONPOOL (e.g., wordpress domains) and common suffixes (e.g., .com or .net).

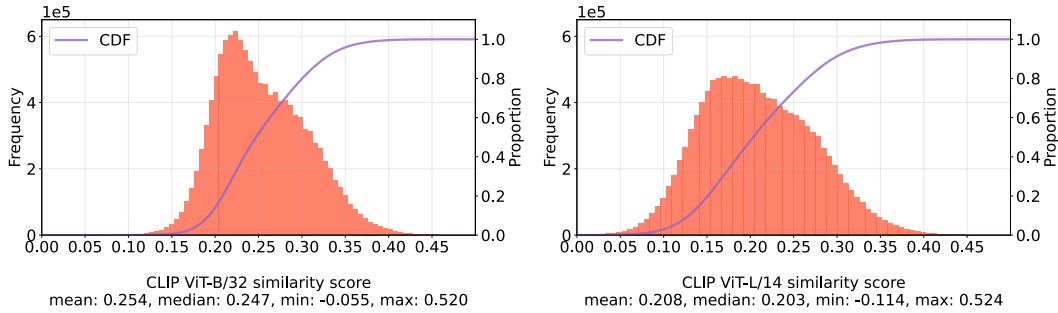

Figure 8: Image-text similarity score distributions using CLIP ViT-B/32 *(left)* and ViT-L/14 *(right)* models. We plot samples from the `small` COMMONPOOL, which are an i.i.d. sample of the `xlarge` COMMONPOOL.

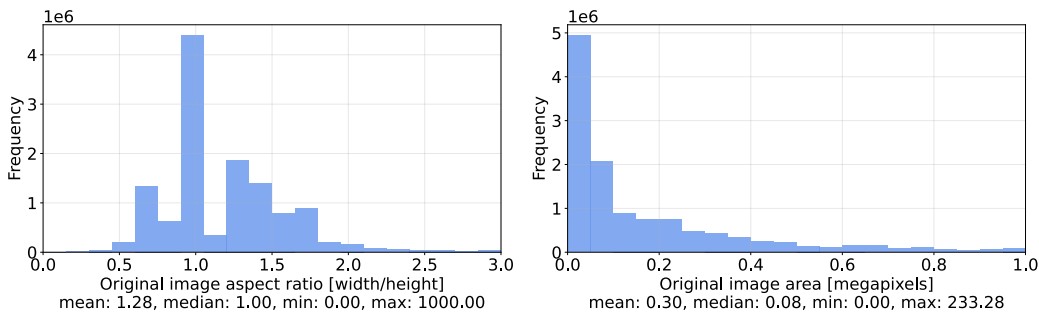

Figure 9: Statistics for images in the `small` COMMONPOOL, before applying resizing.

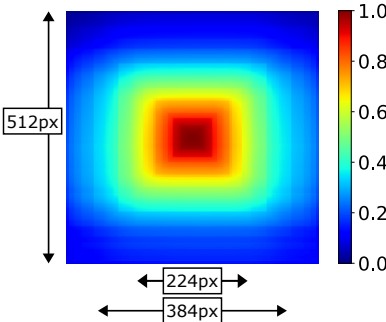

Figure 10: **Image pixel heatmap.** Each entry in the above heatmap represents the estimated probability that a pixel is occupied. The center entry has a value of 1.0 as every image has a center pixel. We compute the heatmap over the `small` COMMONPOOL. Note that image sizes are bounded as we resize all images such that their max dimension does not exceed 512 pixels during dataset download.

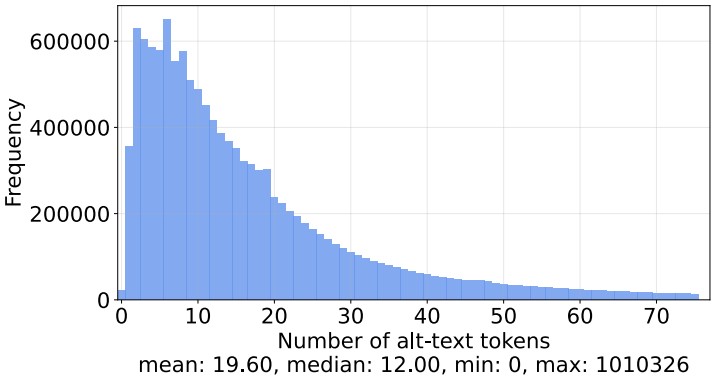

Figure 11: Distribution of token length for alt-text in the `small` COMMONPOOL. The CLIP BPE tokenizer is used for tokenization.

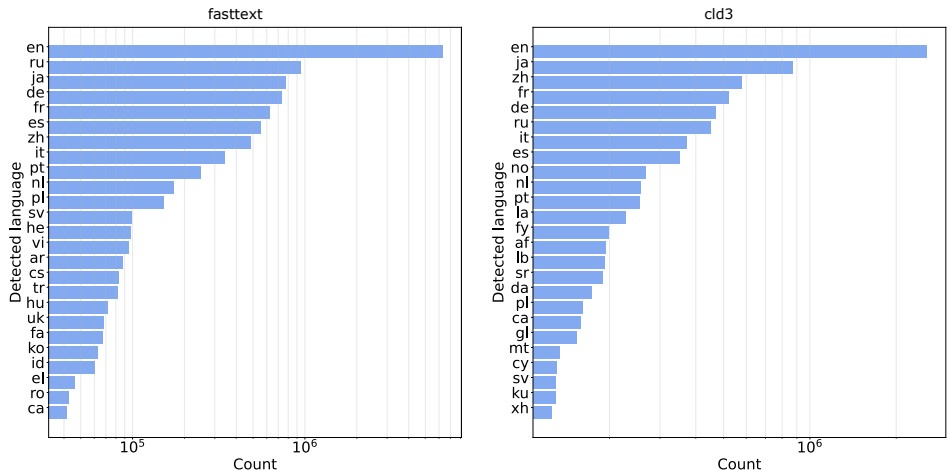

Figure 12: Counts for the top 25 most frequent languages in the `small` COMMONPOOL, as predicted by fasttext *(left)* and cld3 *(right)*.

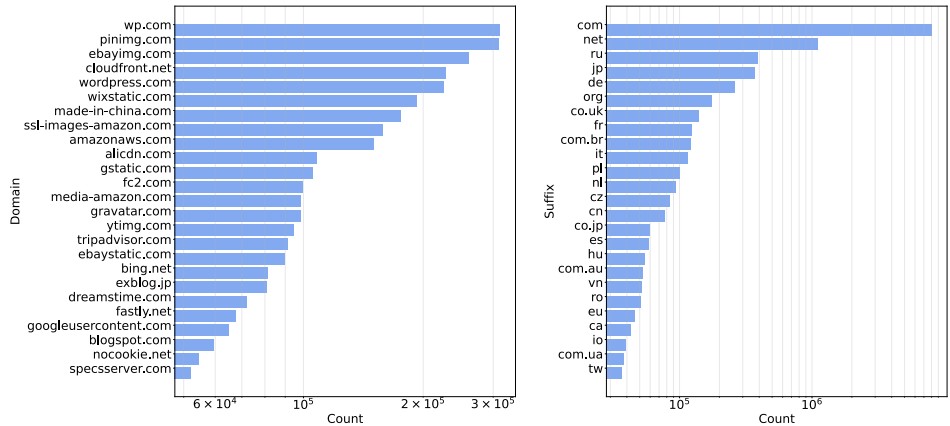

Figure 13: Counts for the top 25 most frequent domains *(left)* and suffixes *(right)* in the `small` COMMONPOOL.

## J Efficient training on data subsets

When training at large scale, it is important to use efficient access patterns to load training data. This typically means that data must be loaded using large sequential reads instead of random reads in order to maximize throughput. In DATACOMP, this is facilitated by the WebDataset[5] format which stores the training examples in tar files (called "shards") and WebDataLoader which makes it easy to load data stored in this format.

Given an arbitrary subset of a pool, we would like to efficiently train on that subset. Because WebDataset format does not permit efficient random access (a feature inherited from tar), we must read through the entire pool to select the required images. There are two ways to implement this filtering:

1. **Filter during training:** we apply a predicate during training data loading that discards data not present in the subset.

2. **Filter before training:** we iterate over the pool, selecting the images in the subset, and write them to a new WebDataset.

After some profiling, we concluded that option 1 had too much overhead in the case where the subset is much smaller than the pool. To see why, note that if the subset is an $p$-fraction of the pool size, then we would end up reading a $1/p$ factor more data than needed for training. Instead, we give an implementation of option 2, which performs at most twice as many reads as needed for training.[6]

Our tool, called the *resharder*, reads a set of uids in NumPy array format, scans through the pool, selecting those examples, and writes them to a new WebDataset. The resharder uses multiprocessing to make good use of hardware and can be distributed over many computers to further increase throughput. The resharder also supports streaming data to and from cloud storage such as Amazon S3. The resharder is provided to participants as part of the competition tooling.

## K Effect of duplicates in the training data

Given that COMMONPOOL was constructed by scraping the web for image and text pairs, there is a likelihood that some of our images are duplicates of each other, even if they originated from different web sources and have different captions. Here we examine the effect of removing such duplicates. We used the technique proposed by Webster et al. [144], where CLIP image features are first compressed and then used to do an approximate nearest neighbor search. After this process, two images $x$ and $y$ are considered duplicates if $\frac{|d_{ADC}(x,x) - d_{ADC}(x,y)|}{d_{ADC}(x,x)} < T_{ADC}$, where $T_{ADC}$ is some threshold and $d_{ADC}(x,x)$ is the distance of a vector with its quantized version used for approximate nearest neighbor search. For each image, we search duplicates across its 1000 nearest neighbors, and keep it if it's the one with the highest CLIP ViT-L/14 similarity score across its duplicates. Results can be seen in Table 9, both when this technique is used by itself and in conjunction with ViT-B/32 filtering. We can see that results are similar to when only using CLIP filtering.

---

[5]https://github.com/webdataset/webdataset
[6]Since in DATACOMP, the number of examples seen is equal to the pool size.

Table 9: Effect of deduplication of training set for the medium size COMMONPOOL. The filtering performed here is CLIP B32 score top 30% (see Table 26). Higher threshold values lead to more samples being labeled as duplicates.

| Subset | Training dataset size | ImageNet accuracy | Average performance |
|---|---|---|---|
| $T_{ADC} = 0.1$, without filtering | 99.8M | 0.195 | 0.275 |
| $T_{ADC} = 0.2$, without filtering | 85.9M | 0.200 | 0.277 |
| $T_{ADC} = 0.5$, without filtering | 29.6M | 0.227 | 0.295 |
| $T_{ADC} = 0.1$, with filtering | 33.5M | 0.288 | 0.337 |
| $T_{ADC} = 0.2$, with filtering | 30.6M | 0.289 | 0.337 |
| $T_{ADC} = 0.5$, with filtering | 15.5M | 0.252 | 0.311 |

Table 10: Batch size ablation at the `medium` scale. We compare the standard DATACOMP `medium` configuration, with batch size 4096 against an ablated configuration with batch size 8192 (`medium: batch size 2x`). We find that the rankings of the baseline filtering strategies are relatively consistent. More precisely, the rank correlation is 0.96 on ImageNet and 0.98 for the Average over 38 datasets.

| Scale | Filtering strategy | Dataset size | Samples seen | ImageNet | Average over 38 datasets | Delta ranking ImageNet | Delta ranking Average |
|---|---|---|---|---|---|---|---|
| medium | No filtering | 128M | 128M | 0.176 | 0.258 | - | - |
| | Basic filtering | 30M | 128M | 0.226 | 0.285 | - | - |
| | Text-based | 31M | 128M | 0.255 | 0.307 | - | - |
| | Image-based | 29M | 128M | 0.268 | 0.312 | - | - |
| | LAION-2B filtering | 13M | 128M | 0.230 | 0.292 | - | - |
| | CLIP score (L/14 30%) | 38M | 128M | 0.273 | 0.328 | - | - |
| | Image-based ∩ CLIP score (L/14 30%) | 14M | 128M | 0.297 | 0.328 | - | - |
| medium: batch size 2x | No filtering | 128M | 128M | 0.171 | 0.258 | 0 | 0 |
| | Basic filtering | 30M | 128M | 0.219 | 0.277 | +1 (worse) | 0 |
| | Text-based | 31M | 128M | 0.251 | 0.299 | 0 | -1 (better) |
| | Image-based | 29M | 128M | 0.260 | 0.299 | 0 | 0 |
| | LAION-2B filtering | 13M | 128M | 0.215 | 0.288 | -1 (better) | 0 |
| | CLIP score (L/14 30%) | 38M | 128M | 0.271 | 0.324 | 0 | 0 |
| | Image-based ∩ CLIP score (L/14 30%) | 14M | 128M | 0.276 | 0.311 | 0 | +1 (worse) |

## L  Hyperparameter ablations

Recall that in DATACOMP, we freeze the training procedure and hyperparameters to focus the competition on dataset curation. However, this leads to the natural question: do "better" datasets (i.e., datasets that lead to higher accuracy models on zero-shot downstream tasks) remain consistent when training is modified. Hence we ablate key experimental choices: batch size, model architecture, and number of training steps.

### L.1  Batch size

We ablate over the batch size hyperparameter, doubling the batch size at the `medium` scale, but holding all other hyperparameters constant. As see in Table 10, we find that the delta rankings are largely consistent, for both ImageNet and Average performance, with rankings changing by at most plus or minus one position. More specifically, rank correlation before and after doubling batch size is 0.96 for ImageNet and 0.98 for the Average over 38 datasets metric.

### L.2  Model architecture

We choose to use the ViT architecture [39] because of favorable CLIP scaling trends over vanilla ResNets [62] as reported by Radford et al. [111]. However, we still hope that better datasets for downstream ViT performance will lead to better datasets to train convolutional architectures. We look at the `medium` scale, swapping the ViT-B/32 architecture with a ConvNeXt model [93] with matched giga multiplier–accumulate operations (GMACs). Looking at Table 11, we see that ranking of different filtering methods is again relatively consistent (i.e., 1.0 rank correlation for ImageNet and 0.87 rank correlation for the average metric). We conclude that improvements in dataset filtering have potential to improve more than just CLIP ViT model performance.

### L.3  Number of training steps

Recall that one of our major design decisions for DATACOMP is to fix the hyperparameters associated with model training, following closely hyperparameters from prior work [111]. We choose to fix hyperparameters to place emphasis on data curation and remove confounders arising from hyperparameter differences between participants. Here we ablate our hyperparameter configuration by training `small` baselines for 10× more steps. In Figure 14 we see positive correlation for ImageNet accuracy for the ablated and original hyperparameter configurations. We see similar correlation for average performance. See Table 12 for specific values.

Table 11: Architure ablation at the `medium` scale. We compare the standard DATACOMP `medium` configuration, with a ViT-B/32 model against an ablated configuration (`medium: ConvNeXt`), which uses a ConvNeXt model with the same number of multiply-accumulate operations as the ViT. We find that the rankings of the baseline filtering strategies are relatively consistent. More precisely, the rank correlation is 1.0 on ImageNet and 0.87 for the Average over 38 datasets.

| Scale | Filtering strategy | Dataset size | Samples seen | ImageNet | Average over 38 datasets | Delta ranking ImageNet | Delta ranking Average |
|---|---|---|---|---|---|---|---|
| medium | No filtering | 128M | 128M | 0.176 | 0.254 | - | - |
| | Basic filtering | 30M | 128M | 0.226 | 0.280 | - | - |
| | Text-based | 31M | 128M | 0.255 | 0.301 | - | - |
| | Image-based | 29M | 128M | 0.268 | 0.307 | - | - |
| | LAION-2B filtering | 13M | 128M | 0.230 | 0.287 | - | - |
| | CLIP score (L/14 30%) | 38M | 128M | 0.273 | 0.323 | - | - |
| | Image-based ∩ CLIP score (L/14 30%) | 14M | 128M | 0.297 | 0.323 | - | - |
| medium: ConvNeXt | No filtering | 128M | 128M | 0.178 | 0.255 | 0 | 0 |
| | Basic filtering | 30M | 128M | 0.232 | 0.272 | 0 | 0 |
| | Text-based | 31M | 128M | 0.255 | 0.298 | 0 | 0 |
| | Image-based | 29M | 128M | 0.270 | 0.298 | 0 | +1 (better) |
| | LAION-2B filtering | 13M | 128M | 0.253 | 0.300 | 0 | -2 (better) |
| | CLIP score (L/14 30%) | 38M | 128M | 0.279 | 0.326 | 0 | +1 (worse) |
| | Image-based ∩ CLIP score (L/14 30%) | 14M | 128M | 0.323 | 0.331 | 0 | 0 |

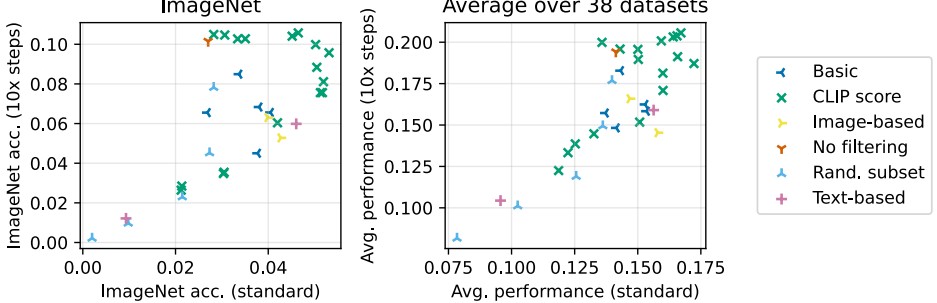

Figure 14: *(left)* The effect of training for $10\times$ steps for for `small` filtering track baselines on ImageNet. *(right)* Similar plot but for Avg. performance. While the ordering of some methods changes quite drastically, we, in general, see a positive correlation.

Table 12: Experiment details when extending the number of steps by 10 times the standard amount for that scale.

| Scale | Filtering | ImageNet | ImageNet dist. shifts | VTAB | Retrieval | Average over 38 datasets |
|---|---|---|---|---|---|---|
| small | No filtering | 0.102 | 0.093 | 0.204 | 0.147 | 0.196 |
| | Random subset(75%) | 0.078 | 0.072 | 0.182 | 0.129 | 0.178 |
| | Random subset(50%) | 0.045 | 0.049 | 0.161 | 0.104 | 0.150 |
| | Random subset(25%) | 0.023 | 0.029 | 0.134 | 0.075 | 0.119 |
| | Random subset(10%) | 0.010 | 0.018 | 0.119 | 0.069 | 0.101 |
| | Random subset(1%) | 0.002 | 0.006 | 0.097 | 0.056 | 0.082 |
| | Caption length | 0.085 | 0.080 | 0.198 | 0.136 | 0.184 |
| | Image size | 0.066 | 0.064 | 0.153 | 0.115 | 0.158 |
| | English (fasttext) | 0.068 | 0.068 | 0.172 | 0.108 | 0.159 |
| | English (fasttext) and caption length | 0.066 | 0.065 | 0.182 | 0.106 | 0.163 |
| | English (fasttext), caption length, and image size | 0.045 | 0.048 | 0.164 | 0.092 | 0.149 |
| | CLIP B32 score top 10% | 0.035 | 0.046 | 0.162 | 0.079 | 0.139 |
| | CLIP B32 score top 20% | 0.076 | 0.076 | 0.182 | 0.099 | 0.172 |
| | CLIP B32 score top 30% | 0.096 | 0.090 | 0.221 | 0.121 | 0.205 |
| | CLIP B32 score top 40% | 0.081 | 0.077 | 0.200 | 0.124 | 0.193 |
| | CLIP B32 score top 50% | 0.106 | 0.097 | 0.211 | 0.134 | 0.205 |
| | CLIP B32 score top 75% | 0.103 | 0.096 | 0.210 | 0.150 | 0.198 |
| | CLIP B32 score top 90% | 0.105 | 0.096 | 0.212 | 0.152 | 0.202 |
| | CLIP B32 threshold at 0.3 + English filter | 0.029 | 0.036 | 0.152 | 0.078 | 0.134 |
| | CLIP B32 threshold at 0.28 + English filter | 0.035 | 0.041 | 0.168 | 0.086 | 0.145 |
| | CLIP B32 threshold at 0.3 | 0.076 | 0.078 | 0.199 | 0.102 | 0.182 |
| | CLIP L14 score top 10% | 0.026 | 0.037 | 0.130 | 0.073 | 0.123 |
| | CLIP L14 score top 20% | 0.060 | 0.064 | 0.161 | 0.096 | 0.153 |
| | CLIP L14 score top 30% | 0.088 | 0.087 | 0.199 | 0.115 | 0.188 |
| | CLIP L14 score top 40% | 0.100 | 0.096 | 0.217 | 0.122 | 0.207 |
| | CLIP L14 score top 50% | 0.104 | 0.098 | 0.212 | 0.136 | 0.203 |
| | CLIP L14 score top 75% | 0.103 | 0.095 | 0.189 | 0.146 | 0.191 |
| | CLIP L14 score top 90% | 0.105 | 0.095 | 0.203 | 0.145 | 0.198 |
| | Image-based clustering (ImageNet1k) | 0.053 | 0.053 | 0.162 | 0.091 | 0.146 |
| | Image-based clustering (ImageNet21k) | 0.063 | 0.059 | 0.173 | 0.108 | 0.167 |
| | Text-based clustering (ImageNet1k) | 0.012 | 0.018 | 0.120 | 0.062 | 0.104 |
| | Text-based clustering (ImageNet21k) | 0.262 | 0.216 | 0.305 | 0.246 | 0.300 |
| | Intersect IN1k image clustering and CLIP B32 score top 30% | 0.058 | 0.059 | 0.179 | 0.098 | 0.161 |
| | Intersect IN1k image clustering and CLIP L14 score top 30% | 0.049 | 0.051 | 0.171 | 0.090 | 0.150 |
| | Intersect IN21k image clustering and CLIP B32 score top 30% | 0.071 | 0.070 | 0.192 | 0.107 | 0.175 |
| | Intersect IN21k image clustering and CLIP L14 score top 30% | 0.064 | 0.065 | 0.200 | 0.096 | 0.173 |
| medium | No filtering | 0.370 | 0.304 | 0.387 | 0.355 | 0.383 |
| | English (fasttext), caption length, and image size | 0.317 | 0.269 | 0.324 | 0.271 | 0.334 |
| | CLIP B32 score top 30% | 0.436 | 0.351 | 0.433 | 0.345 | 0.430 |
| | CLIP B32 score top 40% | 0.434 | 0.353 | 0.448 | 0.365 | 0.442 |
| | CLIP B32 score top 50% | 0.426 | 0.352 | 0.439 | 0.377 | 0.433 |
| | CLIP B32 score top 75% | 0.398 | 0.325 | 0.396 | 0.374 | 0.411 |
| | Image-based clustering (ImageNet1k) | 0.363 | 0.294 | 0.347 | 0.279 | 0.347 |
| | Image-based clustering (ImageNet21k) | 0.374 | 0.303 | 0.372 | 0.318 | 0.372 |
| | Intersect IN1k image clustering and CLIP B32 score top 30% | 0.415 | 0.330 | 0.413 | 0.310 | 0.403 |
| | Intersect IN1k image clustering and CLIP L14 score top 30% | 0.405 | 0.325 | 0.399 | 0.295 | 0.387 |

# M  Detector-based baselines

While controlling for factors such as class balance is common in the supervised settings, experimenting with analogous strategies in the context of multimodal datasets and CLIP training is a pertinent direction. Towards this end, we use the Detic detector [160] to annotate the `medium` pool (128M samples) by extracting bounding boxes and class labels for the 1203 LVIS [54] objects categories. Following the original Detic paper, we retain predictions whose confidence score exceeds 0.5. Based on these annotations, we construct the following five strategies:

- *Object exists*: Subset for which there exists at least one detection from the 1203 LVIS categories.

- *Object centered*: Subset for which there exists at least one detection from the 1203 LVIS categories with a bounding box center falling in the center grid cell of a 3x3 grid superimposed on the image.

- Balancing by class: We define 1204 buckets—1203 buckets corresponding to the LVIS classes and an additional bucket for images that do not have any detections. For each image in the medium pool, we assign the image to the bucket(s) corresponding to the detected classes. We then construct a dataset such that there are an equal number of samples from each bucket and the total number of samples specified by a particular scale (e.g., 128M samples for medium scale). Note, for rare classes there can be many repeated samples and for common classes only a subset of the total samples will be in the dataset.

- Balancing by position: We define 26 buckets—0, 1, ..., 24 corresponding to 5x5 grid locations in an image. An image is added to bucket(s) when it contains a bounding box whose center falls in the bucket's grid cell. The 25th bucket contains images for which there are no detections. We again construct a dataset such that there are an equal number of samples from each bucket.

- Balancing by count: We define 12 buckets—0, 1, ..., 10 corresponding to zero to ten detections in an image and a twelfth bucket corresponding to images with more than ten detections. We yet again construct a dataset such that there are an equal number of samples from each bucket.

We employ each of these strategies on the `medium` scale. Since the above strategies can be composed with any starting pool, we additionally apply each of the above Detic-based strategies to our previous best `medium` scale filtered pool: Image-based ∩ CLIP score (L/14 30%). This yields five more datasets for 10 baselines in total.

Our results are summarized in the Table 13. In summary: 1) The Image-based ∩ CLIP score (L/14 30%) baseline still performs best. 2) Balancing data in the context of multimodal CLIP training

Table 13: Detector-baseed baselines at the `medium` scale. We start with No filtering and Image-based *cap* CLIP score (L/14 30%) pools and apply five additional filtering and balancing strategies described in Appendix M. We find that even with these more sophisticated strategies, the No filtering and Image-based *cap* CLIP score (L/14 30%) still performs best at medium scale. Properly balancing multimodal data remains an open direction for future work.

| Scale | Filtering strategy | Samples seen | ImageNet | Average over 38 datasets |
|---|---|---|---|---|
| | No filtering | 128M | 0.176 | 0.258 |
| | ∩ Object exists | 128M | 0.181 | 0.263 |
| medium | ∩ Object centered | 128M | 0.187 | 0.263 |
| | ∩ Balance by class | 128M | 0.038 | 0.141 |
| | ∩ Balance by position | 128M | 0.040 | 0.148 |
| | ∩ Balance by object count | 128M | 0.127 | 0.221 |
| | Image-based ∩ CLIP score (L/14 30%) | 128M | 0.297 | 0.328 |
| | ∩ Object exists | 128M | 0.289 | 0.319 |
| medium | ∩ Object centered | 128M | 0.247 | 0.286 |
| | ∩ Balance by class | 128M | 0.034 | 0.136 |
| | ∩ Balance by position | 128M | 0.036 | 0.136 |
| | ∩ Balance by object count | 128M | 0.068 | 0.169 |

Table 14: Experimental configuration for each scale, including the size of the pool we provide, the model architecture and hyperparameters.

| Scale | Model | Train compute (MACs) | Pool size | # samples seen | Learning rate | AdamW $\beta_2$ | Warmup | Batch size |
|---|---|---|---|---|---|---|---|---|
| small | ViT-B/32 | $9.5 \times 10^{16}$ | 12.8M | 12.8M | 5e-4 | 0.98 | 500 | 4096 |
| medium | ViT-B/32 | $9.5 \times 10^{17}$ | 128M | 128M | 5e-4 | 0.98 | 500 | 4096 |
| large | ViT-B/16 | $2.6 \times 10^{19}$ | 1.28B | 1.28B | 5e-4 | 0.98 | 500 | 8192 |
| xlarge | ViT-L/14 | $1.1 \times 10^{21}$ | 12.8B | 12.8B | 1e-3 | 0.95 | 10k | 90112 |

remains an open problem. All balancing strategies lead to divergence of the CLIP contrastive loss and result in poor model performance. We hypothesize that this is due to the long-tailed nature of the data distribution, which leads to many repeated samples in our balanced data construction. This in turn, increases the likelihood that samples are contrasted with themselves in the loss computation.

## N   Training details

The full set of hyperparameters used for each scale is shown in Table 14. For choosing hyperparameters, we follow the OpenCLIP library [69], an open source reproduction of OpenAI's CLIP. For the small, medium, and large tracks, these hyperparameters are equal to those in the CLIP paper, except with reduced batch size so that training runs on reasonable hardware. For the xlarge track, batch size is increased from that in OpenAI's CLIP to accelerate training by allowing the use of many GPUs simultaneously with high utilization. For this run we also double the learning rate following prior work [28].

## O   Evaluation details

Models are evaluated over a wide range of 38 tasks to measure proficiency in various domains. We include 22 of the 27 classification tasks in the test suite of Radford et al. [111], excluding the few datasets that have license restrictions, are in video format, or are no longer available in their original form. We include 6 datasets that were designed to test generalization of models trained on ImageNet. We also include a majority of the Visual Task Adaptation Benchmark, excluding 3 datasets that are ill-suited for zero-shot evaluation [156]. We include 3 datasets from the WILDS benchmark, which tests robustness to distribution shifts and spurious correlations [83, 127]. Finally, we include 2 additional datasets, Dollar Street and GeoDE, which test robustness of classification performance across income levels and geographical regions [122, 114]. Furthermore, we evaluate zero-shot image and text retrieval on the Flickr30k and MSCOCO datasets, and image association on the WinoGAViL dataset [151, 26, 17]. The complete list of evaluation tasks is given in Table 15. We show a sample from each dataset in Figure 15.

**Prompt choice.** Since we perform zero-shot evaluation, prompt and class name selection is important, and can have a significant impact on the results. To avoid heavy prompt engineering and overtuning to individual models, we opt to use the prompt templates used in Radford et al. [111] whenever possible. Most datasets come with pre-defined class names, but some are overwritten with more descriptive labels, again based on previous literature. For datasets with no precedent in zero-shot evaluation, we reuse prompt templates from other datasets with a similar domain and task (e.g., SVHN is evaluated with MNIST prompts and class names).

**Evaluation metrics.** For the majority of classification tasks, the primary evaluation metric is accuracy. For certain datasets with class imbalances, we instead compute mean per-class accuracy, as done in Radford et al. [111]. On the WILDS benchmark datasets, we use the primary metric specified for each dataset on their leaderboard. Dollar Street and GeoDE test model generalization across socioeconomic and geographic diversity. Thus, for Dollar Street, we compute worst-group top-5 accuracy, with groups defined by income level, emulating Rojas et al. [122]; for GeoDE, we compute worst-group accuracy, with groups defined by region (Africa, Americas, West Asia, East Asia, Southeast Asia, and Europe), as defined in Ramaswamy et al. [114]. For the image-text retrieval tasks, Flickr and MSCOCO, we compute both image and text recall (fraction of text captions for which the correct image was selected and vice versa), and plot their arithmetic mean. On WinoGAViL, we compute the

Table 15: Evaluation tasks.

| Task type | Dataset | Task | Test set size | Number of classes | Main metric | Clean |
|---|---|---|---|---|---|---|
| Classification | Caltech-101 [45] | Object recognition | 6,085 | 102 | mean per class | ✓ |
| | CIFAR-10 [86] | Visual recognition | 10,000 | 10 | accuracy | ✓ |
| | CIFAR-100 [86] | Visual recognition | 10,000 | 100 | accuracy | ✓ |
| | CLEVR Counts [76, 156] | Counting | 15,000 | 8 | accuracy | |
| | CLEVR Distance [76, 156] | Distance prediction | 15,000 | 6 | accuracy | |
| | Country211 [111, 140] | Geolocation | 21,100 | 211 | accuracy | ✓ |
| | DTD [30] | Texture classification | 1,880 | 47 | accuracy | ✓ |
| | EuroSAT [63, 156] | Satellite imagery recognition | 5,400 | 10 | accuracy | ✓ |
| | FGVC Aircraft [95] | Aircraft recognition | 3,333 | 100 | mean per class | ✓ |
| | Food-101 [18] | Food recognition | 25,250 | 101 | accuracy | ✓ |
| | GTSRB [137] | Traffic sign recognition | 12,630 | 43 | accuracy | ✓ |
| | ImageNet 1k [37] | Visual recognition | 50,000 | 1,000 | accuracy | ✓ |
| | ImageNet Sketch [143] | Visual recognition | 50,889 | 1,000 | accuracy | ✓ |
| | ImageNet V2 [121] | Visual recognition | 10,000 | 1,000 | accuracy | ✓ |
| | ImageNet-A [65] | Visual recognition | 7,500 | 200 | accuracy | ✓ |
| | ImageNet-O [65] | Visual recognition | 2,000 | 200 | accuracy | ✓ |
| | ImageNet-R [64] | Visual recognition | 30,000 | 200 | accuracy | ✓ |
| | KITTI distance [48, 156] | Distance prediction | 711 | 4 | accuracy | |
| | MNIST [89] | Digit recognition | 10,000 | 10 | accuracy | ✓ |
| | ObjectNet [13] | Visual recognition | 18,574 | 113 | accuracy | ✓ |
| | Oxford Flowers-102 [102] | Flower recognition | 6,149 | 102 | mean per class | ✓ |
| | Oxford-IIIT Pet [105, 156] | Pet classification | 3,669 | 37 | mean per class | ✓ |
| | Pascal VOC 2007 [42] | Object recognition | 14,976 | 20 | accuracy | ✓ |
| | PatchCamelyon [142, 156] | Metastatic tissue cls. | 32,768 | 2 | accuracy | |
| | Rendered SST2 [156] | Sentiment classification | 1,821 | 2 | accuracy | ✓ |
| | RESISC45 [27, 156] | Satellite imagery recognition | 6,300 | 45 | accuracy | ✓ |
| | Stanford Cars [85] | Vehicle recognition | 8,041 | 196 | accuracy | ✓ |
| | STL-10 [31] | Visual recognition | 8,000 | 10 | accuracy | ✓ |
| | SUN-397 [146] | Scene recognition | 108,754 | 397 | accuracy | ✓ |
| | SVHN [99, 156] | Digit recognition | 26032 | 10 | accuracy | ✓ |
| | iWildCam [14, 83] | Animal recognition | 42,791 | 182 | macro F1 score | ✓ |
| | Camelyon17 [12, 83] | Metastatic tissue cls. | 85,054 | 2 | accuracy | |
| | FMoW [29, 83] | Satellite imagery recognition | 22,108 | 62 | worst-region acc. | ✓ |
| | Dollar Street [122] | Object recognition | 3,503 | 58 | worst-income top-5 acc. | ✓ |
| | GeoDE [114] | Object recognition | 12,488 | 40 | worst-region acc. | ✓ |
| Retrieval | Flickr30k [151] | Image and text retrieval | 31,014 | N/A | R@1 | ✓ |
| | MSCOCO [26] | Image and text retrieval | 5,000 | N/A | R@1 | ✓ |
| | WinoGAViL [17] | Commonsense association | 3,563 | N/A | Jaccard score | ✓ |

Jaccard score (intersection-over-union) for each example, and show results for the harder samples (10 and 12 candidates). More information on WinoGAViL evaluation can be found in Bitton et al. [17].

**Clean subset.** For five of our evaluation tasks (the two CLEVR tasks, the two Camelyon tasks, and KITTI) the zero-shot performance of all evaluated models appears to be close to that of random guessing, and lack correlation to the type of filtering method used (see Figure 27). Consequently, we studied performance averaged only on the remaining 33 tasks, but found not substantial qualitative differences in our results. As a result, we opted to report the average on the full evaluation suite throughout our study.

**Zero-shot vs. fine-tuning protocols.** One critical decision in DATACOMP is how exactly to evaluate models and whether or not to fine-tune models on evaluation tasks (i.e., supervised fine-tuning directly on task training sets). We opt for zero-shot evaluation, where a models are applied to downstream tasks directly to 1) ease computational burden on participants and 2) measure the out-of-the-box generalization capabilities of our models. To validate this design decision, we conduct linear probes on all models presented in Tables 3 and 18 on ImageNet. We follow a standard probing protocol and fine-tune the last linear layer from zero-shot initialization for 40 epochs with learning rate 1e-3, batch size 256, AdamW optimizer with default settings with the exception of weight decay (that we set to zero), and a cosine annealing schedule. As seen in Figure 16, zero-shot and linear probe performance follow similar trends for both filtering and BYOD tracks. Moreover the Spearman rank correlation between the two protocols over the models considered is 0.99 for the filtering track and 1.0 for BYOD. This suggests that better zero-shot models on ImageNet are correlated with better representations of linear probe fine-tuning on ImageNet.

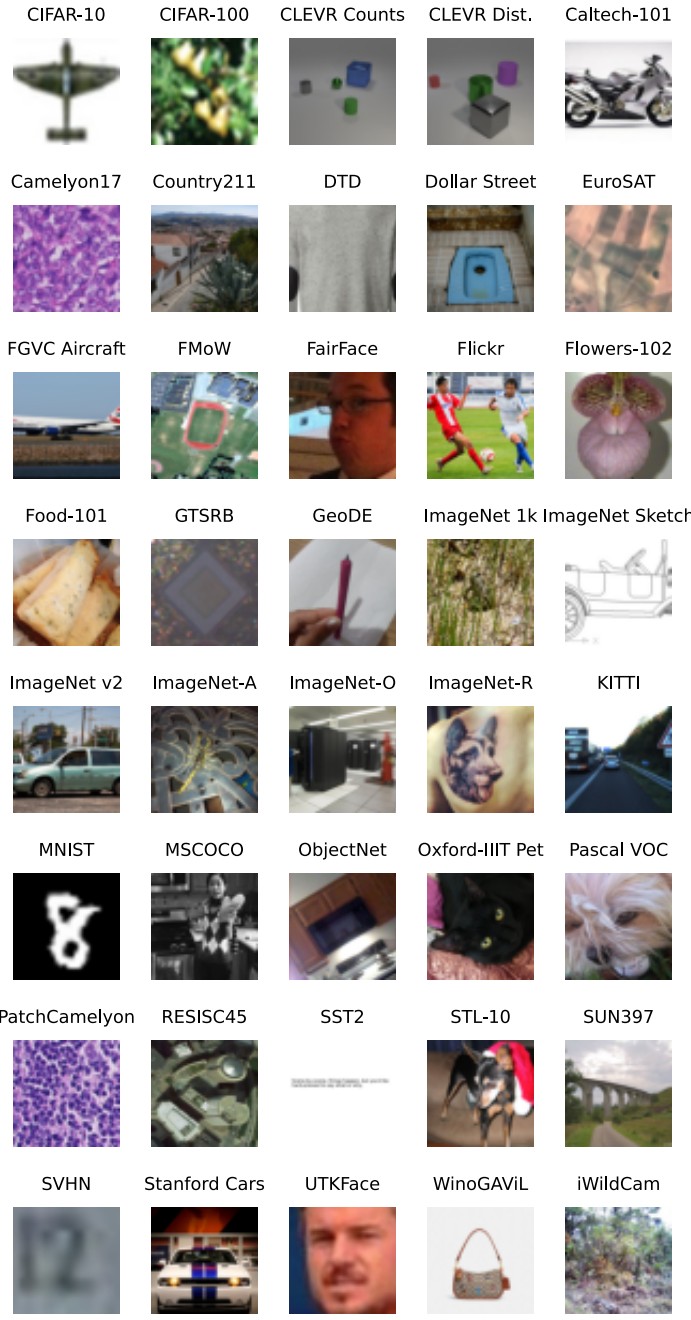

Figure 15: Randomly sampled images from the evaluation datasets we consider.

## O.1  Visual Question Answering

In addition to our evaluation suite containing multiple classification and retrieval tasks, we conducted experiments on visual question answering. More specifically, following Shen et al. [132], we use the CLIP models to contrast images with prompts formed by the questions and each candidate answer, without fine-tuning (i.e., in a zero-shot setting). Using the VQA v1 dataset [2], for each candidate answer, we construct a text prompt that also includes the question following the template `Question:  [question text] Answer:  [answer text]`, as in Ilharco et al. [70]. This text is then fed to CLIP's text encoder. As previously noted by multiple authors, CLIP models struggle on this task, potentially due to the mismatch between the text in the downstream task and the captions seen during pre-training Shen

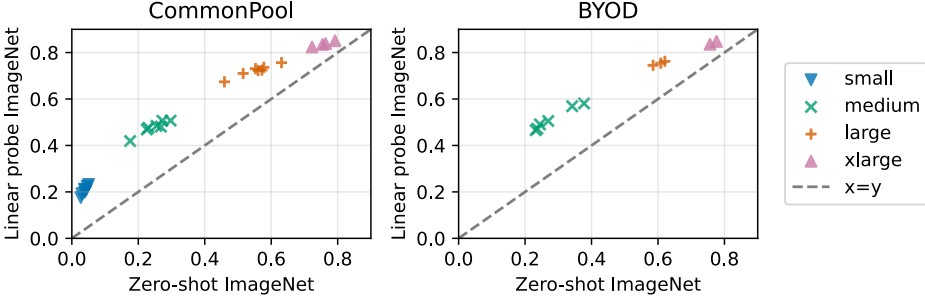

Figure 16: Zero-shot ImageNet and Linear probe ImageNet performance for models from Tables 3 and 18. Relative ordering of models demonstrates high rank correlations of 0.99 and 1.0 for COMMONPOOL and BYOD respectively.

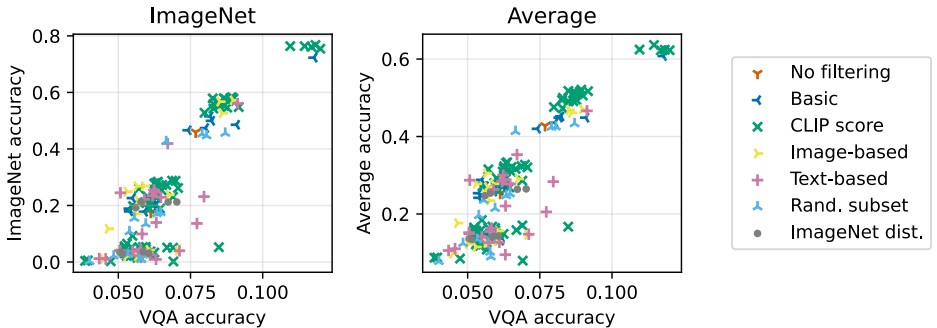

Figure 17: Correlation between zero-shot performance on the VQA v1 dataset and results on ImageNet and our full evaluation suite.

et al. [132], Ilharco et al. [70], Song et al. [134]. Nonetheless, we observe a strong correlation between VQA performance and ImageNet accuracy (0.877) and between VQA performance and average performance on our full evaluation suite. Full results are shown in Figure 17.

## P    Baseline details

Here we provide additional details on the creation of our baseline subsets. To highlight the qualitative differences between the filtering strategies we also provide visualization for *No filtering* (Figure 18), *Basic filtering* (Figure 19), and *CLIP score (L/14 30%)* (Figure 20), which can all be found in Table 3. Notice that No filtering gives relatively noisy data (e.g., matching a bicycle with a caption: "IMG_2187.jpg"), while CLIP score samples give qualitatively more descriptive cations.

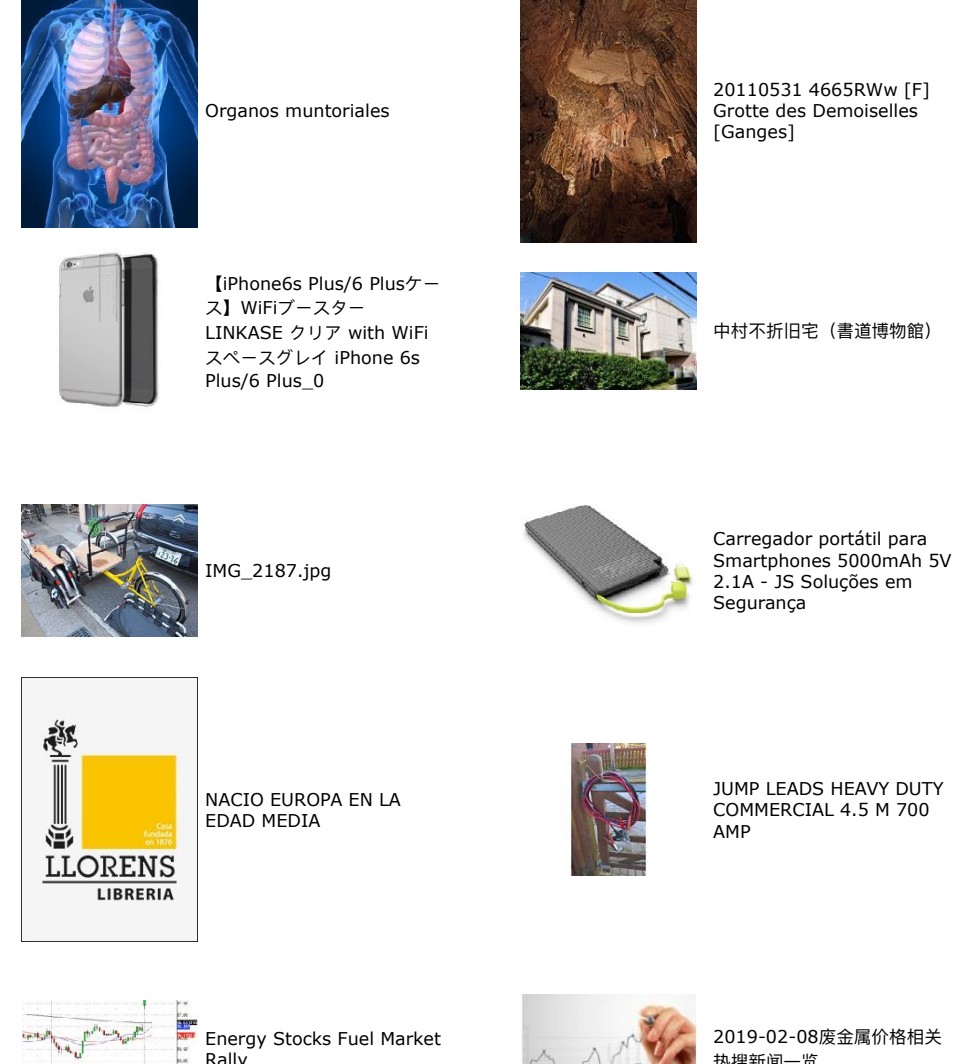

Figure 18: An i.i.d. sample from small CommonPool generated after applying the *No filter* strategy. Hence, these samples represent random images from CommonPool.

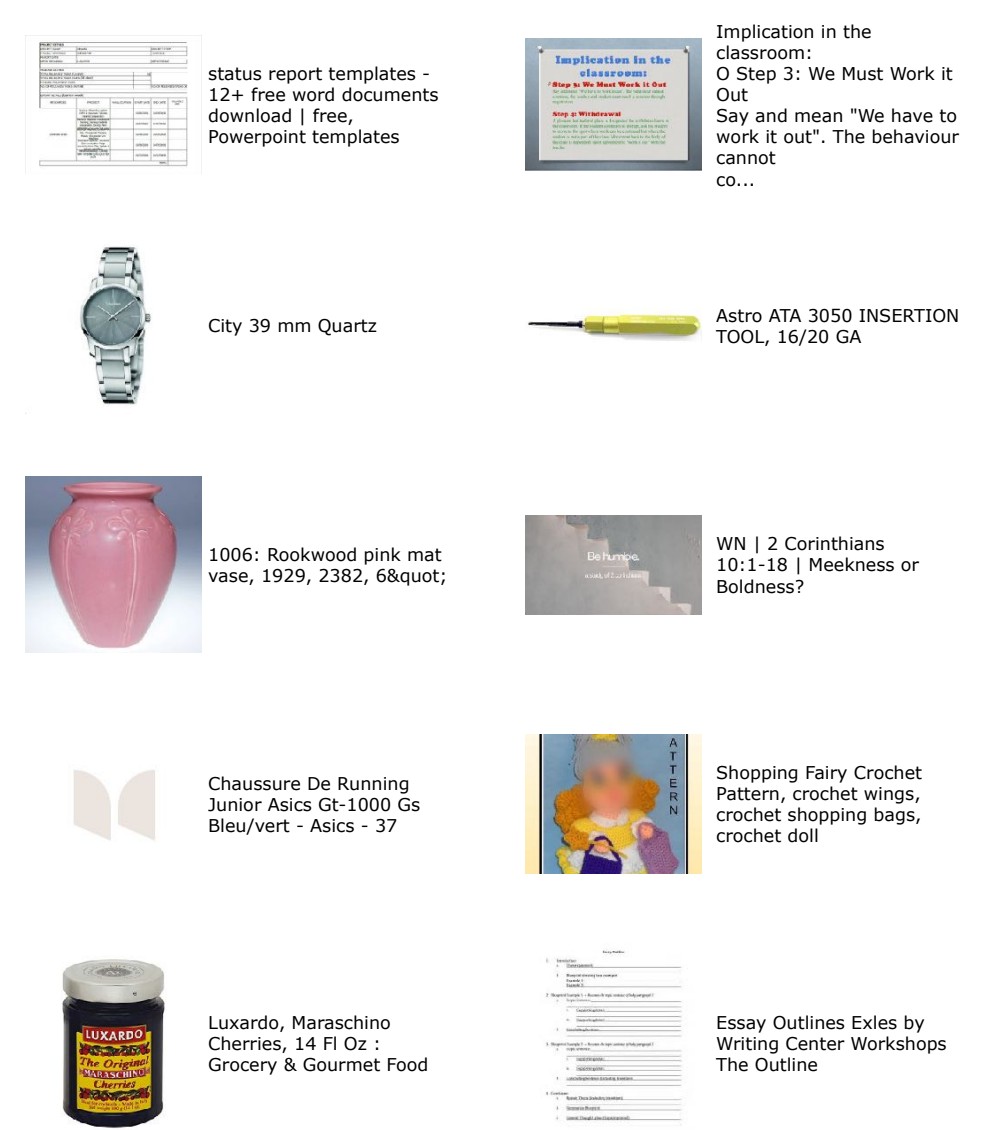

Figure 19: An i.i.d. sample from small COMMONPOOL generated after applying the *Basic filter* strategy.

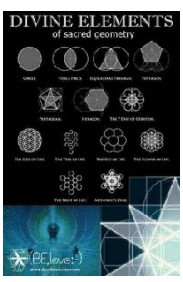 Sacred Geometry Egg Of Life, Sacred Geometry Symbols, Golden Ratio, Flower Of Life, Wicca, Magick, Tattoos, Geometric Nature, Geometric Mandala

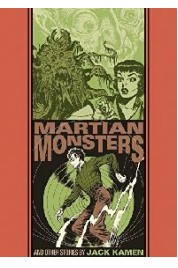 The Martian Monster And Other Stories (The EC Comics Library) ()

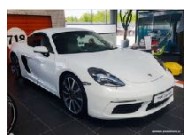 Porsche Cayman S

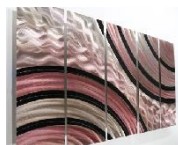 Mesmerizing Black, Silver & Pink Handmade Modern Metal Wall Art Sculpture - Metallic One of a Kind Abstract Painting - OOAK 546 by Jon Allen

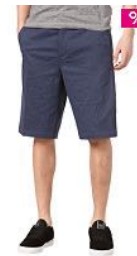 ALPINESTARS Radar Short navy blue

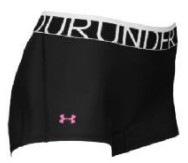 Under Armour Heatgear Gotta Have It Shorty - Women's at Foot Locker

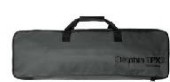 Tripod Delphin TPX3 Silver

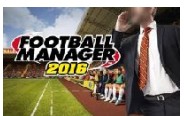 Football Manager 2016

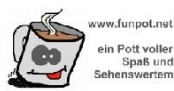 Mett.jpg auf www.funpot.net

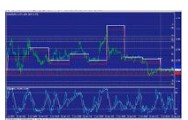 Profitable forex trading systems

Figure 20: An i.i.d. sample from small COMMONPOOL generated after applying the CLIP score (L/14 30%)
strategy.

## P.1 Filtering track

**Basic filtering.** For language detection, we use Fasttext 0.92, version lid.176, and cld3 - library gcld3 3.0.13. We count the number of words in each caption by splitting using whitespaces.

**CLIP thresholds.** We use OpenAI pretrained CLIP ViT-B/32 and ViT-L/14 models [111] to compute the cosine similarity text and image tower outputs as the CLIP scores. On the `small` and `medium` pools, we also experiment with baselines that filter out samples in the top few percentiles of CLIP scores. Specifically, we try baselines that use samples with top {1,2,5}-30% CLIP scores (ViT-B/32 model), and the performance is sightly better on the `small` pool (at most 0.5 gain of averaged accuracy) while slightly worse on the `medium` pool (0.4-0.8 loss of averaged accuracy). In Table 16, we show how the CLIP score thresholds relate to the fraction of the pool retained by the filter.

**Text-based filtering.** Each synset is represented by a synset offset that can be used to retrieve the synset from WordNet. In order to verify if a caption has a word corresponding to a synset from our set we iterate over every word and retrieve the synsets that this word can describe (using nltk.corpus WordNet). Following that, we retrieve the most likely lemma representing that synset, find its synset offset, and check if the number is part of the IN21K or IN1K sets.[7]

**Text-based sampling.** This baseline uses text only to filter labels which mention concepts (synsets) appearing in IN21K, and applies a temperature parameter to control how equally-represented different concepts are in the dataset. For synset $j$, let $N_j$ be the number of examples containing words matched to that synset, where as before for each word we only match the most likely synset. Furthermore, for image-text pair $i$ let $T_i$ be the set of synset matched to the caption.

The probability of sampling example $i$ is proportional to either $\frac{1}{|T_i|} \sum_{j \in T_i} N_j^{\alpha-1}$ (average synset score in the data point) or $\max_{j \in T_i} N_j^{\alpha-1}$ (maximum synset score in the data point), where $\alpha$ is a "temperature" parameter controlling the flatness of the distribution. We sample examples with replacement but discard any example repeated more than 100 times.

**Image-based filtering.** We now provide a detailed description of the Image-based filtering procedure. First, since the core of the procedure concerns only image content, we begin with basic text-bsaed filtering: we remove from the pool only all examples with non-English captions (as determined by fasttext), and all examples whose captions have less than two words or less than six characters.

Next, we use clustering of image embeddings to select a subset of examples whose image content is related to a clean training set of interest. Let $e_1, \ldots, e_M$ denote the CLIP image embeddings of the remaining examples in the pool. We cluster these embeddings into $K = 10^5$ clusters using Faiss with 20 iterations, and let $c_1, \ldots, c_K$ denote the resulting cluster centers. Due to memory constraints, for the `large` and `xlarge` pools, we perform the clustering on a random subset of about 160M examples (that pass the basic text-based filtering). For an embedding vector $v$, let

$$I(v) = \arg\max_{i \leq K} \langle v, c_i \rangle$$

denote the index of the cluster center nearest to $v$ as measured by inner product. Let $f_1, \ldots, f_N$ denote the CLIP image embeddings of a clean supervised training set (we experiment with either ImageNet 1K or ImageNet 21K), and let

$$\mathcal{S} = \{I(f_i) \mid 1 \leq i \leq N\}$$

be the set of cluster indices who are nearest neighbors to some clean training set image. We then keep only images in the pool whose nearest cluster center is in $\mathcal{S}$. That is, out of the $M$ examples passing the text-based filtering, the output subset keeps the examples with indices

$$\{1 \leq j \leq M \mid I(e_j) \in \mathcal{S}\}.$$

**Image-based sampling.** In addition to filtering methods, we experiment with cluster-based sampling methods. First, we compute the score of $i$-th cluster $s_i$ as the number of ImageNet data assigned to this cluster. Then, for parameter $\alpha > 0$ we define a distribution over the pool by sampling cluster $i$ with probability $\frac{s_i^\alpha}{\sum_j s_j^\alpha}$ and uniformly sampling an example for the cluster, rejecting any example repeated more than 100 times. We try 5 different $\alpha$, i.e., $\{0, 0.2, 0.5, 1.0, 2.0\}$, and the best average accuracy is obtained when $\alpha = 0.2$, while the performance is still worse than the image-based filtering on the `small` and `medium` pool. We therefore do not include this line of baselines in the experiments of `large` pool.

---

[7] For the ImageNet 21K synsets, we have used the list in https://storage.googleapis.com/bit_models/imagenet21k_wordnet_ids.txt

Table 16: CLIP threshold filtering configurations. "Fraction" denotes the size of the filtered subset relative to the pool.

| CLIP model | En. filtering | Threshold | Fraction |
|---|---|---|---|
| ViT-B/32 | ✗ | 0.384 | 1% |
| ViT-B/32 | ✗ | 0.358 | 3% |
| ViT-B/32 | ✓ | 0.300 | 10.2% |
| ViT-B/32 | ✗ | 0.325 | 10% |
| ViT-B/32 | ✓ | 0.28 | 7.4% |
| ViT-B/32 | ✗ | 0.300 | 20% |
| ViT-B/32 | ✗ | 0.281 | 30% |
| ViT-B/32 | ✗ | 0.263 | 40% |
| ViT-B/32 | ✗ | 0.247 | 50% |
| ViT-B/32 | ✗ | 0.215 | 75% |
| ViT-B/32 | ✗ | 0.193 | 90% |
| ViT-L/14 | ✗ | 0.364 | 1% |
| ViT-L/14 | ✗ | 0.334 | 3% |
| ViT-L/14 | ✓ | 0.300 | 5.4% |
| ViT-L/14 | ✗ | 0.295 | 10% |
| ViT-L/14 | ✓ | 0.280 | 3.3% |
| ViT-L/14 | ✗ | 0.266 | 20% |
| ViT-L/14 | ✗ | 0.243 | 30% |
| ViT-L/14 | ✗ | 0.222 | 40% |
| ViT-L/14 | ✗ | 0.203 | 50% |
| ViT-L/14 | ✗ | 0.160 | 75% |
| ViT-L/14 | ✗ | 0.129 | 90% |

**ImageNet distance filtering.** We rank the samples in the pool by the minimum embedding distance (1 minus cosine similarity) between its image and the ImageNet images; both embeddings are obtained from OpenAI pretrained CLIP ViT-L/14 model [111]. Then we select top images by different fractions as in image-based filtering methods.

## P.2  BYOD track

We experiment with the following data sources:

- CC12M [24]: images and HTML alt-text crawled and filtered from web pages.

- YFCC15M: this is the 15M subset of the YFCC100M dataset [140] that Radford et al. [111] used for dataset ablation in their CLIP paper.

- RedCaps [38]: 12M images and corresponding captions were crawled from 350 manually curated subreddits between 2008 and 2020.

- Shutterstock: 106M images and captions were obtained from the Shutterstock website in 2021 [101]. We use the "photos" subset of this dataset, with 58M samples, which we found performed best, unless specified otherwise.

- WIT [136]: Image-text pairs from Wikipedia pages. We use the attribution fields as captions, which we found performed best.

- COYO [20]: A collection of 700M image-text pairs from Common Crawl.

- LAION-2B [129]: A 2.32 billion english subset of LAION-5B.

- LAION-COCO: A dataset with 600M images from LAION-5B and synthetic captions.[8]

- LAION-A: According to laion.ai, LAION-A is a 900M subset of LAION-2B [129] with the aesthetic filtering procedure used in LAION-aesthetic[9] and pHash deduplication [72].

In Table 17, we use some heuristics to measure the quality of some external data sources. First, following Nguyen et al. [101], we train a CLIP model on a 5M random subset from each source, and evaluate the performance of the resulting models on ImageNet and ImageNet-derived distributions — ImageNet-V2 [121], ImageNet-R [64],

---

[8]https://laion.ai/blog/laion-coco/
[9]https://github.com/LAION-AI/laion-datasets/blob/main/laion-aesthetic.md

Table 17: Measuring the quality of external data sources

| Dataset | Dataset size | ImageNet acc. | Avg. accuracy ImageNet and OOD sets | Avg. cos. sim. (B/32) | Avg. cos. sim. (L/14) |
|---|---|---|---|---|---|
| CC12M | 10M | 27.8 | 34.0 | 0.306 | 0.268 |
| YFCC15M | 15M | 22.6 | 24.6 | 0.262 | 0.198 |
| RedCaps | 11M | 26.8 | 31.5 | 0.281 | 0.240 |
| Shutterstock | 15M | 21.0 | 28.3 | 0.314 | 0.273 |

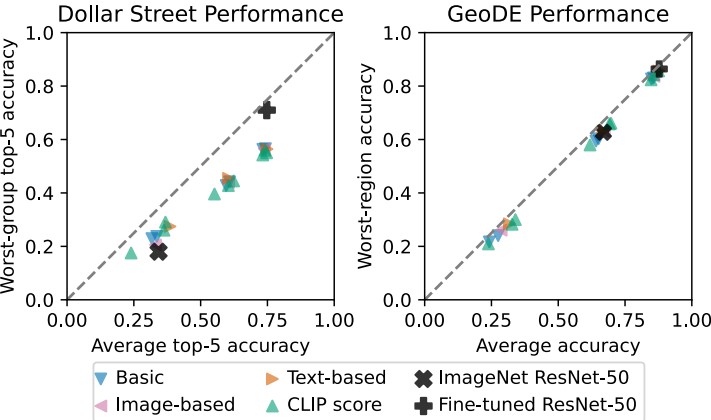

Figure 21: Comparison of average and worst-group scores for Dollar Street and GeoDE diversity datasets. On Dollar Street, our overall higher-performing models display a larger worst-group performance gap (corresponding to lower income households). GeoDE does not show this trend.

ImageNet-Sketch [143] and ObjectNet [13]. Moreover, for each data source, we use OpenAI's pretrained CLIP ViT-B/32 and ViT-L/14 models to compute the cosine similarity between image and text embeddings of a data point, and obtain the average cosine similarity score for the whole dataset.

### P.2.1 Additional results

We present a series of results for the BYOD track in Table 18.

## Q   Fairness and biases

To study the biases displayed by our models, we include two diversity-related datasets, Dollar Street [122] and GeoDE [114], in our evaluation suite, and perform further analysis on the face datasets FairFace [80] and UTKFace [159] with demographic labels, following Radford et al. [111].

### Q.1   Diversity

We break down model performance on the Dollar Street and GeoDE datasets in Figure 21. Dollar Street consists of images of household items taken in homes around the world, and represents a wide socioeconomic range that includes homes with no Internet access [122]. The objects belong to ImageNet categories, and the task is image classification. Standard ImageNet-trained models achieve monotonically increasing performance levels with higher household income levels [122]. Here we use the income-based subgroups defined in Rojas et al. [122], and find a similar bias as discovered in their paper. While our trained models show a smaller worst-group performance gap than an ImageNet-trained ResNet-50, they underperform a model fine-tuned on Dollar Street. Models with higher average accuracy show a larger worst-group gap, which future work should try to address.

GeoDE consists of images of everyday items and objects, which again fall into ImageNet categories. The dataset represents six world regions equally, and primarily aims to promote geographic diversity of datasets [114]. Both ImageNet models and our models show less bias under this distribution compared to Dollar Street, with a smaller worst-group accuracy gap. The trends show that performance across all regions improves steadily with increased scale, and the performance approaches that of a model fine-tuned on GeoDE. While we know that classifiers trained specifically on ImageNet can display geographic biases [114], these biases are not apparent in our GeoDE model evaluations. Future work is needed to investigate the extent to which our models have geographic biases not evaluated in GeoDE.

Table 18: Zero-shot performance for select baselines in the BYOD track. Unless specified otherwise, COMMONPOOL means our pool filtered with CLIP score (L/14, 30%).

| Scale | Data source | Training dataset size | ImageNet | ImageNet dist. shifts | VTAB | Retrieval | Average over 38 datasets |
|---|---|---|---|---|---|---|---|
| | #0 | CC12M | 0.099 | 0.080 | 0.223 | 0.197 | 0.205 |
| | #1 | LAION15M | 0.083 | 0.076 | 0.210 | 0.144 | 0.189 |
| | #2 | RedCaps | 0.076 | 0.066 | 0.177 | 0.141 | 0.168 |
| | #3 | Shutterstock 15M | 0.083 | 0.070 | 0.214 | 0.159 | 0.185 |
| small | #4 | YFCC15M | 0.071 | 0.046 | 0.182 | 0.147 | 0.164 |
| | #5 | #0 + #1 + #2 | 0.097 | 0.084 | 0.208 | 0.161 | 0.195 |
| | #6 | #0 + #1 + #3 | 0.091 | 0.081 | 0.222 | 0.138 | 0.202 |
| | #7 | #0 + #2 + #3 + #4 | 0.095 | 0.075 | 0.205 | 0.164 | 0.186 |
| | #8 | #0–4 | 0.093 | 0.076 | 0.205 | 0.162 | 0.193 |
| | #9 | CC12M | 0.245 | 0.189 | 0.283 | 0.289 | 0.272 |
| | #10 | LAION15M | 0.270 | 0.215 | 0.317 | 0.255 | 0.306 |
| | #11 | RedCaps | 0.237 | 0.166 | 0.271 | 0.178 | 0.263 |
| | #12 | Shutterstock 15M | 0.229 | 0.191 | 0.316 | 0.260 | 0.290 |
| | #13 | YFCC15M | 0.232 | 0.137 | 0.263 | 0.245 | 0.257 |
| | #14 | #9 + #10 + #11 | 0.376 | 0.287 | 0.387 | 0.323 | 0.366 |
| | #15 | #9 + #10 + #12 | 0.342 | 0.278 | 0.362 | 0.345 | 0.357 |
| | #16 | #9 + #11 + #12 + #13 | 0.360 | 0.268 | 0.365 | 0.275 | 0.345 |
| medium | #17 | #9–13 | 0.371 | 0.285 | 0.408 | 0.280 | 0.367 |
| | #18 | Shutterstock illustration | 0.053 | 0.094 | 0.205 | 0.125 | 0.180 |
| | #19 | Shutterstock photo | 0.342 | 0.209 | 0.364 | 0.350 | 0.331 |
| | #20 | Shutterstock vectors | 0.072 | 0.151 | 0.216 | 0.148 | 0.208 |
| | #21 | Shutterstock full | 0.313 | 0.254 | 0.353 | 0.331 | 0.342 |
| | #22 | WIT full | 0.096 | 0.063 | 0.196 | 0.104 | 0.177 |
| | #23 | WIT English | 0.051 | 0.038 | 0.145 | 0.083 | 0.143 |
| | #24 | COYO | 0.272 | 0.235 | 0.301 | 0.254 | 0.304 |
| | #25 | LAION-COCO | 0.209 | 0.205 | 0.293 | 0.359 | 0.297 |
| | #26 | Shutterstock illustration | 0.337 | 0.203 | 0.307 | 0.322 | 0.306 |
| | #27 | Shutterstock photo | 0.485 | 0.304 | 0.432 | 0.427 | 0.398 |
| | #28 | Shutterstock vectors | 0.126 | 0.223 | 0.244 | 0.191 | 0.246 |
| | #29 | Shutterstock full | 0.500 | 0.412 | 0.472 | 0.451 | 0.456 |
| | #30 | COYO | 0.547 | 0.456 | 0.475 | 0.549 | 0.486 |
| | #31 | LAION-COCO | 0.355 | 0.351 | 0.395 | 0.494 | 0.398 |
| | #32 | COYO + LAION-COCO | 0.528 | 0.458 | 0.479 | 0.589 | 0.498 |
| | #33 | LAION-A | 0.611 | 0.474 | 0.501 | 0.542 | 0.505 |
| | #34 | LAION-2B | 0.585 | 0.472 | 0.504 | 0.525 | 0.515 |
| | #35 | COMMONPOOL + #9–13 | 0.602 | 0.498 | 0.541 | 0.416 | 0.537 |
| | #36 | COMMONPOOL + #9–13 (2x upsampled) | 0.613 | 0.507 | 0.559 | 0.433 | 0.543 |
| large | #37 | COMMONPOOL + #9–13 (4x upsampled) | 0.615 | 0.514 | 0.553 | 0.427 | 0.543 |
| | #38 | COMMONPOOL + #9–13 (6x upsampled) | 0.620 | 0.519 | 0.558 | 0.437 | 0.549 |
| | #39 | COMMONPOOL + #9–13 (8x upsampled) | 0.624 | 0.520 | 0.533 | 0.443 | 0.537 |
| | #40 | COMMONPOOL + #9–13 (10x upsampled) | 0.621 | 0.520 | 0.540 | 0.441 | 0.537 |
| | #41 | COMMONPOOL + COYO | 0.561 | 0.472 | 0.504 | 0.508 | 0.513 |
| | #42 | COMMONPOOL + LAION-A | 0.607 | 0.480 | 0.531 | 0.514 | 0.527 |
| | #43 | COMMONPOOL + LAION-COCO | 0.522 | 0.457 | 0.513 | 0.498 | 0.514 |
| | #44 | COMMONPOOL + #9+#11+#13+#19 | 0.609 | 0.508 | 0.546 | 0.439 | 0.536 |
| | #45 | COMMONPOOL + #9+#11+#13+#19 (2x upsampled) | 0.621 | 0.509 | 0.547 | 0.458 | 0.541 |
| | #46 | COMMONPOOL + #9+#11+#13+#19 (4x upsampled) | 0.632 | 0.515 | 0.533 | 0.452 | 0.532 |
| | #47 | COMMONPOOL + #9+#11+#13+#19 (6x upsampled) | 0.635 | 0.515 | 0.535 | 0.471 | 0.532 |
| | #48 | COMMONPOOL + #9+#11+#13+#19 (8x upsampled) | 0.633 | 0.515 | 0.523 | 0.464 | 0.530 |
| | #49 | COMMONPOOL + #9+#11+#13+#19 (10x upsampled) | 0.630 | 0.513 | 0.523 | 0.356 | 0.521 |
| | #50 | LAION-2B | 0.757 | 0.631 | 0.611 | 0.619 | 0.621 |
| xlarge | #51 | COMMONPOOL + #9+#11+#13+#19 | 0.766 | 0.660 | 0.662 | 0.539 | 0.659 |
| | #52 | COMMONPOOL + #9+#11+#13+#19 (6x upsampled) | 0.776 | 0.671 | 0.633 | 0.552 | 0.649 |
| | #53 | COMMONPOOL + #9+#11+#13+#19 (18x upsampled) | 0.771 | 0.667 | 0.629 | 0.554 | 0.643 |

## Q.2 Fairness

Emulating Radford et al. [111], we evaluate our best models from the filtering and BYOD tracks on the human face datasets FairFace and UTKFace, using zero-shot classification to predict the race, gender, and age annotated in these datasets. Following Hanna et al. [59] and Hundt et al. [68], we acknowledge that these evaluations can be problematic as race and gender should not be considered fixed categories, but rather fluid attributes that may change for individuals, based on they way they identify at any given moment—regardless of appearance. We include these evaluations for continuity with prior work and as a probe into model behaviour, but hope future work will consider improved face fairness evaluation. We also note that race, gender, and age classification are not the intended end-goals of the models or benchmark, and we do not condone the use of COMMONPOOL or models trained on COMMONPOOL data for *any* decisions involving people.

Table 19: Overall race, gender, and age classification accuracy of our two best `xlarge` baselines, Image-based ∩ CLIP score (L/14 30%) for the filtering track and COMMONPOOL, CLIP score + 4 external sources (upsampled 6x) for the BYOD track. Race classification was binary (white or non-white) as in Karkkainen & Joo [80].

| Dataset | Track | Race | Gender | Age |
|---|---|---|---|---|
| FairFace | Filtering | 86.4 | 91.7 | 34.3 |
| | BYOD | 76.5 | 93.9 | 33.8 |
| UTKFace | Filtering | 86.2 | 93.8 | 39.5 |
| | BYOD | 86.1 | 95.5 | 38.6 |

Table 20: Gender classification accuracy of our two best `xlarge` baselines, Image-based ∩ CLIP score (L/14 30%) for the filtering track and COMMONPOOL, CLIP score + 4 external sources (upsampled 6x) for the BYOD track.

FairFace

| Track | Gender | Race | | | | | | |
|---|---|---|---|---|---|---|---|---|
| | | Black | White | Indian | Latino/Hispanic | Middle Eastern | Southeast Asian | East Asian |
| Filtering | Male | 79.3 | 91.3 | 90.8 | 90.4 | 95.7 | 83.0 | 80.7 |
| | Female | 95.4 | 96.6 | 94.2 | 96.6 | 96.5 | 97.2 | 98.2 |
| BYOD | Male | 89.2 | 94.8 | 93.2 | 93.4 | 97.4 | 90.2 | 90.6 |
| | Female | 89.2 | 96.0 | 94.2 | 96.0 | 96.2 | 97.1 | 97.0 |

UTKFace

| Track | Gender | Race | | | | |
|---|---|---|---|---|---|---|
| | | Black | White | Indian | Asian | Other |
| Filtering | Male | 95.4 | 92.5 | 91.7 | 73.1 | 84.2 |
| | Female | 97.3 | 98.7 | 97.4 | 98.3 | 97.4 |
| BYOD | Male | 96.8 | 95.9 | 94.7 | 85.7 | 90.4 |
| | Female | 96.3 | 97.7 | 96.8 | 95.9 | 95.6 |

As described in Appendix G, our filleting track models are trained on images with faces blurred. Nevertheless, these models still perform significantly above random chance on face classification. We hypothesize that this is due to a combination of faces bypassing our face blurring filter in the training data, contextual clues outside of the face region, or signal associated with skin color. The BYOD track model performs even better than the filtering track model. We hypothesize that this is because BYOD data is used off-the-shelf and hence contains non-blurred faces. In Table 19, we present overall accuracy for these three traits. Note that race is treated as a binary variable (white or non-white) to enable comparison to prior results, gender is a binary variable (male or female) according to annotations, and age is binned into 9 ranges according to the annotation precision of FairFace. The BYOD model, performs better at distinguishing the annotated gender, but is worse at distinguishing annotated race and age.

We further break down these statistics over the intersection of race and gender, examining gender classification accuracies in Table 20. We find that there are drastic differences in accuracy across different annotated subgroups, varying by both race and gender. The filtering models shows a tendency to misclassify Black, Southeast Asian, and East Asian males as females at 20.7%, 17%, and 19.3% respectively on FairFace. Furthermore, we find that while the BYOD model improves accuracy, on FairFace most of this improvement is on men (ranging from 1.7pp gain to 9.9pp gain), while on women, BYOD offers little change (ranging from 0.6pp gain to 6.2pp drop).

Following Radford et al. [111], we also examined associations of particular demographics with potentially harmful language. We replicate their setup with two classification tasks: (1) including race-gender intersection classes (e.g. "black woman", "indian man", etc.) and several harmful crime-related terms ("thief", "criminal", "suspicious person"); (2) including the same race-gender intersection classes and non-human terms ("animal", "gorilla", "chimpanzee", "orangutan"). We compute the frequency of misclassification of people into one of the harmful categories and run these experiments on FairFace and UTKFace separately. The results are shown in Table 21. Unlike in Radford et al. [111], we find that our models have a very small probability of classifying human faces as non-human, with a max score across all subgroups of 0.1%. However, a significant proportion of people are misclassified as criminal. This again highlights the importance of dataset curation and the risks associated with zero-shot classification on models trained on web-scraped datasets.

Table 21: Harmful misclassification rates of our two best `xlarge` baselines, Image-based ∩ CLIP score (L/14 30%) for the filtering track and COMMONPOOL, CLIP score + 4 external sources (upsampled 6x) for the BYOD track. While very few samples are misclassified as non-human, the filter track model assigns a crime-related label to a significant portion of people, and this is exacerbated by the BYOD model in many cases.

### FairFace

| Track | | Black | White | Indian | Latino/Hispanic | Race Middle Eastern | Southeast Asian | East Asian |
|---|---|---|---|---|---|---|---|---|
| Filtering | Crime-related | 4.4 | 24.3 | 8.8 | 14.3 | 23.7 | 7.4 | 8.6 |
| | Non-human | 0.0 | 0.0 | 0.0 | 0.0 | 0.0 | 0.0 | 0.0 |
| BYOD | Crime-related | 18.4 | 16.8 | 21.5 | 22.9 | 20.9 | 35.3 | 30.9 |
| | Non-human | 0.0 | 0.1 | 0.0 | 0.1 | 0.0 | 0.1 | 0.1 |

### UTKFace

| Track | | Black | White | Race Indian | Asian | Other |
|---|---|---|---|---|---|---|
| Filtering | Crime-related | 6.8 | 16.1 | 9.1 | 6.9 | 13.9 |
| | Non-human | 0.0 | 0.2 | 0.0 | 0.1 | 0.0 |
| BYOD | Crime-related | 12.8 | 10.8 | 15.2 | 13.2 | 18.6 |
| | Non-human | 0.0 | 0.2 | 0.0 | 0.0 | 0.0 |

# R  Extra figures and tables

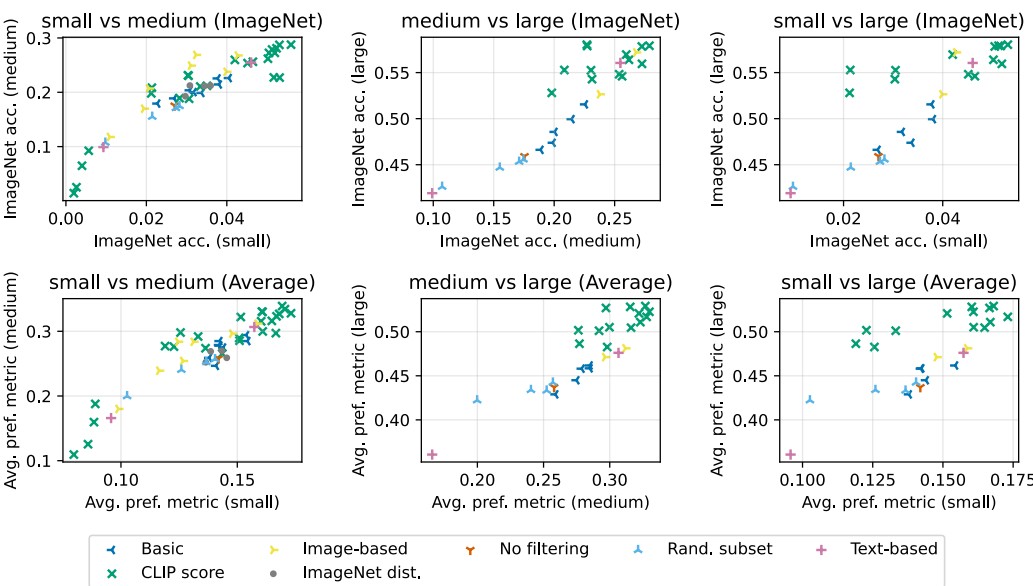

Figure 22: Improving downstream performance at smaller scales correlates positively with performance gains at larger scales. These trends suggests that dataset filtering can be studied effectively at smaller scales, even with less computational resources.

Table 22: Rank correlation between the performance obtained with various filtering strategies at two different scales. Our experimental suggest that the ranking is relatively consistent between scales, especially for the adjacent scale pairs.

| Metric | small vs medium | small vs large | medium vs large |
|---|---|---|---|
| ImageNet acc. | 0.895 | 0.811 | 0.847 |
| Average pref. metric | 0.854 | 0.708 | 0.876 |

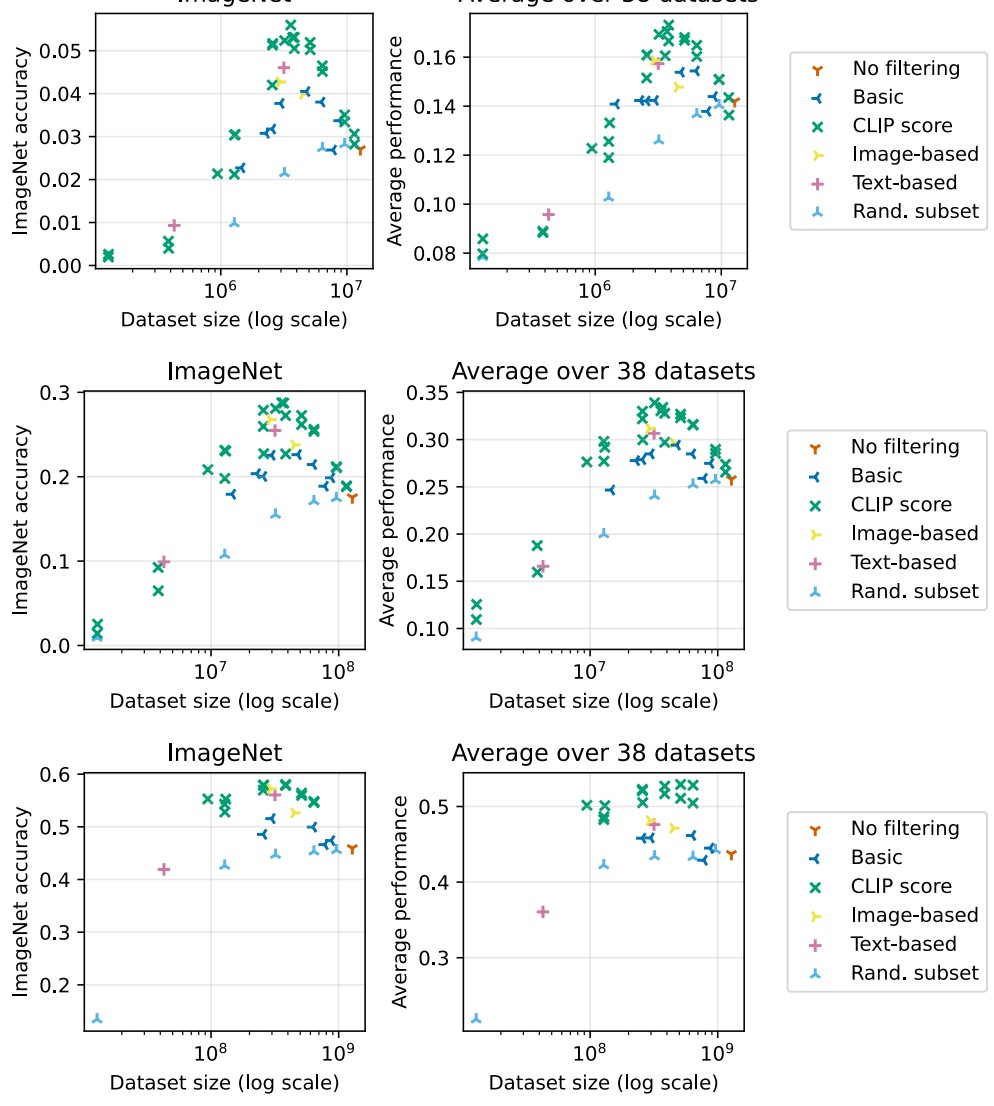

Figure 23: Performance as a function of the number of training samples from the `small` (top), `medium` (middle), and `large` (bottom) scales. There is a significant variance in accuracy even when accounting for the size of the training set.

Table 23: Comparison of ViT-B/32 and ViT-B/16 models across different training datasets.

| Model | Training Dataset | Training dataset size | Training steps | ImageNet | ImageNet dist. shifts | VTAB | Retrieval | Average over 38 datasets |
|---|---|---|---|---|---|---|---|---|
| ViT B/32 | DATACOMP-1B | 1.4B | 13B | 0.692 | 0.551 | 0.577 | 0.538 | 0.579 |
| ViT B/32 | OpenAI's WIT | 0.4B | 13B | 0.633 | 0.485 | 0.526 | 0.501 | 0.525 |
| ViT B/32 | LAION-2B | 2.3B | 34B | 0.666 | 0.522 | 0.561 | 0.560 | 0.569 |
| ViT B/16 | DATACOMP-1B | 1.4B | 13B | 0.735 | 0.608 | 0.621 | 0.578 | 0.615 |
| ViT B/16 | OpenAI's WIT | 0.4B | 13B | 0.683 | 0.559 | 0.546 | 0.527 | 0.563 |
| ViT B/16 | LAION-2B | 2.3B | 34B | 0.702 | 0.566 | 0.572 | 0.583 | 0.587 |

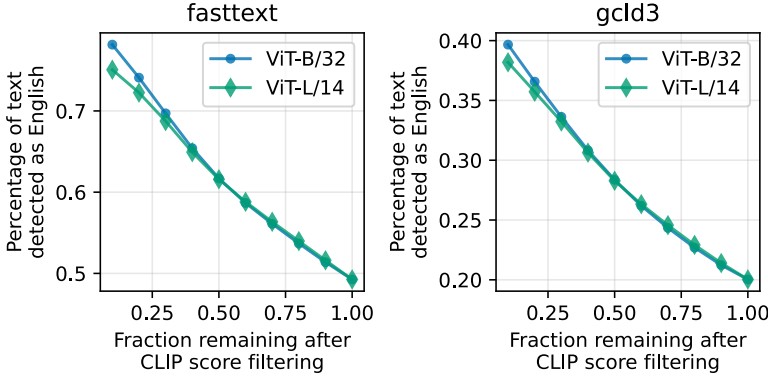

Figure 24: We examine the percentage of texts classified as English after taking the top fraction (on the x-axis) of the `large` billion pool as sorted by CLIP similarity score. We see that doing CLIP filtering implicitly does some English filtering, as image-text pairs with a higher CLIP score are more frequently classified as English.

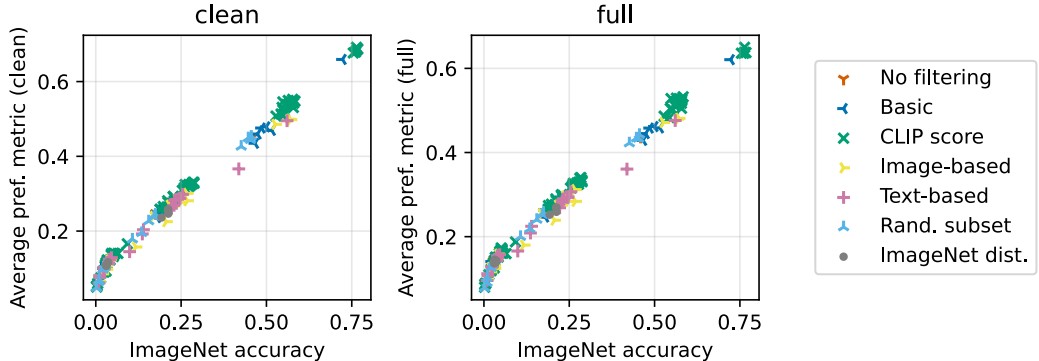

Figure 25: Correlation between ImageNet accuracy and average performance on our suite of evaluation tasks. While ImageNet accuracy strongly correlates with the average performance (both on the clean subset and the full suite), the same is not true for all individual datasets we study, as shown in Appendix R.

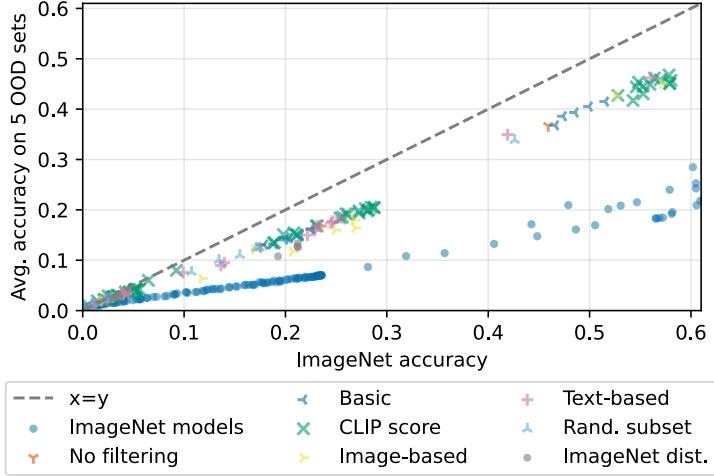

Figure 26: Zero-shot CLIP models trained with various filtering strategies form a reliable trend relating accuracy on ImageNet and related distribution shifts, exhibiting higher effective robustness when compared to ImageNet-trained models from Taori et al. [139].

Table 24: Comparison at the xlarge scale between a 400M subset of CommonPool and OpenAI's WIT which also contains 400M samples. Our 400M subset is created by intersecting IN1k image clustering with English cld3 filtering, then taking the top 400M samples sorted by CLIP L14 score. Our model does better across the various evaluation groupings.

| Model | Training Dataset | Training dataset size | Training steps | ImageNet | ImageNet dist. shifts | VTAB | Retrieval | Average over 38 datasets |
|-------|------------------|-----------------------|----------------|----------|-----------------------|------|-----------|--------------------------|
| ViT L/14 | top 400M by CLIP L14 of Image-based ∩ cld3 | 400M | 13B | 0.763 | 0.657 | 0.641 | 0.595 | 0.638 |
| ViT L/14 | OpenAI's WIT | 400M | 13B | 0.755 | 0.649 | 0.586 | 0.543 | 0.617 |

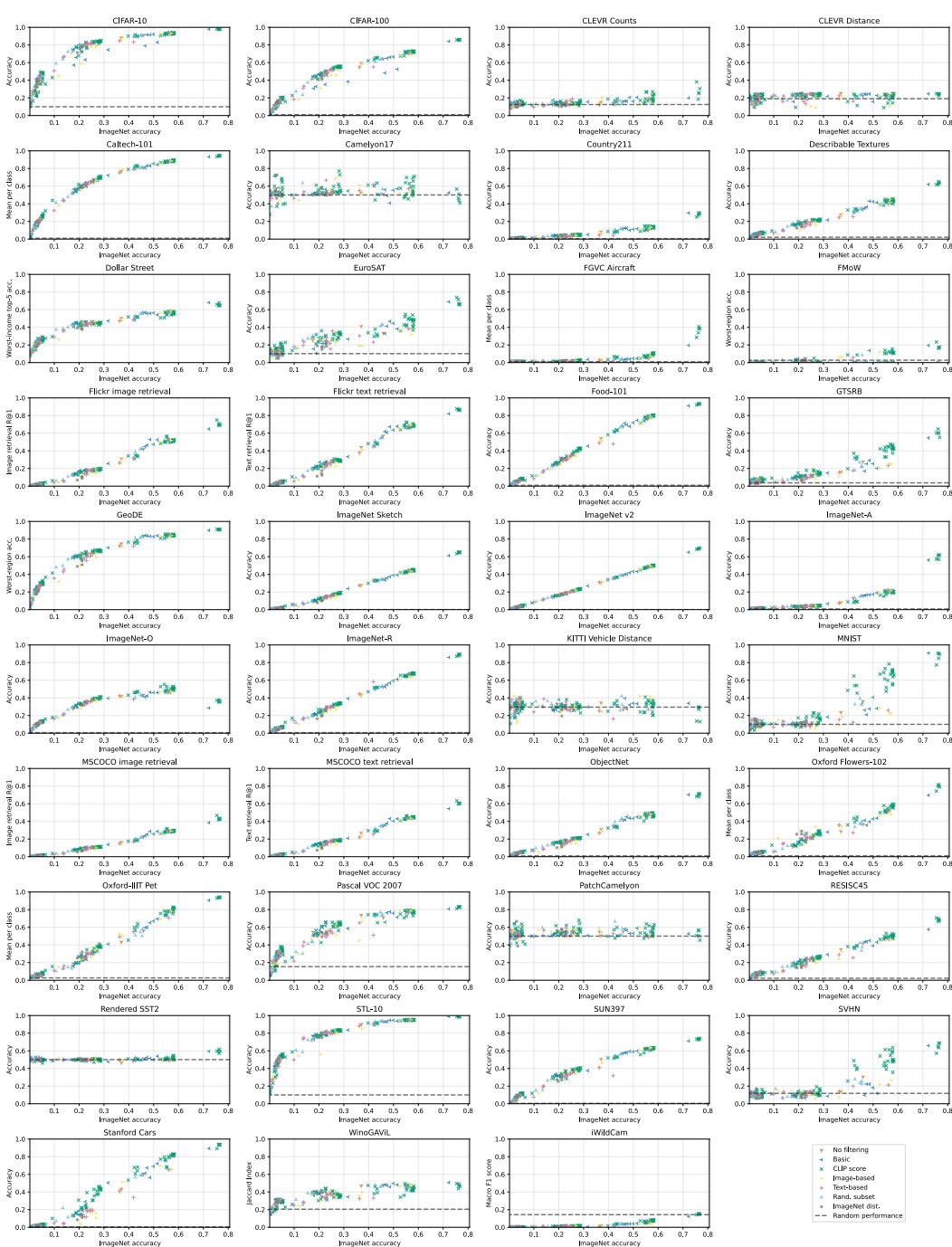

Figure 27: Zero-shot performance on other datasets is often positively correlated with that on ImageNet, but not always. In cases where ImageNet shows close to zero correlation with other datasets, performance on that dataset is often close to random chance.

Table 25: Baseline results for the filtering track, `small` scale.

| Filtering | Training dataset size | ImageNet | ImageNet dist. shifts | VTAB | Retrieval | Average over 38 datasets |
|---|---|---|---|---|---|---|
| No filtering | 12.8M | 0.025 | 0.033 | 0.145 | 0.114 | 0.133 |
| Random subset (75%) | 9.6M | 0.028 | 0.037 | 0.153 | 0.110 | 0.140 |
| Random subset (50%) | 6.4M | 0.027 | 0.037 | 0.147 | 0.111 | 0.137 |
| Random subset (25%) | 3.2M | 0.022 | 0.032 | 0.130 | 0.099 | 0.126 |
| Random subset (10%) | 1.3M | 0.010 | 0.018 | 0.116 | 0.077 | 0.103 |
| Random subset (1%) | 128K | 0.002 | 0.005 | 0.095 | 0.049 | 0.078 |
| Caption length | 8.7M | 0.034 | 0.040 | 0.148 | 0.109 | 0.143 |
| Image size | 7.8M | 0.027 | 0.036 | 0.154 | 0.119 | 0.138 |
| English (fasttext) | 6.3M | 0.038 | 0.045 | 0.164 | 0.124 | 0.154 |
| English (fasttext) and caption length | 4.8M | 0.041 | 0.048 | 0.159 | 0.123 | 0.154 |
| English (fasttext), caption length, and image size | 3.0M | 0.038 | 0.043 | 0.150 | 0.118 | 0.142 |
| English (cld3) | 2.6M | 0.032 | 0.039 | 0.143 | 0.111 | 0.142 |
| English (cld3) and caption length | 2.3M | 0.031 | 0.038 | 0.153 | 0.111 | 0.142 |
| English (cld3), caption length, and image size | 1.5M | 0.023 | 0.030 | 0.154 | 0.092 | 0.141 |
| CLIP B32 score top 1% | 129K | 0.003 | 0.007 | 0.114 | 0.050 | 0.086 |
| CLIP B32 score top 3% | 384K | 0.006 | 0.014 | 0.104 | 0.055 | 0.089 |
| CLIP B32 score top 10% | 1.3M | 0.026 | 0.035 | 0.147 | 0.083 | 0.126 |
| CLIP B32 score top 20% | 2.6M | 0.051 | 0.056 | 0.173 | 0.114 | 0.161 |
| CLIP B32 score top 30% | 3.8M | 0.045 | 0.052 | 0.180 | 0.120 | 0.167 |
| CLIP B32 score top 40% | 5.1M | 0.052 | 0.057 | 0.173 | 0.123 | 0.167 |
| CLIP B32 score top 50% | 6.4M | 0.047 | 0.053 | 0.174 | 0.124 | 0.165 |
| CLIP B32 score top 75% | 9.6M | 0.033 | 0.043 | 0.161 | 0.121 | 0.151 |
| CLIP B32 score top 90% | 11.5M | 0.028 | 0.039 | 0.140 | 0.114 | 0.136 |
| CLIP B32 threshold at 0.3 + English filter | 942K | 0.022 | 0.032 | 0.138 | 0.077 | 0.122 |
| CLIP B32 threshold at 0.28 + English filter | 1.3M | 0.031 | 0.040 | 0.136 | 0.092 | 0.133 |
| CLIP B32 threshold at 0.3 | 2.6M | 0.052 | 0.056 | 0.166 | 0.114 | 0.161 |
| CLIP B32 score 1% to 30% | 3.7M | 0.053 | 0.058 | 0.185 | 0.113 | 0.170 |
| CLIP B32 score 2% to 30% | 3.6M | 0.056 | 0.059 | 0.173 | 0.120 | 0.161 |
| CLIP B32 score 5% to 30% | 3.2M | 0.052 | 0.055 | 0.177 | 0.115 | 0.169 |
| CLIP L14 score top 1% | 128K | 0.002 | 0.007 | 0.111 | 0.050 | 0.080 |
| CLIP L14 score top 3% | 386K | 0.004 | 0.009 | 0.110 | 0.052 | 0.088 |
| CLIP L14 score top 10% | 1.3M | 0.021 | 0.033 | 0.131 | 0.075 | 0.119 |
| CLIP L14 score top 20% | 2.6M | 0.042 | 0.051 | 0.165 | 0.100 | 0.151 |
| CLIP L14 score top 30% | 3.8M | 0.051 | 0.055 | 0.190 | 0.119 | 0.173 |
| CLIP L14 score top 40% | 5.1M | 0.050 | 0.054 | 0.173 | 0.119 | 0.168 |
| CLIP L14 score top 50% | 6.4M | 0.045 | 0.052 | 0.164 | 0.122 | 0.160 |
| CLIP L14 score top 75% | 9.6M | 0.035 | 0.043 | 0.164 | 0.120 | 0.151 |
| CLIP L14 score top 90% | 11.5M | 0.031 | 0.038 | 0.154 | 0.116 | 0.144 |
| Image-based clustering (ImageNet1k) | 2.9M | 0.043 | 0.047 | 0.178 | 0.121 | 0.159 |
| Image-based clustering (ImageNet21k) | 4.5M | 0.035 | 0.045 | 0.154 | 0.122 | 0.148 |
| Image-based sampling, $\alpha$=0 | 12.8M | 0.019 | 0.030 | 0.144 | 0.095 | 0.127 |
| Image-based sampling, $\alpha$=0.2 | 12.8M | 0.031 | 0.036 | 0.133 | 0.100 | 0.131 |
| Image-based sampling, $\alpha$=0.5 | 12.8M | 0.032 | 0.038 | 0.129 | 0.096 | 0.125 |
| Image-based sampling, $\alpha$=1 | 12.8M | 0.021 | 0.028 | 0.128 | 0.078 | 0.116 |
| Image-based sampling, $\alpha$=2 | 12.8M | 0.011 | 0.017 | 0.116 | 0.065 | 0.099 |
| ImageNet distance (L14, top 30%) and English | 2.0M | 0.031 | 0.039 | 0.163 | 0.103 | 0.145 |
| ImageNet distance (L14, top 20%) | 2.6M | 0.030 | 0.035 | 0.155 | 0.102 | 0.136 |
| ImageNet distance (L14, top 30%) | 3.9M | 0.034 | 0.041 | 0.151 | 0.106 | 0.139 |
| ImageNet distance (L14, top 40%) | 5.1M | 0.036 | 0.040 | 0.151 | 0.118 | 0.143 |
| Text-based clustering (ImageNet1k) | 427K | 0.009 | 0.016 | 0.120 | 0.056 | 0.096 |
| Text-based clustering (ImageNet21k) | 3.2M | 0.046 | 0.052 | 0.169 | 0.125 | 0.157 |
| Text-based sampling with average score, $\alpha$=0 | 12.8M | 0.011 | 0.020 | 0.128 | 0.079 | 0.112 |
| Text-based sampling with average score, $\alpha$=0.5 | 12.8M | 0.023 | 0.035 | 0.127 | 0.092 | 0.128 |
| Text-based sampling with average score, $\alpha$=1 | 12.8M | 0.040 | 0.044 | 0.163 | 0.115 | 0.155 |
| Text-based sampling with average score, $\alpha$=1.2 | 12.8M | 0.038 | 0.045 | 0.150 | 0.112 | 0.143 |
| Text-based sampling with max score, $\alpha$=0 | 12.8M | 0.012 | 0.020 | 0.126 | 0.074 | 0.107 |
| Text-based sampling with max score, $\alpha$=0.5 | 12.8M | 0.025 | 0.033 | 0.134 | 0.093 | 0.129 |
| Text-based sampling with max score, $\alpha$=1 | 12.8M | 0.040 | 0.046 | 0.159 | 0.116 | 0.150 |
| Text-based sampling with max score, $\alpha$=1.2 | 12.8M | 0.040 | 0.050 | 0.161 | 0.113 | 0.152 |
| Intersect IN1k image clustering and CLIP B32 score top 30% | 1.4M | 0.049 | 0.053 | 0.150 | 0.103 | 0.148 |
| Intersect IN1k image clustering and CLIP L14 score top 30% | 1.4M | 0.039 | 0.045 | 0.162 | 0.094 | 0.145 |
| Intersect IN21k image clustering and CLIP B32 score top 30% | 2.1M | 0.052 | 0.057 | 0.179 | 0.112 | 0.167 |
| Intersect IN21k image clustering and CLIP L14 score top 30% | 2.1M | 0.047 | 0.053 | 0.176 | 0.110 | 0.163 |

Table 26: Baseline results for the filtering track, `medium` scale.

| Filtering | Training dataset size | ImageNet | ImageNet dist. shifts | VTAB | Retrieval | Average over 38 datasets |
|---|---|---|---|---|---|---|
| No filtering | 128M | 0.176 | 0.152 | 0.259 | 0.219 | 0.258 |
| Random subset (75%) | 96.0M | 0.175 | 0.154 | 0.265 | 0.219 | 0.257 |
| Random subset (50%) | 64.0M | 0.171 | 0.151 | 0.258 | 0.216 | 0.252 |
| Random subset (25%) | 32.0M | 0.155 | 0.136 | 0.246 | 0.203 | 0.240 |
| Random subset (10%) | 12.8M | 0.107 | 0.095 | 0.210 | 0.144 | 0.200 |
| Random subset (1%) | 1.3M | 0.009 | 0.017 | 0.102 | 0.065 | 0.090 |
| Caption length | 87.5M | 0.199 | 0.172 | 0.275 | 0.236 | 0.275 |
| Image size | 77.8M | 0.189 | 0.163 | 0.248 | 0.231 | 0.259 |
| English (fasttext) | 63.0M | 0.214 | 0.182 | 0.290 | 0.246 | 0.285 |
| English (fasttext) and caption length | 47.8M | 0.226 | 0.193 | 0.284 | 0.251 | 0.285 |
| English (fasttext), caption length, and image size | 29.8M | 0.226 | 0.193 | 0.297 | 0.253 | 0.294 |
| English (cld3) | 25.6M | 0.200 | 0.175 | 0.296 | 0.235 | 0.279 |
| English (cld3) and caption length | 22.9M | 0.204 | 0.175 | 0.287 | 0.243 | 0.278 |
| English (cld3), caption length, and image size | 14.6M | 0.179 | 0.159 | 0.243 | 0.216 | 0.247 |
| CLIP B32 score top 1% | 1.3M | 0.025 | 0.037 | 0.140 | 0.076 | 0.126 |
| CLIP B32 score top 3% | 3.9M | 0.093 | 0.096 | 0.205 | 0.128 | 0.188 |
| CLIP B32 score top 10% | 12.8M | 0.231 | 0.199 | 0.305 | 0.206 | 0.298 |
| CLIP B32 score top 20% | 25.7M | 0.279 | 0.234 | 0.337 | 0.241 | 0.330 |
| CLIP B32 score top 30% | 38.4M | 0.285 | 0.240 | 0.355 | 0.253 | 0.338 |
| CLIP B32 score top 40% | 51.3M | 0.273 | 0.227 | 0.333 | 0.257 | 0.324 |
| CLIP B32 score top 50% | 64.0M | 0.256 | 0.219 | 0.322 | 0.259 | 0.316 |
| CLIP B32 score top 75% | 96.1M | 0.211 | 0.180 | 0.301 | 0.238 | 0.290 |
| CLIP B32 score top 90% | 115M | 0.189 | 0.165 | 0.279 | 0.229 | 0.274 |
| CLIP B32 threshold at 0.3 + English filter | 9.4M | 0.208 | 0.184 | 0.292 | 0.210 | 0.276 |
| CLIP B32 threshold at 0.28 + English filter | 13.0M | 0.230 | 0.198 | 0.307 | 0.233 | 0.292 |
| CLIP B32 threshold at 0.3 | 25.9M | 0.282 | 0.233 | 0.340 | 0.243 | 0.333 |
| CLIP B32 score 1% to 30% | 37.1M | 0.287 | 0.238 | 0.347 | 0.253 | 0.334 |
| CLIP B32 score 2% to 30% | 35.9M | 0.288 | 0.238 | 0.338 | 0.248 | 0.330 |
| CLIP B32 score 5% to 30% | 32.0M | 0.281 | 0.230 | 0.352 | 0.254 | 0.339 |
| CLIP L14 score top 1% | 1.3M | 0.014 | 0.025 | 0.136 | 0.062 | 0.109 |
| CLIP L14 score top 3% | 3.9M | 0.065 | 0.077 | 0.176 | 0.103 | 0.160 |
| CLIP L14 score top 10% | 12.8M | 0.198 | 0.183 | 0.283 | 0.188 | 0.277 |
| CLIP L14 score top 20% | 25.7M | 0.260 | 0.225 | 0.326 | 0.235 | 0.322 |
| CLIP L14 score top 30% | 38.4M | 0.273 | 0.230 | 0.338 | 0.251 | 0.328 |
| CLIP L14 score top 40% | 51.2M | 0.262 | 0.226 | 0.330 | 0.260 | 0.327 |
| CLIP L14 score top 50% | 64.1M | 0.254 | 0.218 | 0.322 | 0.262 | 0.315 |
| CLIP L14 score top 75% | 96.1M | 0.212 | 0.180 | 0.287 | 0.242 | 0.285 |
| CLIP L14 score top 90% | 115M | 0.188 | 0.164 | 0.258 | 0.225 | 0.266 |
| Image-based clustering (ImageNet1k) | 29.2M | 0.268 | 0.213 | 0.319 | 0.256 | 0.312 |
| Image-based clustering (ImageNet21k) | 45.1M | 0.238 | 0.198 | 0.304 | 0.252 | 0.312 |
| Image-based sampling, $\alpha$=0 | 128M | 0.170 | 0.150 | 0.266 | 0.209 | 0.254 |
| Image-based sampling, $\alpha$=0.2 | 128M | 0.249 | 0.193 | 0.292 | 0.221 | 0.284 |
| Image-based sampling, $\alpha$=0.5 | 128M | 0.269 | 0.196 | 0.301 | 0.216 | 0.284 |
| Image-based sampling, $\alpha$=1 | 128M | 0.207 | 0.145 | 0.264 | 0.166 | 0.239 |
| Image-based sampling, $\alpha$=2 | 128M | 0.118 | 0.082 | 0.207 | 0.110 | 0.180 |
| ImageNet distance (L14, top 30%) and English | 19.8M | 0.212 | 0.158 | 0.272 | 0.178 | 0.259 |
| ImageNet distance (L/14, top 20%) | 25.8M | 0.193 | 0.138 | 0.276 | 0.176 | 0.252 |
| ImageNet distance (L/14, top 30%) | 38.5M | 0.212 | 0.159 | 0.283 | 0.201 | 0.269 |
| ImageNet distance (L/14, top 40%) | 51.3M | 0.212 | 0.165 | 0.273 | 0.212 | 0.270 |
| Text-based clustering (ImageNet1k) | 4.3M | 0.099 | 0.090 | 0.173 | 0.109 | 0.166 |
| Text-based clustering (ImageNet21k) | 31.7M | 0.255 | 0.215 | 0.328 | 0.249 | 0.307 |
| Text-based sampling with average score, $\alpha$=0 | 128M | 0.136 | 0.110 | 0.213 | 0.140 | 0.209 |
| Text-based sampling with average score, $\alpha$=0.5 | 128M | 0.222 | 0.178 | 0.273 | 0.206 | 0.269 |
| Text-based sampling with average score, $\alpha$=1 | 128M | 0.245 | 0.204 | 0.302 | 0.251 | 0.293 |
| Text-based sampling with average score, $\alpha$=1.2 | 128M | 0.231 | 0.200 | 0.298 | 0.240 | 0.289 |
| Text-based sampling with max score, $\alpha$=0 | 128M | 0.140 | 0.116 | 0.242 | 0.138 | 0.225 |
| Text-based sampling with max score, $\alpha$=0.5 | 128M | 0.229 | 0.190 | 0.290 | 0.205 | 0.283 |
| Text-based sampling with max score, $\alpha$=1 | 128M | 0.247 | 0.209 | 0.300 | 0.241 | 0.295 |
| Text-based sampling with max score, $\alpha$=1.2 | 128M | 0.235 | 0.200 | 0.298 | 0.239 | 0.290 |
| Intersect IN1k image clustering and CLIP B32 score top 30% | 14.2M | 0.305 | 0.243 | 0.342 | 0.250 | 0.328 |
| Intersect IN1k image clustering and CLIP L14 score top 30% | 14.0M | 0.297 | 0.239 | 0.346 | 0.231 | 0.328 |
| Intersect IN21k image clustering and CLIP B32 score top 30% | 21.1M | 0.298 | 0.244 | 0.347 | 0.256 | 0.336 |
| Intersect IN21k image clustering and CLIP L14 score top 30% | 20.8M | 0.290 | 0.241 | 0.339 | 0.244 | 0.328 |

Table 27: Baseline results for the filtering track, `large` scale.

| Filtering | Training dataset size | ImageNet | ImageNet dist. shifts | VTAB | Retrieval | Average over 38 datasets |
|---|---|---|---|---|---|---|
| No filtering | 1.28B | 0.459 | 0.378 | 0.426 | 0.419 | 0.437 |
| Random subset (75%) | 960M | 0.456 | 0.379 | 0.435 | 0.415 | 0.442 |
| Random subset (50%) | 640M | 0.453 | 0.377 | 0.427 | 0.413 | 0.433 |
| Random subset (25%) | 320M | 0.447 | 0.373 | 0.424 | 0.407 | 0.434 |
| Random subset (10%) | 128M | 0.426 | 0.350 | 0.417 | 0.396 | 0.442 |
| Random subset (1%) | 12.8M | 0.135 | 0.118 | 0.219 | 0.135 | 0.218 |
| Caption length | 874M | 0.474 | 0.392 | 0.438 | 0.443 | 0.445 |
| Image size | 777M | 0.466 | 0.375 | 0.421 | 0.438 | 0.429 |
| English (fasttext) | 630M | 0.500 | 0.414 | 0.449 | 0.460 | 0.462 |
| English (fasttext), caption length, and image size | 298M | 0.516 | 0.423 | 0.446 | 0.480 | 0.458 |
| English (cld3) | 256M | 0.486 | 0.405 | 0.462 | 0.472 | 0.458 |
| CLIP B32 score top 10% | 128M | 0.543 | 0.440 | 0.471 | 0.435 | 0.483 |
| CLIP B32 score top 20% | 257M | 0.578 | 0.465 | 0.516 | 0.463 | 0.515 |
| CLIP B32 score top 30% | 384M | 0.578 | 0.466 | 0.525 | 0.475 | 0.527 |
| CLIP B32 score top 40% | 512M | 0.560 | 0.454 | 0.512 | 0.478 | 0.511 |
| CLIP B32 score top 50% | 640M | 0.546 | 0.450 | 0.504 | 0.484 | 0.505 |
| CLIP B32 threshold at 0.3 + English filter | 94.3M | 0.553 | 0.447 | 0.511 | 0.482 | 0.502 |
| CLIP B32 threshold at 0.28 + English filter | 130M | 0.553 | 0.453 | 0.510 | 0.495 | 0.501 |
| CLIP B32 threshold at 0.3 | 258M | 0.579 | 0.464 | 0.501 | 0.465 | 0.505 |
| CLIP L14 score top 10% | 128M | 0.528 | 0.444 | 0.482 | 0.413 | 0.486 |
| CLIP L14 score top 20% | 257M | 0.570 | 0.466 | 0.524 | 0.455 | 0.521 |
| CLIP L14 score top 30% | 384M | 0.578 | 0.474 | 0.538 | 0.466 | 0.529 |
| CLIP L14 score top 40% | 512M | 0.564 | 0.462 | 0.533 | 0.468 | 0.529 |
| CLIP L14 score top 50% | 641M | 0.548 | 0.455 | 0.539 | 0.469 | 0.528 |
| Image-based clustering (ImageNet1k) | 294M | 0.572 | 0.454 | 0.483 | 0.481 | 0.481 |
| Image-based clustering (ImageNet21k) | 450M | 0.527 | 0.433 | 0.468 | 0.463 | 0.471 |
| Text-based clustering (ImageNet1k) | 42.7M | 0.419 | 0.355 | 0.340 | 0.309 | 0.361 |
| Text-based clustering (ImageNet21k) | 317M | 0.561 | 0.465 | 0.465 | 0.479 | 0.476 |
| Intersect IN1k image clustering and CLIP B32 score top 30% | 143M | 0.632 | 0.498 | 0.525 | 0.504 | 0.528 |
| Intersect IN1k image clustering and CLIP L14 score top 30% | 140M | 0.631 | 0.508 | 0.546 | 0.498 | 0.537 |
| Intersect IN21k image clustering and CLIP B32 score top 30% | 211M | 0.605 | 0.481 | 0.531 | 0.494 | 0.519 |
| Intersect IN21k image clustering and CLIP L14 score top 30% | 208M | 0.506 | 0.416 | 0.466 | 0.424 | 0.471 |

Table 28: Baseline results for the filtering track, `xlarge` scale.

| Filtering | Training dataset size | ImageNet | ImageNet dist. shifts | VTAB | Retrieval | Average over 38 datasets |
|---|---|---|---|---|---|---|
| No filtering | 12.8B | 0.723 | 0.612 | 0.611 | 0.569 | 0.621 |
| CLIP B32 score top 30% | 3.84B | 0.764 | 0.640 | 0.628 | 0.599 | 0.638 |
| CLIP B32 threshold at 0.28 + English filter | 1.3B | 0.755 | 0.637 | 0.624 | 0.620 | 0.636 |
| CLIP L14 score top 20% | 2.56B | 0.761 | 0.649 | 0.630 | 0.575 | 0.636 |
| CLIP L14 score top 25% | 3.2B | 0.768 | 0.656 | 0.621 | 0.585 | 0.637 |
| CLIP L14 score top 30% | 3.84B | 0.764 | 0.655 | 0.643 | 0.588 | 0.650 |
| Intersect IN1k image clustering and CLIP L14 score top 30% | 1.38B | 0.792 | 0.679 | 0.652 | 0.608 | 0.663 |

# S    Datasheet

## S.1    Motivation

Q1 **For what purpose was the dataset created?** Was there a specific task in mind? Was there a specific gap that needed to be filled? Please provide a description.

- The purpose of DATACOMP and the associated COMMONPOOL dataset is to enable study of what makes a strong image-text dataset, which supports a broad range of applications. Prior work mainly focuses on data curation in the context of supervised datasets and smaller scales. For a fuller treatment see Section 2. In our initial release of DATACOMP we focus on 38 downstream image classification and image retrieval tasks. For details see Section 3.5 and Appendix O.

Q2 **Who created the dataset (e.g., which team, research group) and on behalf of which entity (e.g., company, institution, organization)?**

- DATACOMP and COMMONPOOL were created by a group of researchers with the following affiliations, listed in alphabetical order: Allen Institute for Artificial Intelligence (AI2), Apple, Columbia University, Google Research, Graz University of Technology, Hebrew University, Juelich Supercomputing Center, LAION, Research Center Juelich, StabilityAI, Tel Aviv University, University of Illinois Urbana-Champaign, University of Texas at Austin, University of Washington.

Q3 **Who funded the creation of the dataset?** If there is an associated grant, please provide the name of the grantor and the grant name and number.

- Compute for this research was generously provided by StabilityAI. For more specific acknowledgments, see the acknowledgment section at the end of the main paper.

Q4 **Any other comments?**

- We hope that COMMONPOOL will help to facilitate data-centric questions in ML and AI towards the next generation of web-scale datasets, that 1) yield higher accuracy models and 2) models that are safer and more equitable.

## S.2    Composition

Q5 **What do the instances that comprise the dataset represent (e.g., documents, photos, people, countries)?** *Are there multiple types of instances (e.g., movies, users, and ratings; people and interactions between them; nodes and edges)? Please provide a description.*

- Each instance is a pair of url and corresponding image alt-text. The url points to an image that a user can then try to download. Each sample is also tagged with metadata, discussed in Q25.

Q6 **How many instances are there in total (of each type, if appropriate)?**

- There are 12.8B instances in COMMONPOOL. For breakdowns and statistics see Appendix I.

Q7 **Does the dataset contain all possible instances or is it a sample (not necessarily random) of instances from a larger set?** *If the dataset is a sample, then what is the larger set? Is the sample representative of the larger set (e.g., geographic coverage)? If so, please describe how this representativeness was validated/verified. If it is not representative of the larger set, please describe why not (e.g., to cover a more diverse range of instances, because instances were withheld or unavailable).*

- We find ∼88B possible samples in common crawl. These samples are globally shuffled to ensure i.i.d. sampling for all sampling based parts of the downstream pipeline. Of these samples we attempt to download ∼40B samples. Due to various download issues, such as dead links and throttling, we are able to successfully download ∼16.8B samples. After NSFW filtering and evaluation set deduplication we end up with ∼13.1B viable samples, from which we randomly sample 12.8B for COMMONPOOL. For a complete treatment and visualization of our data processing funnel, see Appendix H. For each sample we also release metadata shown in Table 8.

Q8 **What data does each instance consist of?** *"Raw" data (e.g., unprocessed text or images) or features? In either case, please provide a description.*

- Each sample contains an image url for download and an associated alt-text caption. Additionally, each sample contains metadata fields shown in Table 8 (e.g., image aspect ratio and CLIP features).

Q9 **Is there a label or target associated with each instance?** *If so, please provide a description.*

- We do not provide any category labels; however, the text associated with each image can be considered a soft, noisy label for each sample. Such labels are common in modern image-text training paradigms (e.g., image-text representation alignment, image captioning objectives, text-conditional image generation objectives, etc.).

Q10 **Is any information missing from individual instances?** *If so, please provide a description, explaining why this information is missing (e.g., because it was unavailable). This does not include intentionally removed information, but might include, e.g., redacted text.*

- No, each sample is an image-text pair.

Q11 **Are relationships between individual instances made explicit (e.g., users' movie ratings, social network links)?** *If so, please describe how these relationships are made explicit.*

- No, the dataset is released as it is with no explicit attempt to establish relationships between instances.

Q12 **Are there recommended data splits (e.g., training, development/validation, testing)?** *If so, please provide a description of these splits, explaining the rationale behind them.*

- No. The test tasks are existing image classification tasks. We run a deduplication model to try to prevent test set contamination in COMMONPOOL.

Q13 **Are there any errors, sources of noise, or redundancies in the dataset?** *If so, please provide a description.*

- COMMONPOOL is sourced from Common Crawl, which can be thought of as a snapshot of the internet. Hence, there can be considerable noise (e.g., alt-text being unrelated to its associated image), duplicate data, etc.

Q14 **Is the dataset self-contained, or does it link to or otherwise rely on external resources (e.g., websites, tweets, other datasets)?** *If it links to or relies on external resources, a) are there guarantees that they will exist, and remain constant, over time; b) are there official archival versions of the complete dataset (i.e., including the external resources as they existed at the time the dataset was created); c) are there any restrictions (e.g., licenses, fees) associated with any of the external resources that might apply to a future user? Please provide descriptions of all external resources and any restrictions associated with them, as well as links or other access points, as appropriate.*

- The data is not self-contained and rather links other external resources on the internet. Links point to resources distributed across the internet. There is no guarantee that the resources will exist in perpetuity or that that the resources will not change. To mitigate against data poisoning in future COMMONPOOL downloads, we release SHA256 hashes of images. Due to the size of the dataset, it is not possible to provide it in an archival form.

Q15 **Does the dataset contain data that might be considered confidential (e.g., data that is protected by legal privilege or by doctor–patient confidentiality, data that includes the content of individuals' non-public communications)?** *If so, please provide a description.*

- The dataset is comprised of data that was readily available on the public internet at the time of our download. However, it is possible that the dataset contains confidential information (e.g., private data that is hosted publicly for nefarious reasons or out of ignorance of said data being confidential).

Q16 **Does the dataset contain data that, if viewed directly, might be offensive, insulting, threatening, or might otherwise cause anxiety?** *If so, please describe why.*

- Considering the plurality of people and their backgrounds across the world, it is highly likely that there is content in COMMONPOOL that may upset people. Common Crawl scrapes the internet, which has pornographic, hateful, racist, sexist, and otherwise abhorrent and toxic material. While we attempt to do thorough NSFW filtering, these methods are not 100% accurate. At the 12.8B scale at which we operate, it is highly likely that there is still toxic content in the dataset. We consider the dataset as a research artifact and hope future work will look critically at COMMONPOOL in the hopes of developing even better safety filters.

Q17 **Does the dataset relate to people?** *If not, you may skip the remaining questions in this section.*

- People may appear in the dataset; however, in an effort to preserve privacy, our downloading tooling automatically blurs all detected faces in COMMONPOOL images.

Q18 **Does the dataset identify any subpopulations (e.g., by age, gender)?**

- While COMMONPOOL does not explicitly identify subpopulations in its metadata, it is plausible to extract such information for some images using the corresponding textual caption.

Q19 **Is it possible to identify individuals (i.e., one or more natural persons), either directly or indirectly (i.e., in combination with other data) from the dataset?** *If so, please describe how.*

- We conjecture that even with our face blurring procedure, it may still be possible to identify individuals. Face blurring relies of a face detection model, which could fail (See Appendix G for experimental validation of the employed detector). It is also possible to identify certain celebrities or athletes, who may wear distinctive clothing that is associated with them. It is also likely that names are contained in textual captions, though it is not guaranteed that these names correspond to people in images due to the inherent noisiness of internet captions.

Q20 **Does the dataset contain data that might be considered sensitive in any way (e.g., data that reveals racial or ethnic origins, sexual orientations, religious beliefs, political opinions or union memberships, or locations; financial or health data; biometric or genetic data; forms of government identification, such as social security numbers; criminal history)?** *If so, please provide a description.*

- Yes. COMMONPOOL is created using images and corresponding alt-text that are available on the public internet. Given the 12.8B scale of COMMONPOOL, it is highly likely that there is sensitive data in the dataset. To mitigate against making sensitive content more accessible, we 1) run NSFW image filtering and 2) NSFW text filtering when generating COMMONPOOL, discarding all samples that are flagged. Additionally we 3) provide automatic face blurring in our COMMONPOOL download scripts to blur all detected faces.

Q21 **Any other comments?**

- COMMONPOOL is a research artifact, and we hope it will be useful for those studying how to make internet-scale datasets safer.

## S.3 Collection Process

Q22 **How was the data associated with each instance acquired?** *Was the data directly observable (e.g., raw text, movie ratings), reported by subjects (e.g., survey responses), or indirectly inferred/derived from other data (e.g., part-of-speech tags, model-based guesses for age or language)? If data was reported by subjects or indirectly inferred/derived from other data, was the data validated/verified? If so, please describe how.*

- Data is directly downloaded from the public internet.

Q23 **What mechanisms or procedures were used to collect the data (e.g., hardware apparatus or sensor, manual human curation, software program, software API)?** *How were these mechanisms or procedures validated?*

- We iterate on the LAION-5B data collection process, making an effort to emphasize safety. We ran python based processing scripts to parse Common Crawl dumps, download images, filter our NSFW content, deduplicate samples against downstream tests sets, blur faces, and compute CLIP features. We ran processes on 100s of AWS CPU nodes for Common Crawl parsing and data download. Other steps were run on one of StabilityAI's GPU cluster. For software links see Q37. For software validation related to NSFW content filtering and face blurring see Appendices E and G respectively. In brief, for NSFW image filtering, we validate against commercial APIs and on the NSFW test set introduced in LAION-5B. For face detection (used for face blurring), we evaluate against commercial APIs. We find strong performance for both modules.

Q24 **If the dataset is a sample from a larger set, what was the sampling strategy (e.g., deterministic, probabilistic with specific sampling probabilities)?**

- See Q7.

Q25 **Who was involved in the data collection process (e.g., students, crowdworkers, contractors) and how were they compensated (e.g., how much were crowdworkers paid)?**

- The researching authors were involved in the data collection as an open source effort. No researchers were compensated specifically for their involvement in this project.

Q26 **Over what timeframe was the data collected? Does this timeframe match the creation timeframe of the data associated with the instances (e.g., recent crawl of old news articles)?** *If not, please describe the timeframe in which the data associated with the instances was created.*

- Data was downloaded between December 2022 and March 2023. The urls are collected from Common Crawl dumps between 2014 and 2022. Common Crawl dumps may include urls from the early days of the internet. Hence, the download/collection timeframe does not match the creation timeframe. Additionally, future users of COMMONPOOL and its subsets will have to download data themselves using our tooling.

Q27 **Were any ethical review processes conducted (e.g., by an institutional review board)?** *If so, please provide a description of these review processes, including the outcomes, as well as a link or other access point to any supporting documentation.*

- Our dataset collection process iterates on the LAION-5B process, which found IRB review was not necessary as they "do not intervene with the people depicted in the data as well as the data being public." [129]. Additionally, the NeurIPS ethics review found no serious ethical issues with LAION-5B. We take even more stringent safety measures than the original LAION-5B dataset, in that we filter out data that is flagged as NSFW by our detection pipeline and blur detected faces in COMMONPOOL, automatically in our released download tooling. All this being said, a formal ethics review has not been conducted to date.

Q28 **Does the dataset relate to people?** *If not, you may skip the remaining questions in this section.*

- Yes. People may appear in the dataset. Detected faces are blurred when downloading COMMONPOOL with our tooling.

Q29 **Did you collect the data from the individuals in question directly, or obtain it via third parties or other sources (e.g., websites)?**

- We collect data from websites across the internet.

Q30 **Were the individuals in question notified about the data collection?** *If so, please describe (or show with screenshots or other information) how notice was provided, and provide a link or other access point to, or otherwise reproduce, the exact language of the notification itself.*

- Individuals were not notified about the data collection.

Q31 **Did the individuals in question consent to the collection and use of their data?** *If so, please describe (or show with screenshots or other information) how consent was requested and provided, and provide a link or other access point to, or otherwise reproduce, the exact language to which the individuals consented.*

- Following our usage of Common Crawl and https://github.com/rom1504/img2dataset for download images, we respect robots.txt files, which specify parts of websites that a crawler may access. It is, however, possible that images of people, medical images, etc. were uploaded to the internet without a person's consent. To mitigate against such safety concerns we make an effort to do rigorous NSFW filtering and blur all detected faces automatically in our download tooling.

Q32 **If consent was obtained, were the consenting individuals provided with a mechanism to revoke their consent in the future or for certain uses?** *If so, please provide a description, as well as a link or other access point to the mechanism (if appropriate).*

- In conjunction with LAION, we use https://laion.ai/dataset-requests/ to monitor user takedown requests. We will also make an effort to provide a user with the url at which their sensitive content is hosted—if they do not have this information already—, so they can take further action as they see fit (e.g., contacting the host to request that the content is taken down from the internet).

Q33 **Has an analysis of the potential impact of the dataset and its use on data subjects (e.g., a data protection impact analysis) been conducted?** *If so, please provide a description of this analysis, including the outcomes, as well as a link or other access point to any supporting documentation.*

- We conduct a fairness evaluation on models trained on COMMONPOOL and its derivative. See Appendix Q for details. Birhane et al. [15] conduct an extensive study in the context of LAION-400M, which is an image-text dataset also sourced from Common Crawl, finding a plethora of dangerous and unsafe content. Our dataset differs from LAION-400M in that we conduct NSFW preprocessing and face blurring for detected faces. COMMONPOOL only contains samples that pass our NSFW safety checks and our download tooling automatically blurs detected faces. However, since COMMONPOOL is created from the internet, it is still likely that it contains some harmful data.

Q34 **Any other comments?**

- We hope that future work will use COMMONPOOL to study how to construct safer, web-scale datasets.

## S.4 Preprocessing, Cleaning, and/or Labeling

Q35 **Was any preprocessing/cleaning/labeling of the data done (e.g., discretization or bucketing, tokenization, part-of-speech tagging, SIFT feature extraction, removal of instances, processing of missing values)?** *If so, please provide a description. If not, you may skip the remainder of the questions in this section.*

- Yes. See Q7. For more details see Appendix H.

**Q36 Was the "raw" data saved in addition to the preprocessed/cleaned/labeled data (e.g., to support unanticipated future uses)?** *If so, please provide a link or other access point to the "raw" data.*

- Raw data is not available or distributed due to safety considerations. We distribute only urls that are in the dataset on HuggingFace—and not urls of images our preprocessing flagged as NSFW.

**Q37 Is the software used to preprocess/clean/label the instances available?** *If so, please provide a link or other access point.*

- We use the following, open-source software to aid in data processing:
  - Apache Spark: https://spark.apache.org
  - Ray: https://www.ray.io
  - img2dataset: https://github.com/rom1504/img2dataset
  - OpenAI CLIP: https://github.com/openai/CLIP
  - Near dedulicate detector: https://github.com/lyakaap/ISC21-Descriptor-Track-1st
  - Face detector: https://github.com/deepinsight/insightface
  - Detoxify, for detecting toxic language: https://github.com/unitaryai/detoxify
  - A modified version of the following NSFW image detector: https://github.com/LAION-AI/CLIP-based-NSFW-Detector. Specifically, we use the dataset used to train this model to train our own 4-layer MLP classifier.

**Q38 Any other comments?**

- COMMONPOOL and DATACOMP would not be possible without tools developed by the open-source community.

### S.5 Uses

**Q39 Has the dataset been used for any tasks already?** *If so, please provide a description.*

- The full dataset (and subsets) have been used to train several CLIP models at various scales and compute budgets as presented in our main paper. We evaluate these models zero-shot on 38 downstream image classification and retrieval tasks. See Section 3.5 and Appendix O for more details.

**Q40 Is there a repository that links to any or all papers or systems that use the dataset?** *If so, please provide a link or other access point.*

- No. However, there is a leaderboard associated with DATACOMP. Interested parties can investigate the submissions and further study publications that make use of our data. See: https://www.datacomp.ai/leaderboard.html.

**Q41 What (other) tasks could the dataset be used for?**

- The dataset could also be used for training image captioning models and language-conditional image generation models. Note: generative image models trained on COMMONPOOL are not expected to generate recognizable human faces as our download tooling automatically blurs detected faces. COMMONPOOL could be used for sociological studies, for example, examining societal biases or to better understand what is on the public internet.

**Q42 Is there anything about the composition of the dataset or the way it was collected and preprocessed/cleaned/labeled that might impact future uses?** *For example, is there anything that a future user might need to know to avoid uses that could result in unfair treatment of individuals or groups (e.g., stereotyping, quality of service issues) or other undesirable harms (e.g., financial harms, legal risks) If so, please provide a description. Is there anything a future user could do to mitigate these undesirable harms?*

- COMMONPOOL and its derivatives are not intended for production ready products, including but not limited to those related to race, gender identity or expression, ethnicity, sexual orientation, age, socioeconomic status, disability, religion, national origin or creed. COMMONPOOL is not suitable for any software that makes decisions involving people. COMMONPOOL is collected from the internet and hence reflects many of the biases, unfairness, and stereotypes currently existing in our societies. COMMONPOOL is intended as a research artifact to study multimodal dataset curation and the effect of data curation strategies on downstream models.

**Q43 Are there tasks for which the dataset should not be used?** *If so, please provide a description.*

- COMMONPOOL in its current form or the subsets presented in this paper should not be used in software that makes decisions related to people. The known biases (Appendix Q) make deploying software, especially widely decimated production-level products, built on COMMONPOOL incredibly irresponsible. COMMONPOOL is designed as a research artifact for academic exploration. We also do not condone the use of COMMONPOOL in surveillance or military applications.

Q44 **Any other comments?**

- Our goal with COMMONPOOL and DATACOMP was to put a benchmark in place so the community can start measuring dataset progress along many different axes (e.g., model performance on diverse tasks). We believe this is crucial to develop more performant and safer datasets.

## S.6 Distribution

Q45 **Will the dataset be distributed to third parties outside of the entity (e.g., company, institution, organization) on behalf of which the dataset was created?** *If so, please provide a description.*

- Yes. We use HuggingFace datasets for public release.

Q46 **How will the dataset be distributed (e.g., tarball on website, API, GitHub)?** *Does the dataset have a digital object identifier (DOI)?*

- The dataset will be distributed via HuggingFace datasets at [https://huggingface.co/datasets/mlfoundations/datacomp_pools/tree/main](https://huggingface.co/datasets/mlfoundations/datacomp_pools/tree/main)

Q47 **When will the dataset be distributed?**

- DATACOMP will be available starting May 2023.

Q48 **Will the dataset be distributed under a copyright or other intellectual property (IP) license, and/or under applicable terms of use (ToU)?** *If so, please describe this license and/or ToU, and provide a link or other access point to, or otherwise reproduce, any relevant licensing terms or ToU, as well as any fees associated with these restrictions.*

- We distribute the url-text sample and metadata under a standard CC-BY-4.0 licence.

Q49 **Have any third parties imposed IP-based or other restrictions on the data associated with the instances?** *If so, please describe these restrictions, and provide a link or other access point to, or otherwise reproduce, any relevant licensing terms, as well as any fees associated with these restrictions.*

- We do not copyright samples in the dataset.

Q50 **Do any export controls or other regulatory restrictions apply to the dataset or to individual instances?** *If so, please describe these restrictions, and provide a link or other access point to, or otherwise reproduce, any supporting documentation.*

- The dataset is provided as an index of url-text pairs.

Q51 **Any other comments?**

- We provide several subsets of COMMONPOOL (between 12.8M samples and the full dataset of 12.8B samples). Hence, it is possible to download and experiment with subset of the data.

## S.7 Maintenance

Q52 **Who will be supporting/hosting/maintaining the dataset?**

- HuggingFace currently hosts the url-text pairs and metadata. The DATACOMP team will be responsible for maintaining the dataset.

Q53 **How can the owner/curator/manager of the dataset be contacted (e.g., email address)?**

- We can be contacted at `contact@datacomp.ai`.

Q54 **Is there an erratum?** *If so, please provide a link or other access point.*

- Currently there are no errata. If issues are discovered, we will communicate with the public via our website [https://datacomp.ai](https://datacomp.ai).

Q55 **Will the dataset be updated (e.g., to correct labeling errors, add new instances, delete instances)?** *If so, please describe how often, by whom, and how updates will be communicated to users (e.g., mailing list, GitHub)?*

- At the present time there is no intention to update COMMONPOOL for scientific reasons. However, we will respond to user takedown requests (see Q56). COMMONPOOL is inherently noisy and the purpose of releasing it is to encourage researchers in the community to study dataset cleaning in the context of image-text samples.

Q56 **If the dataset relates to people, are there applicable limits on the retention of the data associated with the instances (e.g., were individuals in question told that their data would be retained for a fixed period of time and then deleted)?** *If so, please describe these limits and explain how they will be enforced.*

- We will use the following website, https://laion.ai/dataset-requests, for user takedown requests, where "Sample ID" is the sample uid.

Q57 **Will older versions of the dataset continue to be supported/hosted/maintained?** *If so, please describe how. If not, please describe how its obsolescence will be communicated to users.*

- This is the first version of DATACOMP and the associated COMMONPOOL dataset. We do not intend to maintain deprecated version of COMMONPOOL. We will communicate deprication notices through our website: https://datacomp.ai.

Q58 **If others want to extend/augment/build on/contribute to the dataset, is there a mechanism for them to do so?** *If so, please provide a description. Will these contributions be validated/verified? If so, please describe how. If not, why not? Is there a process for communicating/distributing these contributions to other users? If so, please provide a description.*

- All alterations to the dataset will be handled on a case-by-case basis.

Q59 **Any other comments?**

- We encourage community members to contact us at contact@datacomp.ai with inquiries related to dataset maintainence.