# OpenReview forum: "DataComp: In search of the next generation of multimodal datasets"
_NeurIPS.cc/2023/Track/Datasets_and_Benchmarks — NeurIPS 2023 Datasets and Benchmarks Oral_

### Official Review · Reviewer_GJCx · 2023-07-16
**DataComp -- A useful work for the next generation of multimodal datasets**

**Rating:** 9
**Confidence:** 4
**Correctness:** Yes, the evaluation methods and exper…
**Clarity:** Yes, It' s easy to read and understand.

**Strengths:**

This work provide a public environment and massive resources  for the exploration of  the construction of an effective multi-modal dataset.
Results for over three hundred baseline experiments are provided, and evaluation is conducted on 38 downstream tasks.


**Additional Feedback:**

Good job!

**Documentation:**

Yes

**Ethics:**

The authors have considered the protection of privacy and the removal of harmful contents.

**Limitations:**

1, The training target is only suitable for  understanding  tasks, the resulted dataset may not be the best for generative tasks.
2, Only filtering-based methods are included.  However,  In some works, researchers  used a captioning model to augment image captions, like ** LIU, Yulong, ZHU, Guibo, ZHU, Bin, et al. TaiSu: A 166M Large-scale High-Quality Dataset for Chinese Vision-Language Pre-training. Advances in Neural Information Processing Systems, 2022, vol. 35, p. 16705-16717. **

**Opportunities For Improvement:**

As far as I am concerned, this is a complete work. The only problem might be that the training target is only suitable for  understanding  tasks, the resulted dataset may not be the best for generative tasks. In general, this is a valuable work.

**Relation To Prior Work:**

Yes.

**Summary And Contributions:**

This work provide DATACOMP, a new benchmark and competition for multimodal dataset design, aiming to find a smaller but more effective subset from  COMMONPOOL(a dataset of 12.8B image-text pairs collected from Common Crawl,  another contribution of this work.).  The authors have done comprehensive experiments on scaling trends and filtering strategies for dataset design. Based on two most promising filtering baselines, they provide DATACOMP-1B, a new state-of-the-art multimodal dataset, which enables training a CLIP ViT-L/14 model with fewer computational cost  but better zero-shot performances compared with  OpenAI’s original CLIP ViT-L/14.

---

> ### Author Response · Authors · 2023-08-22
> **Response to Reviewer GJCx**
>
> Thank you for your useful comments, and for engaging with us to improve our work. Please see our responses to your individual comments below, and please let us know if there is anything else we can do to further improve our paper.
>
> ***
>
> #### **Generation**
>
> Thank you for pointing out the use of our dataset for image generation tasks. In response to your comments, we ran new experiments on image generation by fine-tuning Stable Diffusion models on DataComp-1B. As also mentioned in our response to Reviewer pyn4, training models for image generation from scratch using our dataset, while an interesting avenue for future work, is unfortunately too costly to run in the time provided for the response. For example, training a Stable Diffusion model from scratch would take approximately 24,000 GPU hours / 50,000 USD (https://www.mosaicml.com/blog/stable-diffusion-2).
>
> Instead of training from scratch, we fine-tuned a Stable Diffusion model, starting from stable-diffusion-2-base and running for 4M samples from each of DataComp-1B and LAION-2B. Results on this preliminary experiment are shown below. FID is measured as shown here: https://github.com/j-min/DallEval/tree/main/quality. Average and standard deviation are reported across 3 runs of fine-tuning. The base model is not fine-tuned.
>
> | Data | FID score |
> |------|-----------|
> | DataComp-1B | 15.75 $\pm$ 1.20 |
> | LAION-2B	   | 15.81 $\pm$ 0.96 |
> | Stable Diffusion 2 Base | 16.46 |
>
> While this is a preliminary experiment and does not show a complete picture on training models for image generation using our data, it does show the potential of our dataset for image generation. We hope that we or others can explore this direction as part of future work.
>
>
> ***
>
> #### **Using captioning models to augment datasets.**
>
> We agree with the reviewer’s suggestion. In our paper we wanted to provide strong baselines for our benchmark, but also wanted to leave more complex avenues of investigation (e.g., figuring out how best to use captioning models to augment datasets) to future participants. **Since DataComp was released, the suggested idea has been successfully validated by follow-up work**. In particular, Ngyuen et al. [1] showed how automatically generated captions can be used to improve image-text datasets. Their best method using captions generated by BLIP2 outperform the best filtering method proposed in our paper by 2 percentage points on ImageNet and 4 percentage points on average across the 38 tasks at the medium scale. They also show large improvements on retrieval, specifically on the Flickr and MS-COCO datasets. Overall, their experiments and results empirically confirm that generated captions can be useful sources of augmentation. Also thank you for the reference. We **added the reference** to our related work section on large-scale multimodal datasets (L108).
>
> ***
>
> #### **References**
>
> [1] Nguyen, Thao, et al. "Improving Multimodal Datasets with Image Captioning." arXiv preprint https://arxiv.org/abs/2307.10350, (2023). https://arxiv.org/abs/2307.10350.

---

> > ### Comment · Reviewer_GJCx · 2023-08-25
> >
> > Thanks for your reply. I believe this work will benifit the VLP community very much.

---

### Official Review · Reviewer_N9Hd · 2023-07-17
**The first step in finding the next generation of multimodal datasets is here!**

**Rating:** 9
**Confidence:** 4
**Clarity:** Yes. The paper is clear and easy to r…

**Strengths:**

### Strengths

The topic that this paper addresses is becoming increasingly relevant with the advent of new foundation models and open-source models. A lot of papers have investigated model architecture, training methods, and optimization techniques, yet only a few focus on datasets. The recent trend in scaling up models requires massive amounts of data to be put into the models, but the quality of this data has received less attention. For example, from the perspective of language models, we know that high-quality data can lead to better models [1, 2]. This paper focuses on visual language models, in particular CLIP [3], to investigate the quality and importance of the data, something that feels missing from the literature.

The evaluation conducted in this paper is extensive, containing over 300 baselines over four different scaling power and seven filtering methods. The results leads to introduction of `DataComp-1B` dataset that is not only smaller than `LAION-2B` [4] but also lead to training CLIP-L/14 model that outperforms its LAION variant with $6.1$pp.

The amount of consideration put into this paper is exemplary. Each section in the main text is accompanied by a corresponding section in the supplementary material, which provides detailed explanations and further insights. Additionally, the paper provides an analysis of fairness and bias in the trained models which is extremely valuable.

### References:
- [1] - Zhou, Chunting, et al. "Lima: Less is more for alignment." arXiv preprint arXiv:2305.11206 (2023).
- [2] - Mukherjee, Subhabrata, et al. "Orca: Progressive learning from complex explanation traces of gpt-4." arXiv preprint arXiv:2306.02707 (2023).
- [3] - Radford, Alec, et al. "Learning transferable visual models from natural language supervision." International conference on machine learning. PMLR, 2021.
- [4] - Schuhmann, Christoph, et al. "Laion-5b: An open large-scale dataset for training next generation image-text models." Advances in Neural Information Processing Systems 35 (2022): 25278-25294.

**Additional Feedback:**

- Is there any reason why the leaderboard contains only the baseline models?
- From my perspective, the `small` scale is the most important, as it is something that most students can afford to run. My only concern is the consistency between the `small` and larger scales. I liked (and appreciated) the analysis conducted in section 5.2, but your filtering methods seem fairly simple. Do we have strong evidence that this trend holds across other filtering processes? btw, why there is not a `small` vs `xlarge` in Table 21?

**Correctness:**

Yes. The construction of the dataset is based on filtering `CommonCrawl`, which is a pretty standard way to create a large dataset these days. All the models trained in this paper use a fixed training code and are compared against each other on several external validation sets (e.g., ImageNet). During the filtering process, the authors made sure that these validation sets were excluded from the training set.

**Documentation:**

The main text provides an overview of how the dataset is constructed (both `CommonPool` and `DataComp-1B`), and the appendix contains full details about each filtering step, as well as information about the dataset sheet.
All additional details can be found on the [Project's homepage](https://www.datacomp.ai/) or the [Github page](https://github.com/mlfoundations/datacomp).

**Ethics:**

No. The dataset underwent several filtering processes, for example, the removal of NSFW content and blurring of individuals' faces. I don't see any major concerns here.

**Limitations:**

Yes. The limitations section of the paper already addressed possible impacts.

**Opportunities For Improvement:**

The experiment conducted in this paper is extremely valuable and perhaps very expensive, but I feel the filtering processes used in this paper are pretty simple and not novel; they are just a collection of things the research community has explored before. I understand the main objective of this paper is something else, but it would be a good addition to have more interesting filtering (such as controlling for number of a particular object or animal in the dataset, number of objects in each image, positioning of objects inside the image, etc).


**Relation To Prior Work:**

Yes. The paper has cited previous contributions.


**Summary And Contributions:**

### Summary

The main goal of this paper is to discuss dataset curation and the importance of datasets in multimodal models. To this end, the authors have introduced DataComp, a new benchmark and testbed for dataset curation experiments. In this benchmark, the model architect and the training code are fixed, and the participant should find the suitable data, either by applying some filtering mechanism to the provided dataset `CommonPool` or by bringing their own data. This benchmark contains multiple training scales, ranging from *small* (suitable for low-budget tests) up to *xlarge* (industry scale budget).

To iterate on the main contributions of the paper:

1. Introduction of the DataComp benchmark for designing multimodal (image-text) datasets.
2. Introduction of `CommonPool`, a curated dataset comprising more than 12 billion image-text pairs.
3. Extensive evaluation and experiments conducted using various filtering settings.
4. Study of consistency trends across different scales.
5. Introduction of `DataComp-1B`, a subset of `CommonPool`, which enables the training of CLIP models that outperform those trained on the `LAION` [1] and `OpenAI’s WIT` [2] datasets.


### References:
- [1] - Schuhmann, Christoph, et al. "Laion-5b: An open large-scale dataset for training next generation image-text models." Advances in Neural Information Processing Systems 35 (2022): 25278-25294.
- [2] - Radford, Alec, et al. "Learning transferable visual models from natural language supervision." International conference on machine learning. PMLR, 2021.

---

> ### Author Response · Authors · 2023-08-22
> **Response to Reviewer N9Hd (part 1/2)**
>
> Thank you for your encouraging review. We address specific comments below, and please let us know if there is anything else we can help clarify or do to further strengthen our paper.
>
> ***
>
> #### **More complex filtering methods: controlling for object classes, number of objects, and object positions.**
>
> Thank you for the valuable suggestion! In response we **added new experiments and discussion to the manuscript (Appendix M)**. While controlling for such factors is common in the supervised settings, experimenting with analogous strategies in the context of multimodal datasets and CLIP training is a pertinent direction. Inspired by your comments, we use the Detic detector [1] to annotate the medium pool (128M sample pool) by extracting bounding boxes and class labels for the 1203 LVIS [2] objects categories. Following the original Detic paper, we retain predictions whose confidence score exceeds 0.5. Based on these annotations, we construct the following **five new strategies:**
>
> - **Object exists:** Subset for which there exists at least one detection from the 1203 LVIS categories.
> - **Object centered:** Subset for which there exists at least one detection from the 1203 LVIS categories with a bounding box center falling in the center grid cell of a 3x3 grid superimposed on the image.
> - **Balancing by class:** We define 1204 buckets—1203 buckets corresponding to the LVIS classes and an additional bucket for images that do not have any detections. For each image in the medium pool, we assign the image to the bucket(s) corresponding to the detected classes. We then construct a dataset such that there are an equal number of samples from each bucket and the total number of samples specified by a particular scale (e.g., 128M samples for medium scale). Note, for rare classes there can be many repeated samples and for common classes only a subset of the total samples will be in the dataset.
> - **Balancing by position:** We define 26 buckets—0, 2, …, 24 corresponding to 5x5 grid locations in an image. An image is added to bucket(s) when it contains a bounding box whose center falls in the bucket’s grid cell. The 25th bucket contains images for which there are no detections. We again construct a dataset such that there are an equal number of samples from each bucket.
> - **Balancing by count:** We define 12 buckets—0, 1, 2, …, 10 corresponding to zero to ten detections in an image and a twelfth bucket corresponding to images with more than ten detections. We yet again construct a dataset such that there are an equal number of samples from each bucket.
>
> We employ each of these strategies at the medium scale. Since the above strategies can be composed with any starting pool, we additionally apply each of the above Detic-based strategies to our previous best medium scale filtered pool: Image-based ∩ CLIP score (L/14 30%). This yields a total of 10 new datasets.
>
> Our results are summarized in the table below. In summary: 1) The Image-based ∩ CLIP score (L/14 30%) baseline still performs best. 2) Balancing data in the context of multimodal CLIP training remains an open problem. All balancing strategies lead to divergence of the CLIP contrastive loss and result in poor model performance. We hypothesize that this is due to the long-tailed nature of the data distribution, which leads to many repeated samples in our balanced data construction. This in turn, increases the likelihood that samples are contrasted with themselves in the loss computation.
>
> |                                               | ImageNet | Average over 38 datasets |
> |-----------------------------------------------|----------|--------------------------|
> | Baseline: Nofilter                            | 0.176    | 0.258                    |
> | &nbsp;&nbsp;&nbsp;&nbsp;∩ Object exists                               | 0.181    | 0.263                    |
> | &nbsp;&nbsp;&nbsp;&nbsp;∩ Object centered                             | 0.187    | 0.263                    |
> | &nbsp;&nbsp;&nbsp;&nbsp;∩ Balancing by class                          | 0.038    | 0.141                    |
> | &nbsp;&nbsp;&nbsp;&nbsp;∩ Balancing by position                       | 0.040    | 0.148                    |
> | &nbsp;&nbsp;&nbsp;&nbsp;∩ Balancing by object count                   | 0.127    | 0.221                    |
> | Baseline: Image-based ∩ CLIP score (L/14 30%) | 0.297    | 0.328                    |
> | &nbsp;&nbsp;&nbsp;&nbsp;∩ Object exists                               | 0.289    | 0.319                    |
> | &nbsp;&nbsp;&nbsp;&nbsp;∩ Object centered                             | 0.247    | 0.286                   |
> | &nbsp;&nbsp;&nbsp;&nbsp;∩ Balancing by class                          | 0.034    | 0.136                    |
> | &nbsp;&nbsp;&nbsp;&nbsp;∩ Balancing by position                       | 0.036    | 0.136                    |
> | &nbsp;&nbsp;&nbsp;&nbsp;∩ Balancing by object count                   | 0.068    | 0.169                    |
>
> ***

---

> > ### Author Response · Authors · 2023-08-22
> > **Response to Reviewer N9Hd (part 2/2)**
> >
> > #### **DataComp leaderboard**
> >
> > Thanks for the question. At the time of our submission, DataComp was relatively new (it appeared on arxiv at the end of April) and the only methods implemented were the strategies from our paper, which we called baselines. Since then, we have seen new submissions, which are now on the leaderboard (https://www.datacomp.ai/leaderboard.html). For example, T-MARS [3] improves medium scale filtering track ImageNet zero-shot numbers to 0.330, which is a 3.3 percentage point improvement over the prior state-of-the-art at this scale. Likewise work from Nguyen et al. [4] improves large scale BYOD track ImageNet zero-shot numbers to 0.643, which is a 2.2 percentage point improvement.
> >
> > ***
> >
> > #### **Consistency in data curation strategies across scales for more complex strategies.**
> >
> > Thank you for acknowledging our effort to understand the consistency of various methods across compute scales. Results from Nguyen et al. [4] point towards consistency across scales for more complex data curation strategies. In particular, Nguyen et al. experiment with using synthetic captions to augment data from CommonPool. Their method yields the 2nd best strategy at the small scale and the best known strategy at medium and large scales in the BYOD track.
> >
> > ***
> >
> > #### **Correlation between small and xlarge scale results.**
> >
> > Thank you for the question, which we agree is important. We originally did not present the small vs xlarge correlation in the paper due to the small number of experiments we ran at the xlarge scale (which are computationally expensive with about 40,000 A100 hours per training run). However, we present rank correlation (Spearman's rank correlation coefficient) in the table below.
> >
> >
> > | Metric            | small vs xl| medium vs xl| large vs xl|
> > |-------------------|-------------|------------|------------|
> > | ImageNet acc.     | 0.638       | 0.986      | 1.000      |
> > | Avg. pref. metric | 0.574       | 0.794      | 0.986      |
> >
> > While the correlation between small and xlarge is 0.638 for ImageNet and 0.574 for average, these numbers increase substantially for the medium and large scales. We would also like to point out that the medium scale is feasible for many academic labs. To put this in context, training at the small scale corresponds roughly to fine-tuning a model on ImageNet (e.g., fine-tuning for 10 epochs or ~4 A100 hours), while training a medium scale model corresponds to training on ImageNet from scratch (e.g., training for 100 epochs or ~40 A100 hours). Lastly, Figure 21 suggests that while there are some filtering strategies that do well at larger scales that would be difficult to find at smaller scales, better filtering strategies at smaller scales typically translate to better filtering strategies at larger scales.
> >
> > ***
> >
> > #### **References**
> >
> > [1] Zhou, Xingyi, et al. “Detecting twenty-thousand classes using image-level supervision.” ECCV (2022). https://arxiv.org/abs/2201.02605.
> >
> > [2] Gupta, Agrim, et al. “”LVIS: A dataset for large vocabulary instance segmentation.” CVPR (2019). https://arxiv.org/abs/1908.03195.
> >
> > [3] Maini, Pratyush, et al. “T-MARS: Improving Visual Representations by
> > Circumventing Text Feature Learning.” arXiv preprint arXiv:2307.03132 (2023). https://arxiv.org/abs/2307.03132.
> >
> > [4] Nguyen, Thao, et al. "Improving Multimodal Datasets with Image Captioning." arXiv preprint arXiv:2307.10350 (2023). https://arxiv.org/abs/2307.10350.

---

> > > ### Comment · Reviewer_N9Hd · 2023-08-26
> > >
> > > I want to thank the authors for their thoughtful response and the new experiments conducted in Section M of the appendix. I am confident that the community will benefit a lot from this paper; the insights and results derived from the experiments are invaluable.

---

### Official Review · Reviewer_pyn4 · 2023-07-19
**Exceptional large scale dataset with high quality analysis and performance**

**Rating:** 9
**Confidence:** 5
**Clarity:** The paper is well written.

**Strengths:**

(1) The proposed CommonPool is the largest openly available dataset for training vision-and-language models.

(2) The data collection and filtering process has been comprehensively discussed to support community investigation and further improvement.

(3) Experiments support that CLIP models trained on DataComp-1B exceed the performance of original CLIP models trained on the private dataset.

(4) Over 300 baseline experiments are conducted to ensure the quality of their dataset and the validity of their conclusions.

**Additional Feedback:**

Great paper. I believe this paper ranks at least the top 15% of the whole submissions.

**Correctness:**

The dataset creation, testing, and evaluation methodology are all sound. This paper appears correct about its claims.


**Documentation:**

The dataset and processing pipeline is open-source and well-documented.

**Ethics:**

The authors have discussed two major preprocessing steps implemented to ensure the safety and privacy of the datasets:

Safety Preprocessing: The authors took rigorous measures to remove unsafe content from the Common Crawl data. They employed Detoxify to remove samples containing unsafe text, such as explicit or threatening language. For images with explicit content, they trained a classifier using the NSFW dataset and CLIP ViT-L/14 features. This resulted in the removal of approximately 19% of image-text pairs, reducing the initial sample size from around 16.8 billion to approximately 13.6 billion.

Face Detection and Blurring: To preserve individual privacy, they used a face detector to find and blur faces in the images. The authors confirmed that this blurring process had minimal impact on the performance of the models, echoing observations made by other researchers.

**Limitations:**

The authors adequately addressed the limitations and potential negative societal impact of their work.

**Opportunities For Improvement:**

(1) The task coverage could have been expanded. For example, VQA and image generation are not considered in the paper.

(2) The authors cited a relevant work and conducted experiments to show that face blurring does not lead to performance degradation of a trained model. However, this is likely because the task coverage is limited. If other tasks as image generation are considered, face blurring does have detrimental effect to the performance.

(3) The model coverage could have also been expanded, as only CLIP is considered. Image-conditioned language models and image generation models could have been explored.

**Relation To Prior Work:**

The paper clearly discusses how it relates to prior work.

**Summary And Contributions:**

The paper introduces five major contributions related to the development of multimodal datasets:

The main contribution is the development of DataComp, a benchmark for multimodal dataset design. Unlike traditional benchmarks where the dataset is fixed and researchers propose new algorithms, DataComp encourages innovation in creating new training sets. Models are evaluated based on a testbed of various classification and retrieval tasks.

The paper also introduces CommonPool, a dataset consisting of 12.8 billion image-text pairs collected from the Common Crawl. This is currently the largest public image-text dataset and is aimed at improving the safety of image-text datasets.

The authors investigate scaling trends in dataset design. DataComp includes four scales, ranging from 12.8M to 12.8B samples, allowing researchers with different resources to participate in the benchmark.

The paper presents over 300 baseline experiments, including various filtering techniques. One of the key findings is that smaller, more rigorously filtered datasets can often lead to models that generalize better than those trained on larger datasets from the same pool.

Lastly, the paper introduces DataComp-1B, a new state-of-the-art multimodal dataset obtained by combining their two most promising filtering baselines. The new dataset enabled them to train a model to an ImageNet zero-shot accuracy of 79.2%, which is a significant computational cost reduction compared to training larger models and it outperforms the original CLIP model by OpenAI.

---

> ### Author Response · Authors · 2023-08-22
> **Response to Reviewer pyn4 (part 1/2)**
>
> Thank you for your thoughtful feedback and remarks. In response to your comments, we ran several new experiments, which we hope will make our paper stronger. Our responses are detailed below.
>
> ***
>
> #### **Task coverage**
>
> Thank you for your suggestion! We agree about the importance of a comprehensive evaluation suite, and ran additional experiments in light of your comments. Our original evaluation suite already consists of dozens of tasks on diverse domains, including object recognition, counting, distance prediction, geolocation, texture classification, satellite imagery recognition, classification tasks in the medical domain, scene recognition, image retrieval, text retrieval and commonsense association. In addition, we also provided multiple bias and fairness analyses.
>
> In response to your comments, **we ran additional VQA evaluations** for all models we trained. More specifically, following Shen et al [1], we use CLIP models to contrast images with prompts formed by the questions and each candidate answer, without fine-tuning (as done for our other DataComp evaluations). Using the VQA v1 dataset [2], for each candidate answer, we construct a text prompt that also includes the question following the template “Question: [question text] Answer: [answer text]”. This text is fed to CLIP’s text encoder. As previously noted by multiple authors, CLIP models struggle on this task, potentially due to the mismatch between the text in the downstream task and the captions seen during pre-training [1,3,4]. Nonetheless, **we observe a strong correlation between VQA performance and ImageNet accuracy (0.877)** and between VQA performance and average accuracy on the rest of our evaluation suite (0.872). **Figure 17 in Appendix O.1** of our updated manuscript shows our full results.
>
> In addition to VQA, we also ran new experiments on image generation, as detailed in the “model coverage / image generation” section below.
>
> ***
>
> #### **Face blurring**
>
> Thank you for bringing up that face blurring a dataset can affect the quality of image generation. As pointed out, our experiments and previous literature suggest that such blurring has minimal effect on recognition tasks like ImageNet (Yang et al., 2021 [5], and our Appendix G). However, we agree that the effects of this design decision may be amplified in generative settings. Our competition is primarily focused on discriminative tasks, and when designing our dataset, we wanted to prioritize safety by better protecting the privacy of individuals through automatically blurring faces in our download tooling. That being said, we acknowledge that there are trade-offs between protecting people’s privacy and creating generative models that can reliably generate faces. We added this discussion in our Appendix G and hope others will consider these trade-offs when curating datasets of their own.
>
> ***
>
> #### **Model coverage and image generation**
>
> Thank you for pointing out the possibility of using models other than CLIP to evaluate our dataset. For the rebuttal, we additionally used our data for image generation, and **ran new experiments on image generation by fine-tuning Stable Diffusion models on DataComp-1B**. We note that training models for image generation from scratch, while an interesting avenue for future work, is unfortunately too costly to run in the time provided for the response. For example, training a Stable Diffusion model from scratch would take approximately 24,000 GPU hours / 50,000 USD. (https://www.mosaicml.com/blog/stable-diffusion-2).
>
> Instead of training from scratch, we fine-tuned a Stable Diffusion model, starting from stable-diffusion-2-base and running for 4M samples from each of DataComp-1B and LAION-2B. Results on this preliminary experiment are shown below. FID is measured as shown here: https://github.com/j-min/DallEval/tree/main/quality. Average and standard deviation are reported across 3 runs of fine-tuning. The base model is not fine-tuned.
>
> | Data | FID score |
> |------|-----------|
> | DataComp-1B | 15.75 $\pm$ 1.20 |
> | LAION-2B	   | 15.81 $\pm$ 0.96 |
> | Stable Diffusion 2 Base | 16.46 |
>
> While this is a preliminary experiment and does not show a complete picture on training models for image generation using our data, it does show the potential of our dataset for image generation. We hope that we or others can explore this direction as part of future work.
>
> ***

---

> > ### Author Response · Authors · 2023-08-22
> > **Response to Reviewer pyn4 (part 2/2)**
> >
> > #### **References**
> >
> > [1] Shen, Sheng, et al. "How much can clip benefit vision-and-language tasks?." arXiv preprint arXiv:2107.06383 (2021). https://arxiv.org/abs/2107.06383.
> >
> > [2] Antol, Stanislaw, et al. "Vqa: Visual question answering." Proceedings of the IEEE international conference on computer vision. 2015. https://arxiv.org/abs/1505.00468.
> >
> > [3] Ilharco, Gabriel, et al. "Patching open-vocabulary models by interpolating weights." Advances in Neural Information Processing Systems 35 (2022): 29262-29277. https://arxiv.org/abs/2208.05592.
> >
> > [4] Song, Haoyu, et al. "Clip models are few-shot learners: Empirical studies on vqa and visual entailment." arXiv preprint arXiv:2203.07190 (2022). https://arxiv.org/abs/2203.07190.
> >
> > [5] Yang, Kaiyu, et al. "A study of face obfuscation in imagenet." International Conference on Machine Learning. PMLR, 2022.

---

> ### Author Response · Authors · 2023-08-28
> **End of the author / reviewer discussion period**
>
> Dear Reviewer pyn4,
>
> We would like to thank you again for your thoughtful feedback and remarks. This is just a brief reminder that the author / reviewer discussion period ends in about 23h (Aug 29 8 pm UTC / 1 pm PT). In case you have any remaining questions, we would be happy to respond promptly.
>
> Best,
> The DataComp Team

---

### Official Review · Reviewer_TaGm · 2023-07-23
**Data-centric approach to multi-modal data, many contributions**

**Rating:** 8
**Confidence:** 3
**Correctness:** Seems good.
**Clarity:** Yes.

**Strengths:**

1. Data centric approach is good.
2. Big and comprehensive paper, with a lot packed into it.
3. Love the attention to multi-modal data, definitely an under-explored area
4. Cool results

**Additional Feedback:**

I have increased my score to 8. I think this is super interesting work, and the fact that the underlying datasets are problematic is important to explicitly acknowledge. This is a tremendous resource but it comes with very real risks.

**Documentation:**

I couldn't tell anything about licensing of the dataset in the paper or by poking around on the website. Is it CC-BY? Unknown? etc.

**Ethics:**

Good news: The authors paid careful attention to issues of safety, PII filtering, etc.

Bad news: I see very little / no information on the licensing status of the datasets discussed.

**Limitations:**

Licensing. I hate to sound like a broken record on this topic.

**Opportunities For Improvement:**

Licensing - please tell the reader the license for the data!

How do you handle BYOD? Seems like there's lots of potential risks, would love to read about that.

**Relation To Prior Work:**

Reasonably so.

**Summary And Contributions:**

1. Benchmark competition for multi-modal dataset design and a variety of results from that. The data-centric competition idea doesn't appear to be new, but the focus on multi-modal data is new and the scale is novel.

2. Common pool image-text dataset and a dataset filtering competition on it.

3. Look at 'neural scaling laws for data', for the above dataset and filtering.

4. Many experiments and neat results

5. A new dataset created by applying the best filtering approaches.

---

> ### Comment · Reviewer_TaGm · 2023-08-21
> **Early response?**
>
> Hi DataComp team - Just as a heads up, I will be on vacation with no access to internet starting August 23 for roughly two weeks until September 5. I realize this means you have a somewhat shorter period to respond. If feasible, I would love to have an earlier response so I can incorporate your comments and feedback into my final review and give them full weight.
>
> Thanks :)

---

> > ### Author Response · Authors · 2023-08-25
> > **Early response**
> >
> > Dear Reviewer TaGM,
> >
> > Thank you again for your kind reminder and offer to increase your score if we address the licensing question. We hope you are having a good trip! We posted our response to your concerns on August 21 (see below). We would be very grateful if you get a chance to respond and / or adjust your score, and would be happy to provide further clarifications if requested.
> >
> > Safe travels!
> >
> > Best,
> >
> > The DataComp Team

---

> ### Author Response · Authors · 2023-08-28
> **End of the author / reviewer discussion period**
>
> Dear Reviewer TaGM,
>
> This is just a brief reminder that the author / reviewer discussion period ends in about 23h (Aug 29 8 pm UTC / 1 pm PT). We understand that you are traveling right now and may not be able to participate in the discussion period. We would still very much appreciate it if you could revisit your review and score after your return in light of our response particularly regarding the licensing questions you raised.
>
> Thank you for your feedback and safe travels!
>
> Best,
>
> The DataComp Team

---

### Author Response · Authors · 2023-08-22
**Overall response**

We thank the reviewers for their time, constructive comments, and positive feedback. We are particularly grateful for reviewers pointing out that “the amount of consideration put into this paper is exemplary” (N9Hd), that the paper is “great” (pyn4)”,  “big and comprehensive” (TaGm), and that “the work is complete” (GJCx). All reviewers appreciated our experimentation, with many reviewers citing in particular the 300+ baselines that we ran (pyn4, N9Hd, GJCx). All reviewers also noted our attention to dataset safety (e.g., NSFW removal and face blurring).

Here we summarize the major additions in response to the reviews. **We have uploaded a revised draft which includes new experiments and clarifications as suggested by reviewers.** We added details to the paper and data repository about licensing (i.e., we distribute image url-text pairs under a CC-BY-4.0 license). Moreover, we added evaluation on VQA to the paper, finding strong positive correlation with performance on ImageNet. We also explored applications to image generation for purposes of the rebuttal. While our competition focuses on discriminative tasks, we acknowledge that text-conditional image generation is a critical use case for multimodal datasets, and find that fine-tuning StableDiffusion on DataComp-1B is competitive with fine-tuning on LAION-2B. We also added more complex filtering baselines accounting for object categories, object count, and object position to the paper.

**We address all reviewer feedback in more detail in the comments below.** The reviews have already improved our paper substantially and we believe that additional feedback will further strengthen our work. We are eager to engage further with the reviewers and are happy to provide written clarification or additional empirical support!

---

### Decision · Program_Chairs · 2023-09-22

**Decision:**

Accept (Oral)

**Comment:**

- This paper presents DataComp, a new image-text dataset and benchmark designed to offer new avenues for research and evaluation.
- Both reviewers and the Area Chair agree that this work promises significant contributions to the vision-language research community. Not only does it provide a large-scale dataset, but it also includes a comprehensive benchmark and leaderboard, as well as an extensive set of baseline experiments.
- The authors have engaged in the discussion with reviewers to address questions and concerns, including dataset licensing and specific requirements for image generation, etc.
- Considering the reviews and the potential impact of the paper, the Area Chair strongly recommends acceptance of the paper.